# Kernel Alignment Risk Estimator:
# Risk Prediction from Training Data

**Arthur Jacot**
Ecole Polytechnique Fédérale de Lausanne
`arthur.jacot@epfl.ch`

**Berfin Şimşek**
Ecole Polytechnique Fédérale de Lausanne
`berfin.simsek@epfl.ch`

**Francesco Spadaro**
Ecole Polytechnique Fédérale de Lausanne
`francesco.spadaro@epfl.ch`

**Clément Hongler**
Ecole Polytechnique Fédérale de Lausanne
`clement.hongler@epfl.ch`

**Franck Gabriel**
Ecole Polytechnique Fédérale de Lausanne
`franck.gabriel@epfl.ch`

## Abstract

We study the risk (i.e. generalization error) of Kernel Ridge Regression (KRR) for a kernel $K$ with ridge $\lambda > 0$ and i.i.d. observations. For this, we introduce two objects: the Signal Capture Threshold (SCT) and the Kernel Alignment Risk Estimator (KARE). The SCT $\vartheta_{K,\lambda}$ is a function of the data distribution: it can be used to identify the components of the data that the KRR predictor captures, and to approximate the (expected) KRR risk. This then leads to a KRR risk approximation by the KARE $\rho_{K,\lambda}$, an explicit function of the training data, agnostic of the true data distribution. We phrase the regression problem in a functional setting. The key results then follow from a finite-size analysis of the Stieltjes transform of general Wishart random matrices. Under a natural universality assumption (that the KRR moments depend asymptotically on the first two moments of the observations) we capture the mean and variance of the KRR predictor. We numerically investigate our findings on the Higgs and MNIST datasets for various classical kernels: the KARE gives an excellent approximation of the risk, thus supporting our universality assumption. Using the KARE, one can compare choices of Kernels and hyperparameters directly from the training set. The KARE thus provides a promising data-dependent procedure to select Kernels that generalize well.

## 1 Introduction

Kernel Ridge Regression (KRR) is a widely used statistical method to learn a function from its values on a training set [27, 29]. It is a non-parametric generalization of linear regression to infinite-dimensional feature spaces. Given a positive-definite kernel function $K$ and (noisy) observations $y^\epsilon$ of a true function $f^*$ at a list of points $X = x_1, \ldots, x_N$, the $\lambda$-KRR estimator $\hat{f}^\epsilon_\lambda$ of $f^*$ is defined by

$$\hat{f}^\epsilon_\lambda(x) = \frac{1}{N} K(x, X) \left( \frac{1}{N} K(X, X) + \lambda I_N \right)^{-1} y^\epsilon,$$

where $K(x, X) = (K(x, x_i))_{i=1,..,N} \in \mathbb{R}^N$ and $K(X, X) = (K(x_i, x_j))_{i,j=1,..,N} \in \mathbb{R}^{N \times N}$.

Despite decades of intense mathematical progress, the rigorous analysis of the generalization of kernel methods remains a very active and challenging area of research. In recent years, many new

kernels have been introduced for both regression and classification tasks; notably, a large number of kernels have been discovered in the context of deep learning, in particular through the so-called Scattering Transform [22], and in close connection with deep neural networks [7, 17], yielding ever-improving performance for various practical tasks [1, 10, 18, 28]. Currently, theoretical tools to select the relevant kernel for a given task, i.e. to minimize the generalization error, are however lacking.

While a number of bounds for the risk of Linear Ridge Regression (LRR) or KRR [6, 15, 31, 23] exist, most focus on the rate of convergence of the risk: these estimates typically involve constant factors which are difficult to control in practice. Recently, a number of more precise estimates have been given [21, 9, 24, 20, 5]; however, these estimates typically require a priori knowledge of the data distribution. It remains a challenge to have estimates based on the training data alone, enabling one to make informed decisions on the choices of the ridge and of the kernel.

## 1.1 Contributions

We consider a generalization of the KRR predictor $\hat{f}_\lambda^\epsilon$: one tries to reconstruct a true function $f^*$ in a space of continuous functions $\mathcal{C}$ from noisy observations $y^\epsilon$ of the form $(o_1(f^*) + \epsilon e_1, \ldots, o_N(f^*) + \epsilon e_N)$, where the observations $o_i$ are i.i.d. linear forms $\mathcal{C} \to \mathbb{R}$ sampled from a distribution $\pi$, $\epsilon$ is the level of noise, and the $e_1, \ldots, e_N$ are centered of unit variance. We work under the universality assumption that, for large $N$, only the first two moments of $\pi$ determine the behavior of the first two moments of $\hat{f}_\lambda^\epsilon$. We obtain the following results:

1. We introduce the Signal Capture Threshold (SCT) $\vartheta(\lambda, N, K, \pi)$, which is determined by the ridge $\lambda$, the size of the training set $N$, the kernel $K$, and the observations distribution $\pi$ (more precisely, the dependence on $\pi$ is only through its first two moments). We give approximations for the expectation and variance of the KRR predictor in terms of the SCT.

2. Decomposing $f^*$ along the kernel principal components of the data distribution, we observe that in expectation, the predictor $\hat{f}_\lambda^\epsilon$ captures only the signal along the principal components with eigenvalues larger than the SCT. If $N$ increases or $\lambda$ decreases, the SCT $\vartheta$ shrinks, allowing the predictor to capture more signal. At the same time, the variance of $\hat{f}_\lambda^\epsilon$ scales with the derivative $\partial_\lambda \vartheta$, which grows as $\lambda \to 0$, supporting the classical bias-variance tradeoff picture [14].

3. We give an explicit approximation for the expected MSE risk $R^\epsilon(\hat{f}_\lambda^\epsilon)$ and empirical MSE risk $\hat{R}^\epsilon(\hat{f}_\lambda^\epsilon)$ for an arbitrary continuous true function $f^*$. We find that, surprisingly, the expected risk and expected empirical risk are approximately related by

$$\mathbb{E}[R^\epsilon(\hat{f}_\lambda^\epsilon)] \approx \frac{\vartheta(\lambda)^2}{\lambda^2} \mathbb{E}[\hat{R}^\epsilon(\hat{f}_\lambda^\epsilon)].$$

4. We introduce the Kernel Alignment Risk Estimator (KARE) as the ratio $\rho$ defined by

$$\rho(\lambda, N, y^\epsilon, G) = \frac{\frac{1}{N}(y^\epsilon)^T \left(\frac{1}{N}G + \lambda I_N\right)^{-2} y^\epsilon}{\left(\frac{1}{N}\mathrm{Tr}\left[\left(\frac{1}{N}G + \lambda I_N\right)^{-1}\right]\right)^2},$$

where $G$ is the Gram matrix of $K$ on the observations. We show that the KARE approximates the expected risk; unlike the SCT, it is agnostic of the true data distribution. This result follows from the fact that $\vartheta(\lambda) \approx 1/m_G(-\lambda)$, where $m_G(z) = \mathrm{Tr}\left[\left(\frac{1}{N}G - zI_N\right)^{-1}\right]$ is the Stieltjes Transform of the Gram matrix.

5. Empirically, we find that the KARE predicts the risk on the Higgs and MNIST datasets. We see empirically that our results extend extremely well beyond the Gaussian observation setting, thus supporting our universality assumption (see Figure 1).

Our proofs (see the Appendix) rely on a generalized and refined version of the finite-size analysis of [16] of generalized Wishart matrices, obtaining sharper bounds and generalizing the results to operators. Our analysis relies in particular on the complex Stieltjes transform $m_G(z)$, evaluated at $z = -\lambda$, and on fixed-point arguments.

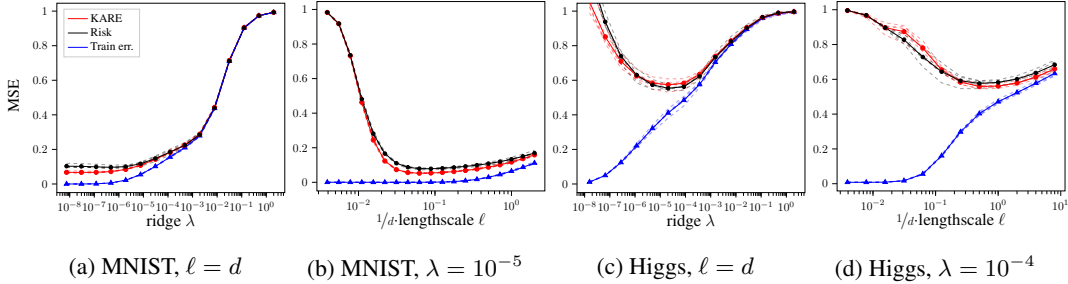

| | | | |
|---|---|---|---|
| (a) MNIST, $\ell = d$ | (b) MNIST, $\lambda = 10^{-5}$ | (c) Higgs, $\ell = d$ | (d) Higgs, $\lambda = 10^{-4}$ |

Figure 1: Comparison between the KRR risk and the KARE for various choices of normalized lengthscale $\ell/d$ and ridge $\lambda$ on the MNIST dataset (restricted to the digits 7 and 9, labeled by 1 and $-1$ respectively, $N = 2000$) and on the Higgs dataset (classes 'b' and 's', labeled by $-1$ and 1, $N = 1000$) with the RBF Kernel $K(x, x') = \exp(-\|x-x'\|_2^2/\ell)$ (see the Appendix for experiments with the Laplacian and $\ell_1$-norm kernels). KRR predictor risks, and KARE curves (shown as dashed lines, 5 samples) concentrate around their respective averages (solid lines).

## 1.2 Related Works

The theoretical analysis of the risk of KRR has seen tremendous developments in the recent years. In particular, a number of upper and lower bounds for kernel risk have been obtained [6, 31, 23] in various settings: notably, convergence rates (i.e. without control of the constant factors) are obtained in general settings. This allows one to abstract away a number of details about the kernels (e.g. the lengthscale), which don't influence the asymptotic rates. However, this does not give access to the risk at finite data size (crucial to pick e.g. the correct lengthscale or the NTK depth [17]).

A number of recent results have given precise descriptions of the risk for ridge regression [9, 20], for random features [24, 16], and in relation to neural networks [21, 5]. These results rely on the analysis of the asymptotic spectrum of general Wishart random matrices, in particular through the Stieltjes transform [30, 3]. The limiting Stieltjes transform can be recovered from the formula for the product of freely independent matrices [13]. To extend these asymptotic results to finite-size settings, we generalize and adapt the results of [16].

While these techniques have given simple formulae for the KRR predictor expectation, approximating its variance has remained more challenging. For this reason the description of the expected risk in [21] is stated as a conjecture. In [20] only the bias component of the risk is approximated. In [9] the expected risk is given only for random true functions (in a Bayesian setting) with a specific covariance. In [5], the expected risk follows from a heuristic spectral analysis combining a PDE approximation and replica tricks. In this paper, we approximate the variance of the predictor along the principal components, giving an approximation of the risk for any continuous true function.

The SCT is related to a number of objects from previous works, such as the effective dimension of [32, 6], the companion Stieltjes transform of [9, 20], and particularly the effective ridge of [16]. The SCT can actually be viewed as a direct translation to the KRR risk setting of [16].

## 1.3 Outline

In Section 2, we first introduce the Kernel Ridge Regression (KRR) predictor in functional space (Section 2.1) and formulate its train error and risk for random observations (Section 2.2).

The rest of the paper is then devoted to obtaining approximations for the KRR risk. In Section 3, the Signal Capture Threshold (SCT) is introduced and used to study the mean and variance of the KRR predictor (Sections 3.1 and 3.2). An approximation of the SCT in terms of the observed data is then given (Section 3.4). In Section 4, the expected risk and the expected empirical risk are approximated in terms of the SCT and its derivative w.r.t. the ridge $\lambda$. The SCT approximation of Section 3.4, together with the estimates of Section 4.1, leads to an approximation of the KRR risk by the Kernel Alignment Risk Estimator (KARE).

## 2 Setup

Given a compact $\Omega \subset \mathbb{R}^d$, let $\mathcal{C}$ denote the space of continuous $f : \Omega \to \mathbb{R}$, endowed with the supremum norm $\|f\|_\infty = \sup_{x \in \Omega} |f(x)|$. In the classical regression setting, we want to reconstruct a true function $f^* \in \mathcal{C}$ from its values on a training set $x_1, \ldots, x_N$, i.e. from the noisy labels $y^\epsilon = (f^*(x_1) + \epsilon e_1, \ldots, f^*(x_N) + \epsilon e_N)^T$ for some i.i.d. centered noise $e_1, \ldots, e_N$ of unit variance and noise level $\epsilon \geq 0$.

In this paper, the observed values (without noise) of the true function $f^*$ consist in observations $o_1, \ldots, o_N \in \mathcal{C}^*$, where $\mathcal{C}^*$ is the dual space, i.e. the space of bounded linear functionals $\mathcal{C} \to \mathbb{R}$. We thus represent the training set of $N$ observations $o_1, \ldots, o_N$ by the *sampling operator* $\mathcal{O} : \mathcal{C} \to \mathbb{R}^N$ which maps a function $f \in \mathcal{C}$ to the vector of observations $\mathcal{O}(f) = (o_1(f), \ldots, o_N(f))^T$.

The classical setting corresponds to the case where the observations are evaluations of $f^*$ at points $x_1, \ldots, x_N \in \Omega$, i.e. $o_i(f^*) = f^*(x_i)$ for $i = 1, \ldots, N$. In time series analysis (when $\Omega \subset \mathbb{R}$), the observations can be the averages $o_i(f^*) = \frac{1}{b_i - a_i} \int_{a_i}^{b_i} f^*(t) dt$ over time intervals $[a_i, b_i] \subset \mathbb{R}$.

### 2.1 Kernel Ridge Regression Predictor

The regression problem is now stated as follows: given noisy observations $y_i^\epsilon = o_i(f^*) + \epsilon e_i$ with i.i.d. centered noises $e_1, \ldots, e_N$ of unit variance, how can one reconstruct $f^*$?

**Definition 1.** *Consider a continuous positive kernel $K : \Omega \times \Omega \to \mathbb{R}$ and a ridge parameter $\lambda > 0$. The Kernel Ridge Regression (KRR) predictor with ridge $\lambda$ is the function $\hat{f}_\lambda^\epsilon : \Omega \to \mathbb{R}$*

$$\hat{f}_\lambda^\epsilon = \frac{1}{N} K \mathcal{O}^T (\frac{1}{N} \mathcal{O} K \mathcal{O}^T + \lambda I_N)^{-1} y^\epsilon$$

*where $\mathcal{O}^T : \mathbb{R}^N \to \mathcal{C}^*$ is the adjoint of $\mathcal{O}$ defined by $(\mathcal{O}^T y)(f) = y^T \mathcal{O}(f)$ and where we view $K$ as a map $\mathcal{C}^* \to \mathcal{C}$ with $(K\mu)(x) = \mu(K(x, \cdot))$.*

**Remark.** *The KRR predictor arises naturally in the following setup: assuming a (centered) Gaussian Bayesian prior on the true function with covariance operator $K$ and noise amplitude $\epsilon$, the expected posterior, for observed labels $y^\epsilon$ is given by $\hat{f}_\lambda^\epsilon$ for $\lambda = \epsilon^2$.*

We call the $N \times N$ matrix $G = \mathcal{O} K \mathcal{O}^T$ the *Gram matrix*: in the classical setting, when the observations are $o_i = \delta_{x_i}$ (with $\delta_x(f) = f(x)$), $G$ is the usual Gram matrix, i.e. $G_{ij} = K(x_i, x_j)$.

### 2.2 Training Error and Risk

We consider the least-squares error (MSE loss) of the KRR predictor, taking into account randomness of: (1) the test point, random observation $o$ to which is added a noise $\epsilon e$ (2) the training data, made of $N$ observations $o_i$ plus noises $\epsilon e_i \sim \nu$, where $o, o_1, \ldots, o_n \sim \pi$ and $e, e_1, \ldots, e_N$ are i.i.d. The expected risk of the KRR predictor is thus taken w.r.t. the test and training observations and their noises. Unless otherwise specified, the expectations are taken w.r.t. all these sources of randomness.

For (fixed) observations $o_1, \ldots, o_N$, the *empirical risk* or *training error* of the KRR predictor $\hat{f}_\lambda^\epsilon$ is

$$\hat{R}^\epsilon(\hat{f}_\lambda^\epsilon) = \frac{1}{N} \sum_{i=1}^N (o_i(\hat{f}_\lambda^\epsilon) - y_i^\epsilon)^2 = \frac{1}{N} \left\| \mathcal{O}(\hat{f}_\lambda^\epsilon) - y^\epsilon \right\|^2.$$

For a random observation $o$ sampled from $\pi$ and a noise $\epsilon e$ (where $e \sim \nu$ is centered of unit variance as before), the *risk* $R^\epsilon(\hat{f}_\lambda^\epsilon)$ of the KRR predictor $\hat{f}_\lambda^\epsilon$ is defined by

$$R^\epsilon(\hat{f}_\lambda^\epsilon) = \mathbb{E}_{o \sim \pi, e \sim \nu} \left[ (o(f^*) + \epsilon e - o(\hat{f}_\lambda^\epsilon))^2 \right].$$

Describing the observation variance by the bilinear form $\langle f, g \rangle_S = \mathbb{E}_{o \sim \pi} [o(f) o(g)]$ and the related semi-norm $\|f\|_S = \langle f, f \rangle_S^{1/2}$, the risk can be rewritten as $R^\epsilon(\hat{f}_\lambda^\epsilon) = \|\hat{f}_\lambda^\epsilon - f^*\|_S^2 + \epsilon^2$.

From now on, we will assume that $\langle \cdot, \cdot \rangle_S$ is a scalar product; note that in the classical setting, when $o$ is the evaluation of $f^*$ at a point $x \in \Omega$ with $x \sim \sigma$, the $S$-norm is given by $\|f\|_S^2 = \int_\Omega f(x)^2 \sigma(dx)$.

The following three operators $\mathcal{C} \to \mathcal{C}$ are central to our analysis:

**Definition 2.** *The KRR reconstruction operator $A_\lambda : \mathcal{C} \to \mathcal{C}$, the KRR Integral Operator $T_K : \mathcal{C} \to \mathcal{C}$, and its empirical version $T_K^N : \mathcal{C} \to \mathcal{C}$ are defined by*

$$A_\lambda = \frac{1}{N} K \mathcal{O}^T (\frac{1}{N} \mathcal{O} K \mathcal{O}^T + \lambda I_N)^{-1} \mathcal{O},$$

$$(T_K f)(x) = \langle f, K(x, \cdot) \rangle_S = \mathbb{E}_{o \sim \pi} \left[ o(f) o(K(x, \cdot)) \right],$$

$$(T_K^N f)(x) = \frac{1}{N} K \mathcal{O}^T \mathcal{O} f(x) = \frac{1}{N} \sum_{i=1}^{N} o_i(f) o_i(K(x, \cdot)).$$

Note that in the noiseless regime (i.e. when $\epsilon = 0$), we have $\hat{f}_\lambda^\epsilon \big|_{\epsilon=0} = A_\lambda f^*$. Also note that $A_\lambda$ and $T_K^N$ are random operators, as they depend on the random observations. The operator $T_K$ is the natural generalization to our framework of the integration operator $f \mapsto \int K(x, \cdot) f(x) \sigma(dx)$, which is defined with random observations $\delta_x$ with $x \sim \sigma$ in the classical setting.

The reconstruction and empirical integral operators are linked by $A_\lambda = T_K^N (T_K^N + \lambda I_\mathcal{C})^{-1}$, which follows from the identity $\left( \frac{1}{N} \mathcal{O} K \mathcal{O}^T + \lambda I_N \right)^{-1} \mathcal{O} = \mathcal{O} \left( \frac{1}{N} K \mathcal{O}^T \mathcal{O} + \lambda I_\mathcal{C} \right)^{-1}$. As $N \to \infty$, we have that $T_K^N \to T_K$, and it follows that

$$A_\lambda \to \tilde{A}_\lambda := T_K (T_K + \lambda I_\mathcal{C})^{-1}. \tag{1}$$

## 2.3 Eigendecomposition of the Kernel

We will assume that the kernel $K$ can be diagonalized by a countable family of eigenfunctions $(f^{(k)})_{k \in \mathbb{N}}$ in $\mathcal{C}$ with eigenvalues $(d_k)_{k \in \mathbb{N}}$, orthonormal with respect to the scalar product $\langle \cdot, \cdot \rangle_S$, such that we have (with uniform convergence):

$$K(x, x') = \sum_{k=1}^{\infty} d_k f^{(k)}(x) f^{(k)}(x').$$

The functions $f^{(k)}$ are also eigenfunctions of $T_K$: we have $T_K f^{(k)} = d_k f^{(k)}$. We will also assume that $\mathrm{Tr}\,[T_K] = \sum_{k=1}^{\infty} \langle f^{(k)}, T_K(f^{(k)}) \rangle_S = \sum_{k=1}^{\infty} d_k$ is finite. Note that in the classical setting $K$ can be diagonalized as above (by Mercer's theorem), and $\mathrm{Tr}\,[T_K] = \mathbb{E}_{x \sim \sigma}\,[K(x,x)]$ is finite. Computing the eigendecomposition of $T_K$ is difficult for general kernels and data distributions, but explicit formulas exist for special cases, such as for the RBF kernel and isotropic Gaussian inputs as described in Section 1.5 of the Appendix.

## 2.4 Gaussianity Assumption

As seen in Equation (1) above, $\tilde{A}_\lambda$ only depends on the second moment of $\pi$ (through $\langle \cdot, \cdot \rangle_S$), suggesting the following assumption, with which we will work in this paper:

**Assumption A.** *As far as one is concerned with the first two moments of the $A_\lambda$ operator, for large but finite $N$, we will assume that the observations $o_1, \ldots, o_N$ are centered Gaussian, i.e. that for any tuple of functions $(f_1, \ldots, f_N)$, the vector $(o_1(f_1), \ldots, o_N(f_N))$ is a mean zero Gaussian vector.*

Though our proofs use this assumption, the ideas in [21, 4] suggest a path to extend them beyond the Gaussian case, where our numerical experiments (see Figure 1) suggest that our results remain true. See Section 2.1 of the Appendix for a more detailed discussion.

## 3 Predictor Moments and Signal Capture Threshold

A central tool in our analysis of the KRR predictor $\hat{f}_\lambda^\epsilon$ is the Signal Capture Threshold (SCT):

**Definition 3.** *For $\lambda > 0$, the* Signal Capture Threshold *$\vartheta(\lambda) = \vartheta(\lambda, N, K, \pi)$ is the unique positive solution (see Section 2.2 in the Appendix) to the equation:*

$$\vartheta(\lambda) = \lambda + \frac{\vartheta(\lambda)}{N} \mathrm{Tr} \left[ T_K \left( T_K + \vartheta(\lambda) I_\mathcal{C} \right)^{-1} \right].$$

In this section, we use $\vartheta(\lambda)$ and the derivative $\partial_\lambda \vartheta(\lambda)$ for the estimation of the mean and variance of the KRR predictor $\hat{f}_\lambda^\epsilon$ upon which the Kernel Alignment Risk Estimator of Section 4 is based.

## 3.1 Mean predictor

The expected KRR predictor can be expressed in terms of the expected reconstruction operator $A_\lambda$

$$\mathbb{E}[\hat{f}_\lambda^\epsilon] = \mathbb{E}[\frac{1}{N}K\mathcal{O}^T(\frac{1}{N}\mathcal{O}K\mathcal{O}^T + \lambda I_N)^{-1}y^\epsilon] = \mathbb{E}[A_\lambda]f^*,$$

where we used the fact that $\mathbb{E}_{e_1,\dots,e_N}[y^\epsilon] = \mathcal{O}f^*$.

**Theorem 1** (Theorem 10 in the Appendix). *The expected reconstruction operator $\mathbb{E}[A_\lambda]$ is approximated by the operator $\tilde{A}_\vartheta = T_K(T_K + \vartheta(\lambda)I_{\mathcal{C}})^{-1}$ in the sense that for all $f, g \in \mathcal{C}$,*

$$\left| \left\langle f, \left(\mathbb{E}[A_\lambda] - \tilde{A}_\vartheta\right)g \right\rangle_S \right| \leq \left(\frac{1}{N} + \boldsymbol{P}_0(\frac{\text{Tr}[T_K]}{\lambda N})\right) \left| \left\langle f, \tilde{A}_\vartheta(I_{\mathcal{C}} - \tilde{A}_\vartheta)g \right\rangle_S \right|,$$

*for a polynomial $\boldsymbol{P}_0$ with nonnegative coefficients and $\boldsymbol{P}_0(0) = 0$.*

*Proof.* (Sketch; see the Appendix for details) First we show that $\mathbb{E}\left[\left\langle f^{(k)}, A_\lambda f^{(m)} \right\rangle_S\right] = 0$ whenever $m \neq k$, using the invariance of the observations' distribution $o_i$ w.r.t. reflection along a principal component $f^{(k)}$. This implies that $\mathbb{E}[A_\lambda]$ and $\tilde{A}_\vartheta$ both have the same eigenfunctions $(f^{(k)})_{k \geq 1}$. It thus only remains to show that the eigenvalues of both operators are close: $\mathbb{E}\left[\left\langle f^{(k)}, A_\lambda f^{(k)} \right\rangle_S\right] \approx \frac{d_k}{d_k + \vartheta}$.

The difficulty lies in computing the inverse of $B = \frac{1}{N}\mathcal{O}K\mathcal{O}^T + \lambda I_N$. We use the Sherman-Morrison formula to isolate the contribution along the $k$-th principal component $f^{(k)}$. Defining the kernel $K_{(k)}(x, y) = \sum_{\ell \neq k} d_\ell f^{(\ell)}(x) f^{(\ell)}(y)$ and the vector $\mathcal{O}_k = \mathcal{O}f^{(k)} \in \mathbb{R}^N$, we obtain

$$B^{-1} = B_{(k)}^{-1} - \frac{1}{N} \frac{d_k}{1 + d_k g_k} B_{(k)}^{-1} \mathcal{O}_k \mathcal{O}_k^T B_{(k)}^{-1}$$

for $B_{(k)} = \frac{1}{N}\mathcal{O}K_{(k)}\mathcal{O}^T + \lambda I_N$ and $g_k = \frac{1}{N}\mathcal{O}_k^T B_{(k)}^{-1}\mathcal{O}_k$. Using the above formula we obtain that

$$\left\langle f^{(k)}, A_\lambda f^{(k)} \right\rangle_S = \frac{1}{N} d_k \mathcal{O}_k^T B^{-1} \mathcal{O}_k = \frac{d_k g_k}{1 + d_k g_k}.$$

Since the vector $\mathcal{O}_k$ is independent of $B_{(k)}$ and has i.i.d. $\mathcal{N}(0, d_k)$ entries, $g_k$ concentrates around $\frac{1}{N}\text{Tr}B_{(k)}^{-1}$ which itself can be approximated by the Stieltjes transform $m(z = -\lambda) = \frac{1}{N}\text{Tr}B^{-1}$ (since $B_{(k)}$ is a rank-one deformation of $B$). Expanding the trivial equation $\frac{1}{N}\text{Tr}\left[BB^{-1}\right] = 1$, we obtain the relation

$$\frac{1}{N}\sum_{k=1}^\infty \frac{d_k g_k}{1 + d_k g_k} + \lambda m(-\lambda) = 1$$

which implies that both the $g_k$'s and the Stieltjes transform $m(-\lambda)$ concentrate around the unique solution $\tilde{m}$ to the equation $\frac{1}{N}\sum_{k=1}^\infty \frac{d_k \tilde{m}}{1 + d_k \tilde{m}} + \lambda \tilde{m} = 1$. The SCT is then defined as the reciprocal $\vartheta = 1/\tilde{m}$ and since $g_k \approx \tilde{m}$ we obtain that $\mathbb{E}\left[\left\langle f^{(k)}, A_\lambda f^{(k)} \right\rangle_S\right] = \mathbb{E}\left[\frac{d_k g_k}{1 + d_k g_k}\right] \approx \frac{d_k}{\vartheta + d_k}$ as needed. $\square$

This theorem gives the following motivation for the name SCT: if the true function $f^*$ is an eigenfunction of $T_K$, i.e. $T_K f^* = \delta f^*$, then $\tilde{A}_\vartheta f^* = \frac{\delta}{\vartheta(\lambda) + \delta} f^*$ and we get:

- if $\delta \gg \vartheta(\lambda)$, then $\frac{\delta}{\vartheta(\lambda) + \delta} \approx 1$ and $\mathbb{E}[A_\lambda]f^* \approx f^*$, i.e. the function is learned on average,

- if $\delta \ll \vartheta(\lambda)$, then $\frac{\delta}{\vartheta(\lambda) + \delta} \approx 0$ and $\mathbb{E}[A_\lambda]f^* \approx 0$, i.e. the function is not learned on average.

More generally, if we decompose a true function $f^*$ along the principal components (i.e. eigenfunctions) of $T_K$, the signal along the $k$-th principal component $f^{(k)}$ is captured whenever the corresponding eigenvalue $d_k \gg \vartheta(\lambda)$ and lost when $d_k \ll \vartheta(\lambda)$.

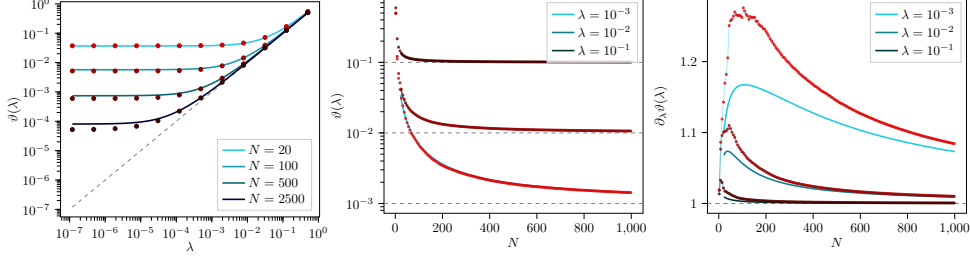

Figure 2: *Signal Capture Threshold and Derivative.* We consider the RBF Kernel on the standard $d$-dimensional Gaussian with $\ell = d = 20$. In blue lines, exact formulas for the SCT $\vartheta(\lambda)$ and $\partial_\lambda \vartheta(\lambda)$, computed using the explicit formula for the eigenvalues $d_k$ of the integral operator $T_K$ given in Section 1.5 of the Appendix; in red dots, their approximation with Proposition 5.

## 3.2   Variance of the predictor

We now estimate the variance of $\hat{f}_\lambda^\epsilon$ along each principal component in terms of the SCT $\vartheta(\lambda)$ and its derivative $\partial_\lambda \vartheta(\lambda)$. Along the eigenfunction $f^{(k)}$, the variance is estimated by $V_k$, where

$$V_k(f^*, \lambda, N, \epsilon) = \frac{\partial_\lambda \vartheta(\lambda)}{N} \left( \left\| (I_{\mathcal{C}} - \tilde{A}_\vartheta) f^* \right\|_S^2 + \epsilon^2 + \left\langle f^{(k)}, f^* \right\rangle_S^2 \frac{\vartheta^2(\lambda)}{(\vartheta(\lambda) + d_k)^2} \right) \frac{d_k^2}{(\vartheta(\lambda) + d_k)^2}.$$

**Theorem 2** (Theorem 15 in the Appendix). *There is a constant $C_1 > 0$ and a polynomial $P_1$ with nonnegative coefficients and with $P_1(0) = 0$ such that*

$$\left| \mathrm{Var}\left( \left\langle f^{(k)}, \hat{f}_\lambda^\epsilon \right\rangle_S \right) - V_k \right| \leq \left( \frac{C_1}{N} + P_1(\frac{\mathrm{Tr}[T_K]}{\lambda N^{\frac{1}{2}}}) \right) V_k.$$

As shown in Section 4.1, understanding the variance along the principal components (rather than the covariances between the principal components) is enough to describe the risk.

## 3.3   Behavior of the SCT

The behavior of the SCT can be controlled by the following (agnostic of the exact spectrum of $T_K$)
**Proposition 3** (Proposition 5 in the Appendix). *For any $\lambda > 0$, we have*

$$\lambda < \vartheta(\lambda, N) \leq \lambda + \frac{1}{N} \mathrm{Tr}[T_K], \qquad 1 \leq \partial_\lambda \vartheta(\lambda, N) \leq \frac{1}{\lambda} \vartheta(\lambda, N),$$

*moreover $\vartheta(\lambda, N)$ is decreasing as a function of $N$.*

**Remark.** *As $N \to \infty$, we have $\vartheta(\lambda, N)$ decreases down to $\lambda$ (see also Figure 2), in agreement with the fact that $A_\lambda \to \tilde{A}_\lambda$.*

As $\lambda \to 0$, the above upper bound for $\partial_\lambda \vartheta$ becomes useless. Still, assuming that the spectrum of $K$ has a sufficiently fast power-law decay, we get:
**Proposition 4** (Proposition 9 in the Appendix). *If $d_k = \Theta(k^{-\beta})$ for some $\beta > 1$, there exist $c_0, c_1, c_2 > 0$ such that for any $\lambda > 0$*

$$\lambda + c_0 N^{-\beta} \leq \vartheta(\lambda, N) \leq c_2 \lambda + c_1 N^{-\beta}, \qquad 1 \leq \partial_\lambda \vartheta(\lambda, N) \leq c_2.$$

## 3.4   Approximation of the SCT from the training data

The SCT $\vartheta$ and its derivative $\partial_\lambda \vartheta$ are functions of $\lambda, N$, and of the spectrum of $T_K$. In practice, the spectrum of $T_K$ is not known: for example, in the classical setting, one does not know the true data distribution $\sigma$. Fortunately, $\vartheta$ can be approximated by $1/m_G(-\lambda)$, where $m_G$ is the *Stieltjes Transform* of the Gram matrix, defined by $m_G(z) = \mathrm{Tr}\left[ (\frac{1}{N} G - z I_N)^{-1} \right]$. Namely, we get:
**Proposition 5** (Proposition 3 in the Appendix). *For any $\lambda > 0, s \in \mathbb{N}$, there is a $c_s > 0$ such that*

$$\mathbb{E}\left[ |1/\vartheta(\lambda) - m_G(-\lambda)|^{2s} \right] \leq \frac{c_s (\mathrm{Tr}[T_K])^{2s}}{\lambda^{4s} N^{3s}}.$$

**Remark.** *Likewise, we have $\partial_\lambda \vartheta \approx \left( \partial_z m_G(z) / m_G(z)^2 \right)|_{z=-\lambda}$, as shown in the Appendix.*

# 4 Risk Prediction with KARE

In this section, we show that the Expected Risk $\mathbb{E}[R^\epsilon(\hat{f}^\epsilon_\lambda)]$ can be approximated in terms of the training data by the Kernel Alignment Risk Estimator (KARE).

**Definition 4.** *The Kernel Alignment Risk Estimator (KARE) $\rho$ is defined by*

$$\rho(\lambda, N, y^\epsilon, G) = \frac{\frac{1}{N}(y^\epsilon)^T \left(\frac{1}{N}G + \lambda I_N\right)^{-2} y^\epsilon}{\left(\frac{1}{N}\mathrm{Tr}\left[\left(\frac{1}{N}G + \lambda I_N\right)^{-1}\right]\right)^2}.$$

In the following, using Theorems 1 and 2, we give an approximation for the expected risk and expected empirical risk in terms of the SCT and the true function $f^*$. This yields the important relation (2) in Section 4.2, which shows that the KARE can be used to efficiently approximate the kernel risk.

## 4.1 Expected Risk and Expected Empirical Risk

The expected risk is approximated, in terms of the SCT and the true function $f^*$, by

$$\tilde{R}^\epsilon(f^*, \lambda, N, K, \pi) = \partial_\lambda \vartheta(\lambda)(\|(I_\mathcal{C} - \tilde{A}_\vartheta)f^*\|_S^2 + \epsilon^2),$$

as shown by the following:

**Theorem 6** (Theorem 16 in the Appendix). *There exists a constant $C_2 > 0$ and a polynomial $P_2$ with nonnegative coefficients and with $P_2(0) = 0$, such that we have*

$$\left|\mathbb{E}[R^\epsilon(\hat{f}^\epsilon_\lambda)] - \tilde{R}^\epsilon(f^*, \lambda, N, K, \pi)\right| \leq \left(\frac{C_2}{N} + P_2(\frac{\mathrm{Tr}[T_K]}{\lambda N^{\frac{1}{2}}})\right) \tilde{R}^\epsilon(f^*, \lambda, N, K, \pi).$$

*Proof.* (Sketch; the full proof is given in the Appendix). From the bias-variance decomposition:

$$\mathbb{E}[R^\epsilon(\hat{f}^\epsilon_\lambda)] = R^\epsilon(\mathbb{E}[\hat{f}^\epsilon_\lambda]) + \sum_{k=1}^\infty \mathrm{Var}(\langle f^{(k)}, \hat{f}^\epsilon_\lambda\rangle_S).$$

By Theorem 1, and a small calculation, the bias is approximately $\|(I_\mathcal{C} - \tilde{A}_\vartheta)f^*\|_S^2 + \epsilon^2$. By Theorem 2, and a calculation, the variance is approximately $(\partial_\lambda \vartheta(\lambda) - 1)(\|(I_\mathcal{C} - \tilde{A}_\vartheta)f^*\|_S^2 + \epsilon^2)$. $\square$

The approximate expected risk $\tilde{R}^\epsilon(f^*, \lambda, N, K, \pi)$ is increasing in both $\vartheta$ and $\partial_\lambda \vartheta$. As $\lambda$ increases, the bias increases with $\vartheta$, while the variance decreases with $\partial_\lambda \vartheta$: this leads to the bias-variance tradeoff. On the other hand, as a function of $N$, $\vartheta$ is decreasing but $\partial_\lambda \vartheta$ is generally not monotone: this can lead to so-called multiple descent curves in the risk as a function of $N$ [19].

Note also that if we decompose the true function along the principal components $f^* = \sum_{k=1}^\infty b_k f^{(k)}$, the risk is approximated by $\tilde{R}^\epsilon(f^*) = \partial_\lambda \vartheta(\lambda)(\sum_{k=1}^\infty \frac{\vartheta(\lambda)^2}{(\vartheta(\lambda)+d_k)^2} b_k^2 + \epsilon^2)$.

**Remark.** *For a decaying ridge $\lambda = cN^{-\gamma}$ for $0 < \gamma < \frac{1}{2}$, as $N \to \infty$, by Proposition 3, we get $\vartheta(\lambda) \to 0$ and $\partial_\lambda \vartheta(\lambda) \to 1$: this implies that $\mathbb{E}[R^\epsilon(\hat{f}^\epsilon_\lambda)] \to \epsilon^2$. Hence the KRR can learn any continuous function $f^*$ as $N \to \infty$ (even if $f^*$ is not in the RKHS associated with $K$).*

**Remark.** *In a Bayesian setting, assuming that $f^*$ is random with zero mean and covariance kernel $\Sigma$, the optimal choices for the KRR predictor are $K = \Sigma$ and $\lambda = \epsilon^2/N$ (see Section 2.7 in the Appendix). When $K = \Sigma$ and $\lambda = \epsilon^2/N$, the formula of Theorem 6 simplifies (see Corollary 18 in the Appendix) to*

$$\mathbb{E}\left[R^\epsilon\left(\hat{f}^\epsilon_\lambda\right)\right] \approx N\vartheta\left(\frac{\epsilon^2}{N}, \Sigma\right).$$

The empirical risk (or train error) $\hat{R}^\epsilon(\hat{f}^\epsilon_\lambda) = \lambda^2(y^\epsilon)^T(\frac{1}{N}G + \lambda I_N)^{-2}y^\epsilon$ can be analyzed with the same theoretical tools. Its approximation in terms of the SCT is given as follows:

**Theorem 7** (Theorem 17 in the Appendix). *There exists a constant $C_3 > 0$ and a polynomial $P_3$ with nonnegative coefficients and with $P_3(0) = 0$ such that we have*

$$\left|\mathbb{E}[\hat{R}^\epsilon(\hat{f}^\epsilon_\lambda)] - \frac{\lambda^2}{\vartheta(\lambda)^2}\tilde{R}^\epsilon(\hat{f}^\epsilon_\lambda, \lambda, N, K, \pi)\right| \leq \left(\frac{1}{N} + P_3(\frac{\mathrm{Tr}[T_K]}{\lambda N})\right) \tilde{R}^\epsilon(f^*, \lambda, N, K, \pi).$$

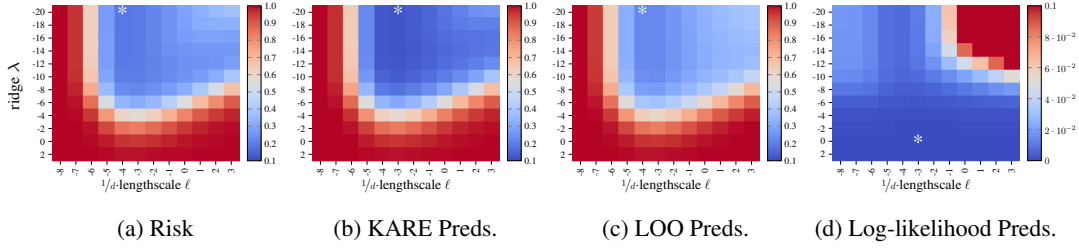

| (a) Risk | (b) KARE Preds. | (c) LOO Preds. | (d) Log-likelihood Preds. |

Figure 3: *Comparision of risk predictors.* We calculate the risk (i.e. test error) of $\hat{f}_\lambda^\epsilon$ on MNIST with the RBF Kernel for various values of $\ell$ and $\lambda$ on $N = 200$ data points (same setup as Fig. 1). We mark the minimum MSE achieved with a star. We display the predictions of KARE and leave-one-out (LOO); both find the hyper-parameters minimizing the risk. We also show the (normalized) log-likehood estimator and observe that it favors large $\lambda$ values. Axes are $\log_2$ scale.

## 4.2 KARE: Kernel Alignment Risk Estimator

While the above approximations (Theorems 6 and 7) for the expected risk and empirical risk depend on $f^*$, their combination yields the following relation, which is surprisingly independent of $f^*$:

$$\mathbb{E}\left[R^\epsilon\left(\hat{f}_\lambda^\epsilon\right)\right] \approx \frac{\vartheta^2}{\lambda^2}\mathbb{E}\left[\hat{R}^\epsilon\left(\hat{f}_\lambda^\epsilon\right)\right]. \qquad (2)$$

Since $\vartheta$ can be approximated from the training set (see Proposition 5), so can the expected risk. Assuming that the risk and empirical risk concentrate around their expectations, we get the KARE:

$$R^\epsilon\left(\hat{f}_\lambda^\epsilon\right) \approx \rho(\lambda, N, y^\epsilon, G) = \frac{\frac{1}{N}(y^\epsilon)^T\left(\frac{1}{N}G + \lambda I_N\right)^{-2}y^\epsilon}{\left(\frac{1}{N}\mathrm{Tr}\left[\left(\frac{1}{N}G + \lambda I_N\right)^{-1}\right]\right)^2}.$$

**Remark.** *As shown in the Appendix, estimating the risk of the expected predictor $\mathbb{E}[\hat{f}_\lambda^\epsilon]$ yields:*

$$R^\epsilon(\mathbb{E}[\hat{f}_\lambda^\epsilon]) \approx \varrho(\lambda, N, y^\epsilon, G) = \frac{(y^\epsilon)^T(\frac{1}{N}G + \lambda I_N)^{-2}y^\epsilon}{\mathrm{Tr}[(\frac{1}{N}G + \lambda I_N)^{-2}]}.$$

*Note that both $\rho$ and $\varrho$ are invariant (as is the risk) under the simultaneous rescaling $K, \lambda \rightsquigarrow \alpha K, \alpha\lambda$.*

The KARE can be used to optimize the risk over the space of kernels, for instance to choose the ridge and length-scale. The most popular kernel selection techniques are (see Figure 3):

- Leave-one-out: accurate estimator of the risk on a test set, it has a closed-form formula similar yet different from the KARE [26].
- Kernel likelihood (Chapter 5 of [25]): efficient to optimize and takes into account the ridge, but not a risk estimator; unlike the risk, not invariant under the simultaneous rescaling $K, \lambda \rightsquigarrow \alpha K, \alpha\lambda$.
- Classical kernel alignment [8]: very efficient to optimize and scale invariant, but not a risk estimator, not sensitive to small eigenvalues and inadequate to select hyperparameters such as the ridge.

The KARE has the following three desirable properties:

- it can be computed efficiently on the training data, and optimized over the space of kernels;
- like the risk, it is invariant under the simultaneous rescaling $K, \lambda \rightsquigarrow \alpha K, \alpha\lambda$;
- it is sensitive to the small Gram matrix eigenvalues and to the ridge $\lambda$.

## 5 Conclusion

In this paper, we introduce new techniques to study the Kernel Ridge Regression (KRR) predictor and its risk. We obtain new precise estimates for the test and train error in terms of a new object, the Signal Capture Threshold (SCT), which identifies the components of a true function that are being learned by the KRR: our estimates reveal a remarkable relation, which leads one to the Kernel Alignment Risk Estimator (KARE). The KARE is a new efficient way to estimate the risk of a kernel predictor based on the training data only. Numerically, we observe that the KARE gives a very accurate prediction of the risk for Higgs and MNIST datasets for a variety of classical kernels.

## Broader Impact

This work is fundamental and may be used in any research area using Kernel methods, possibly leading to indirect social impacts. However, we do not predict any direct social impact.

## Acknowledgements

The authors wish to thank A. Montanari and M. Wyart for useful discussions. This work is partly supported by the ERC SG CONSTAMIS. C. Hongler acknowledges support from the Blavatnik Family Foundation, the Latsis Foundation, and the the NCCR Swissmap.

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
