[Supplementary Material]

# Supplementary Material for Kernel Alignment Risk Estimator: Risk Prediction from Training Data

**Arthur Jacot**
Ecole Polytechnique Fédérale de Lausanne
`arthur.jacot@epfl.ch`

**Berfin Şimşek**
Ecole Polytechnique Fédérale de Lausanne
`berfin.simsek@epfl.ch`

**Francesco Spadaro**
Ecole Polytechnique Fédérale de Lausanne
`francesco.spadaro@epfl.ch`

**Clément Hongler**
Ecole Polytechnique Fédérale de Lausanne
`clement.hongler@epfl.ch`

**Franck Gabriel**
Ecole Polytechnique Fédérale de Lausanne
`franck.gabriel@epfl.ch`

We organize the Supplementary Material (Supp. Mat.) as follows:

1. In Section 1, we present the details for the numerical results presented in the main text (and in the Supp. Mat.) and we present additional experiments and some discussions.
2. In Section 2, we present the proofs of the mathematical results presented in the main text.

## 1 Numerical Results

### 1.1 Empirical Methods

**For the MNIST dataset.** We sample $N$ images of digits 7 and 9 from the MNIST training dataset (image size $d = 24 \times 24$, edge pixels cropped, all pixels rescaled down to $[0, 1]$ and recentered around the mean value) and label each of them with $+1$ and $-1$ labels. We perform KRR with various ridge $\lambda$ on this dataset with the selected kernel $k$ times and calculate the MSE training error, risk, and the KARE for every trial ($k = 10$ for small $N$ and $k = 5$ for $N = 2000$). The risk is approximated using other $N_2 = 1000$ random samples of the MNIST training data.

**For the Higgs Dataset.** We randomly choose $N$ samples among those that do not have any missing features marked with $-999$ from the Higgs training dataset. The samples have $d = 31$ features, and we normalize each feature column down to $[0, 1]$ by dividing by the maximum absolute value observed among the selected samples. We replace the categorical labels 's' and 'b' with regression values $+1$ and $-1$ respectively and perform KRR with various ridge $\lambda$. We repeat this procedure $k$ times, which corresponds to sampling $k$ different training datasets of $N = 1000$ samples to perform kernel regression, and calculate the MSE training error, the risk, and the KARE for every trial ($k = 10$ for small $N$ and $k = 5$ for $N = 1000$). The risk is approximated using other $N_2 = 1000$ random samples of the Higgs training data.

## 1.2 KARE predicts risk for various Kernels

(a) MNIST, $\ell = d$     (b) MNIST, $\lambda = 10^{-6}$     (c) Higgs, $\ell = d$     (d) Higgs, $\lambda = 10^{-5}$

(e) MNIST, $\ell = d$     (f) MNIST, $\lambda = 10^{-5}$     (g) Higgs, $\ell = d$     (h) Higgs, $\lambda = 10^{-5}$

Figure 1: Comparison between the KRR risk and the KARE for various choices of normalized lengthscale $\ell/d$ and ridge $\lambda$ on the MNIST dataset (restricted to the digits 7 and 9, labeled by 1 and $-1$ respectively, $N = 2000$) and on the Higgs dataset (classes 'b' and 's', labeled by $-1$ and 1, $N = 1000$). We present the results for the Laplacian Kernel $K(x, x') = \exp\left(-\|x-x'\|_2/\ell\right)$ (top row) and the $\ell_1$-norm Kernel $K(x, x') = \exp\left(-\|x-x'\|_1/\ell\right)$ (bottom row). KRR predictor risks, and KARE curves (shown as dashed lines, 5 samples) concentrate around their respective averages (solid lines).

## 1.3 KRR predictor in function space

Figure 2: *KRR predictor in function space for various $N$ and $\lambda$ for the RBF Kernel $K$ with $\ell = d = 1$. Observations $o = \delta_x$ are sampled with uniform distribution on $x \sim U[-1, 3]$ (shown in blue) $\hat{f}_\lambda^\epsilon$ is calculated 500 times for different realizations of the training data (10 example predictors are shown in dashed lines), its mean and $\pm 2$ standard deviation are shown in red. The true function $f^*(x) = x^2 + 2\cos(4x)$ is shown in black. Second row. Observations $o = \delta_x$ are sampled with uniform distribution $x \sim U[0, 1.5]$ (shown in blue) and $\hat{f}_\lambda^\epsilon$ is calculated 100 times. The true function $f^*(x) = x^2$ is shown in black.*

## 1.4 KARE predicts risk in average for small $N$

Figure 3: *The estimation predicts the risk in average for small $N = \{100, 500\}$ on MNIST data.* In the top row, we used the RBF Kernel $K(x, z) = \exp(-\|x-z\|_2^2/\ell)$, in the second row, we used the Laplacian Kernel $K(x, z) = \exp(-\|x-z\|_2/\ell)$, and in the bottom row, we used the $\ell_1$-norm Kernel $K(x, z) = \exp(-\|x-z\|_1/\ell)$ for various choices of $\ell$ and $\lambda$. The optimal predictor is calculated using $N$ random samples ($N = 100$ for the plots on the left and $N = 500$ for the ones on the right) from the training data 10 times (dashed curves) and their average is plotted in the solid curves.

## 1.5 SCT and its behavior

In general, it is hard to compute the spectrum $(d_k)_{k \in \mathbb{N}}$ of $T_K$ even when one has the knowledge of the true data distribution. Luckily, following an adaptation from [9, 6], we can obtain an explicit formula for $d_k$ for centered $d$-dimensional Gaussian distribution with covariance matrix $\sigma^2 I_d$, and RBF Kernel $K(x, x') = \exp(-\|x - x'\|^2 / \ell)$. The formula for the distinct eigenvalues $\lambda_k$ is

$$\lambda_k = \left( \sqrt{\frac{1}{2A\sigma^2}} \right)^d B^k, \tag{1}$$

where $A = \frac{1}{4\sigma^2} + \frac{1}{\ell} + c$, $B = \frac{1}{A\ell}$ with $c = \frac{1}{2\sigma} \sqrt{\frac{1}{4\sigma^2} + \frac{2}{\ell}}$. Each $\lambda_k$ has multiplicity

$$n_d(k) = \sum_{j=1}^{k} \binom{d}{j} \binom{k-1}{j-1} \tag{2}$$

for $k \geq 1$. In particular, we have $n_d(0) = 1, n_d(1) = \binom{d}{1}, n_d(2) = \binom{d}{2} + d, \dots$. In general, $n_d(k)$ is the number of ways to partition $k$ into $d$ non-negative integers.

The true SCT is therefore approximated solving the following equation numerically

$$\vartheta = \lambda + \frac{\vartheta}{N} \sum_{i=1}^{k} \frac{n_d(k)\lambda_k}{\lambda_k + \vartheta}. \tag{3}$$

Figure 4: *Behavior of SCT as a function of $\lambda$ and $N$.* True SCT is calculated on the $k = 50$ biggest distinct eigenvalues using the formula 3 for $\ell = d = 5$ and $\sigma = 1$. Red dots are the approximations obtained using Proposition 5 in the main text, i.e. $\vartheta \approx 1/\text{Tr}[(\frac{1}{N}K(X, X) - \lambda I)^{-1}]$.

Note that in the Figure 2 in the main text, we limit the approximation to $k = 10$ for $d = 20$ because the multiplicity $n_d(k)$ grows polynomially with $d^k$.

## 2 Proofs

**Preliminary: Big-P notation**

Throughout our proofs, we will frequently rely on a polynomial analogue of the big-O notation, which we call big-P:

**Definition 1.** *For two functions $f$ and $g$ (of one or several variables, defined on an arbitrary common domain $\mathcal{D}$), we write $f = \mathcal{P}(g)$ if $g$ is nonnegative over $\mathcal{D}$ and there exists a polynomial $\mathbf{P}$ with nonnegative coefficients and $\mathbf{P}(0) = 0$ such that $|f| \leq \mathbf{P}(g)$ over $\mathcal{D}$.*

Note that the big-O notation corresponds to the case when the polynomial $\mathbf{P}$ is of degree at most one.

### 2.1 Gaussianity Assumption

For the sake of simplicity, our proof are made under the assumptions that the observations are Gaussian. However we conjecture that as long as the higher moments are bounded/small enough, the general non-Gaussian case can be reduced to the Gaussian case, up to a small error (as it is common in random matrix theory).

There are two special cases where a weaker Gaussianity property applies, i.e. that the $o_i K \in \mathcal{C}$ are Gaussian processes, and is enough for our proofs as everytime $\mathcal{O}$ appears in the formulas, it is composed with $K$. These two special cases are:

1. For the linear kernel $K(x, y) = x^T y$, $o_i K$ is the linear function $x \mapsto x_i^T x$, which is Gaussian whenever the inputs $x_i$ are sampled from a Gaussian distribution.
2. As noted in [5], this linear case can be generalized to a broader (non-linear) family kernel $K$ in the large input space limit: as shown in [4], in the large width limit the kernel Gram matrix $G$ for such kernel $K$ can be approximated by the Gram matrix of a linear kernel (up to scaling) hence leading back to the previous point.

Let us observe that all the quantities we study (the predictor, the risk and empirical risk) stay the same if any observation $o_i$ is replaced by $-o_i$. Hence a posteriori, by a symmetrization trick we may remove the assumption that the observations are centered (as in general they are not).

### 2.2 Objects of Interest and general strategy

The central object of our analysis is the $N \times N$ Gram matrix $\mathcal{O}K\mathcal{O}^T$, in particular the related Stieltjes transform:

$$m(z) = \frac{1}{N}\mathrm{Tr}\left[B(z)^{-1}\right]$$

where $B(z) = \frac{1}{N}\mathcal{O}K\mathcal{O}^T - zI_N$ and $z \in \mathbb{C} \setminus \mathbb{R}_+$.

From now on, we consider only $z \in \mathbb{H}_{<0} = \{z : \Re(z) < 0\}$. Note that $m(z) = \frac{1}{N}\sum_\ell \frac{1}{\lambda_\ell - z}$ where $\lambda_\ell \geq 0$ are the real eigenvalues of $\frac{1}{N}\mathcal{O}K\mathcal{O}^T$, hence $m(z)$ lies in the cone $\Gamma$ spanned by 1 and $-1/z$, i.e. $\Gamma = \{a - b\frac{1}{z} | a, b \geq 0\}$. We will first show that for $z \in \mathbb{H}_{<0}$, the Stieltjes transform concentrates around the unique solution $\tilde{m}(z)$ to the equation

$$\tilde{m}(z) = -\frac{1}{z}\left(1 - \frac{1}{N}\mathrm{Tr}\left[\tilde{m}(z)T_K\left(I_\mathcal{C} + \tilde{m}(z)T_K\right)^{-1}\right]\right), \tag{4}$$

and then show that the linear map

$$A(z) = \frac{1}{N}K\mathcal{O}^T\left(\frac{1}{N}\mathcal{O}K\mathcal{O}^T - zI_N\right)^{-1}\mathcal{O} = \frac{1}{N}K\mathcal{O}^T B(z)^{-1}\mathcal{O}$$

concentrates around the map $\tilde{A}_{\vartheta(-z)} = T_K\left(T_K + \vartheta(-z)I_\mathcal{C}\right)^{-1}$, where $(T_K f)(x) = \mathbb{E}_{o\sim\pi}\left[o(K(x, \cdot))o(f)\right] = \langle K(x, \cdot), f\rangle_S$ and $\vartheta(-z) = \frac{1}{\tilde{m}(z)}$ is the Signal Capture Threshold. From Equation (4), the SCT can be also defined as the solution to the equation

$$\vartheta(-z) = -z + \frac{\vartheta(-z)}{N}\mathrm{Tr}\left[T_K\left(T_K + \vartheta(-z)I_\mathcal{C}\right)^{-1}\right]. \tag{5}$$

From now on, we denote $\vartheta(-z)$ by $\vartheta$. Note that here, in the Appendix, we use the resolvent notation: in particular the KRR reconstruction operator $A_\lambda$ is equal to $A(-\lambda)$.

### 2.2.1 Spectral decomposition and generalized matrix representation

Throughout this paper it is assumed that there exists an orthonormal basis of continuous functions $\left(f^{(k)}\right)_k$ for the scalar product $\langle \cdot, \cdot \rangle_S$ such that $K = \sum_{k \in \mathbb{N}} d_k f^{(k)} \otimes f^{(k)}$ and $\sum_{k \in \mathbb{N}} d_k < \infty$. For a linear map $M : \mathcal{C} \to \mathcal{C}$, we define the $(k, \ell)-$entry of $M$ as:

$$M_{k\ell} = \left\langle f^{(k)}, M f^{(\ell)} \right\rangle_S.$$

With this notation, the trace of a linear map $M$ becomes $\mathrm{Tr}\,(M) = \sum_{k \in \mathbb{N}} M_{kk}$.

Similarly, using the canonical basis $(b_i)_{i=1,\dots,N}$ of $\mathbb{R}^N$, we define the entries of $\mathcal{O} : \mathcal{C} \to \mathbb{R}^N$ and $\mathcal{O}^T : \mathbb{R}^N \to \mathcal{C}^*$ by

$$\mathcal{O}_{ik} = b_i \cdot \mathcal{O} f^{(k)} = o_i(f^{(k)}), \quad \mathcal{O}_{ik}^T = \mathcal{O}^T b_k(f^{(i)}) = b_k \cdot \mathcal{O} f^{(i)} = o_k(f^{(i)}).$$

Since the observations $o_i$ are i.i.d. Gaussians with zero mean and covariance $\mathbb{E}\,[o_i(f)o_i(g)] = \langle f, g \rangle_S$ and since $\left(f^{(k)}\right)_k$ is an orthonormal basis for the scalar product $\langle \cdot, \cdot \rangle_S$, the entries $\mathcal{O}_{ik}$ are i.i.d standard Gaussians.

Using the spectral decomposition of $K$, the entries of $\mathcal{O}K\mathcal{O}^T$ are given by:

$$\left(\mathcal{O}K\mathcal{O}^T\right)_{i,j} = \sum_\ell d_\ell o_i(f^{(\ell)}) o_j(f^{(\ell)}),$$

where the sum converges absolutely (thanks to the trace assumption on $K$) and the entries of $A$ are then given by:

$$A_{k\ell}(z) = \frac{d_k}{N} \left(\mathcal{O}_{\cdot k}\right)^T \left(\frac{1}{N}\mathcal{O}K\mathcal{O}^T - zI_N\right)^{-1} \mathcal{O}_{\cdot \ell} \tag{6}$$

where $\mathcal{O}_{\cdot k} = \left(o_i(f^{(k)})\right)_{i=1,\dots,N}$.

### 2.2.2 Shermann-Morrison Formula

The Shermann-Morrison formula allows one to study how the inverse of a matrix is modified by a rank one perturbation of the matrix. The matrix $\mathcal{O}K\mathcal{O}^T$ can be seen as a perturbation of $\mathcal{O}K_{(k)}\mathcal{O}^T$ by the rank one matrix $d_k \mathcal{O}_{\cdot k}\mathcal{O}_{\cdot k}^T$, where $K_{(k)} := \sum_{\ell \neq k} d_\ell f^{(\ell)} \otimes f^{(\ell)}$. By doing so, one isolates the contribution of the $k$-th eigenvalue of $K$. Thus, one can compute $B(z)^{-1} = \left(\frac{1}{N}\mathcal{O}K\mathcal{O}^T - zI_N\right)^{-1}$ using the Shermann-Morrison formula:

$$B(z)^{-1} = B_{(k)}(z)^{-1} - \frac{1}{N}\frac{d_k}{1 + d_k g_k(z)} B_{(k)}(z)^{-1} \mathcal{O}_{\cdot k}\mathcal{O}_{\cdot k}^T B_{(k)}(z)^{-1} \tag{7}$$

where $B_{(k)}(z) = \frac{1}{N}\mathcal{O}K_{(k)}\mathcal{O}^T - zI_N$ and $g_k(z) = \frac{1}{N}\mathcal{O}_{\cdot k}^T B_{(k)}(z)^{-1}\mathcal{O}_{\cdot k}$. A crucial property is that, since $o_i\left(f^{(k)}\right)$ does not appear anymore in $\mathcal{O}K_{(k)}\mathcal{O}^T$ and, since for any $\ell \neq k$ and any $i, j$, we have that $o_i\left(f^{(k)}\right)$ is independent from $o_j(f^{(\ell)})$, we obtain that the matrix $B_{(k)}(z)^{-1}$ is independent of $\mathcal{O}_{\cdot k}$.

**Remark.** *Using the diagonalization of $B_{(k)}(z)^{-1} = U^T \mathrm{diag}\left(\frac{1}{\nu_\ell - z}\right) U$ with $U$ orthogonal and $\nu_\ell \geq 0$, we have that $g_k(z) = \frac{1}{N}\sum_\ell \frac{\left[\sum_i U_{\ell,i} o_i(f^{(k)})\right]^2}{\nu_\ell - z}$ lies in the cone spanned by $1$ and $-1/z$, in particular, $\Re(g_k) \geq 0$ on $\mathbb{H}_{<0}$.*

As a result of Equations (6) and (7), the diagonal entries of the operator $A(z) = \frac{1}{N}K\mathcal{O}^T B(z)^{-1}\mathcal{O}$ are equal to

$$A_{kk}(z) = \frac{d_k g_k(z)}{1 + d_k g_k(z)}. \tag{8}$$

**Remark.** *For any $z \in \mathbb{H}_{<0}$, the sum $\sum_k |A_{kk}(z)|$ is almost surely finite. Indeed, notice that*

$$\left|\frac{d_k g_k(z)}{1 + d_k g_k(z)}\right| \leq |d_k g_k(z)| \leq \frac{1}{N} d_k \left\|\mathcal{O}_{\cdot k}\right\|^2 \left\|B_{(k)}(z)^{-1}\right\|_{\mathrm{op}}.$$

*For any $z \in \mathbb{H}_{<0}$, $\left\| B_{(k)}(z)^{-1} \right\|_{\mathrm{op}} \leq \frac{1}{|z|}$ and thus*

$$\left| \frac{d_k g_k(z)}{1 + d_k g_k(z)} \right| \leq \frac{1}{N |z|} d_k \left\| \mathcal{O}_{\cdot k} \right\|^2 .$$

*Since $\mathbb{E}\left[ \sum_k d_k \left\| \mathcal{O}_{\cdot k} \right\|^2 \right] = N \mathrm{Tr}[T_K] < \infty$, we have that $\sum_k |A_{kk}(z)|$ is almost surely finite.*

*The operator $A$ is therefore a.s. trace-class and $\mathrm{Tr}(A) = \sum_k \frac{d_k g_k(z)}{1 + d_k g_k(z)}$, where the sum is absolutely convergent.*

Another important observation is that the Stieltjes transform $m(z)$ and the $g_k(z)$ are closely related.

**Lemma 1.** *For any $z \in \mathbb{H}_{<0}$, a.s. we have*

$$m(z) = -\frac{1}{z} \left( 1 - \frac{1}{N} \sum_{k=1}^{\infty} \frac{d_k g_k(z)}{1 + d_k g_k(z)} \right). \tag{9}$$

*Proof.* Indeed, using the trivial relation $\mathrm{Tr}\left[ B(z) B(z)^{-1} \right] = N$, expanding $B(z)$, we obtain $\mathrm{Tr}\left[ \frac{1}{N} \mathcal{O} K \mathcal{O}^T B(z)^{-1} \right] - z \mathrm{Tr}\left[ B(z)^{-1} \right] = N$. Since $\mathcal{O}$ is an operator from $\mathcal{C}$ to $\mathbb{R}^N$, which is a finite dimensional space, we can apply the cyclic property of the trace and obtain $\mathrm{Tr}\left[ \frac{1}{N} \mathcal{O} K \mathcal{O}^T B(z)^{-1} \right] = \mathrm{Tr}\left[ A(z) \right]$. Thus,

$$\mathrm{Tr}\left[ A(z) \right] - z \mathrm{Tr}\left[ B(z)^{-1} \right] = N.$$

Dividing both sides by $N$ and using Equation (8), we obtain

$$1 = \frac{1}{N} \sum_{k=1}^{\infty} \frac{d_k g_k(z)}{1 + d_k g_k(z)} - z m(z),$$

hence the result. $\qquad\square$

### 2.3 Concentration of the Stieltjes Transform

We will now show that $g_k(z) = \frac{1}{N} \mathcal{O}_{\cdot k}^T B_{(k)}(z)^{-1} \mathcal{O}_{\cdot k}$ is close to $\frac{1}{N} \mathrm{Tr}\left( B_{(k)}(z)^{-1} \right)$, as suggested by the fact that by Wick's formula $\mathbb{E}[g_k] = \frac{1}{N} \mathrm{Tr}\left( \mathbb{E}\left[ B_{(k)}(z)^{-1} \right] \right)$. Since $B(z)$ is obtained using a rank one permutation of $B_{(k)}(z)$, $\frac{1}{N} \mathrm{Tr}\left( B_{(k)}(z)^{-1} \right)$ is close to the Stieltjes transform $m$. As a result, all the $g_k$'s are close to the Stieltjes transform $m$: it is natural to think that for $z \in \mathbb{H}_{<0}$, both $g_k(z)$'s and $m(z)$ should concentrate around the unique solution $\tilde{m}(z)$ in the cone spanned by $1$ and $-1/z$ of the equation

$$\tilde{m}(z) = -\frac{1}{z} \left( 1 - \frac{1}{N} \sum_{k=1}^{\infty} \frac{d_k \tilde{m}(z)}{1 + d_k \tilde{m}(z)} \right). \tag{10}$$

**Remark.** *The existence and the uniqueness of the solution in the cone spanned by $1$ and $-1/z$ of the equation can be argued as follows. If in Equation (10) we truncate the series and consider the sum of the first $M$ terms, one can show that there exists a unique fixed point $\tilde{m}_M(z)$ in the region $R$ given by intersection between the cone spanned by $1$ and $-1/z$ and the cone spanned by $z$ and $1/z$ translated by $+1$ and multiplied by $-1/z$ (see Lemma C.6 in the Supplementary Material of [8]). Since $R$ is a compact region, we can extract a converging subsequence that solves Equation (10), the limit of which can be showed to be unique, again using the same arguments of Lemma C.6 in the Supplementary Material of [8].*

From now on we omit the $z$ dependence and we set $m = m(z)$, $\tilde{m} = \tilde{m}(z)$ and $g_k(z) = g_k$.

#### 2.3.1 Concentration bounds

Using Equation 9 and the definition of the fixed point $\tilde{m}$ (Equation 10), we obtain the following formula for the difference between the Stieltjes transform $m$ and $\tilde{m}$:

$$\begin{aligned}
\tilde{m} - m &= \frac{1}{z} \frac{1}{N} \sum_{k=1}^{\infty} \frac{d_k \left( \tilde{m} - g_k \right)}{(1 + d_k \tilde{m})(1 + d_k g_k)} \\
&= \frac{\tilde{m} - m}{z} \frac{1}{N} \sum_{k=1}^{\infty} \frac{d_k}{(1 + d_k \tilde{m})(1 + d_k g_k)} + \frac{1}{z} \frac{1}{N} \sum_{k=1}^{\infty} \frac{d_k \left( m - g_k \right)}{(1 + d_k \tilde{m})(1 + d_k g_k)},
\end{aligned}$$

where the well-posedness of the two infinite sums of the r.h.s is granted by the fact that:

1. $\left| \frac{d_k}{(1+d_k\tilde{m})(1+d_kg_k)} \right| \leq d_k$ since $\Re(\tilde{m}), \Re(g_k)$ are positive, thus the first sum is absolutely convergent,

2. being the difference of two absolutely convergent series, the second sum is also absolutely convergent.

As a consequence, the difference $\tilde{m} - m$ can be expressed as

$$\tilde{m} - m = \frac{\frac{1}{N}\sum_{k=1}^{\infty} \frac{d_k(m-g_k)}{(1+d_k\tilde{m})(1+d_kg_k)}}{z - \frac{1}{N}\sum_{k=1}^{\infty} \frac{d_k}{(1+d_k\tilde{m})(1+d_kg_k)}}, \tag{11}$$

which allows us to show the concentration of $m$ around $\tilde{m}$ from the concentration of $g_k$ around $m$.

Regarding the concentration of the $g_k$'s around $m$, we have the following result:

**Lemma 2.** *For any $N, s \in \mathbb{N}$ and any $z \in \mathbb{H}_{<0}$, we have*

$$\mathbb{E}\left[ |m - g_k|^{2s} \right] \leq \frac{c_s}{|z|^{2s}N^s},$$

$$\mathbb{E}\left[ |m - m_{(k)}|^{2s} \right] \leq \frac{1}{|z|^{2s}N^{2s}}.$$

*where $c_s$ only depends on $s$.*

*Proof.* The second inequality will be proven while proving the first one. Let $m_{(k)} = \frac{1}{N}\mathrm{Tr}\left[ B_{(k)}^{-1} \right]$ where $B_{(k)}$ was defined in Section 2.2.2. By convexity:

$$\mathbb{E}\left[ |m - g_k|^{2s} \right] \leq 2^{2s-1}\mathbb{E}\left[ |m - m_{(k)}|^{2s} \right] + 2^{2s-1}\mathbb{E}\left[ |m_{(k)} - g_k|^{2s} \right]. \tag{12}$$

**Bound on $\mathbb{E}[|m - m_{(k)}|^{2s}]$:** We obtain the bound on the expectation by showing that a deterministic bound holds for the random variable $|m - m_{(k)}|^{2s}$. Using the Sherman-Morrison formula (Equation (7)), and using the cyclic property of the trace,

$$m = m_{(k)} - \frac{1}{N}\frac{d_kg_k'}{1 + d_kg_k}$$

since the derivative $g_k'(z)$ of $g_k(z)$ is equal to $\frac{1}{N}\mathcal{O}_{\cdot k}^T B(z)^{-2}\mathcal{O}_{\cdot k}$. As a result, we obtain $|m - m_{(k)}|^{2s} = \frac{1}{N^{2s}}\frac{d_k^{2s}|g_k'|^{2s}}{|1+d_kg_k|^{2s}}$. Using the fact that $|1 + d_kg_k| \geq |d_kg_k|$ since $\Re(g_k) \geq 0$,

$$\left| m - m_{(k)} \right|^{2s} \leq \frac{1}{N^{2s}}\frac{|g_k'|^{2s}}{|g_k|^{2s}}.$$

Notice now that

$$\left| \frac{g_k'}{g_k} \right| = \left| \frac{\mathcal{O}_{\cdot k}^T B_{(k)}(z)^{-2}\mathcal{O}_{\cdot k}}{\mathcal{O}_{\cdot k}^T B_{(k)}(z)^{-1}\mathcal{O}_{\cdot k}} \right| \leq \max_{w \in \mathbb{R}^N} \left| \frac{w^T B_{(k)}(z)^{-2}w}{w^T B_{(k)}(z)^{-1}w} \right| \leq \left\| B_{(k)}(z)^{-1} \right\|_{\mathrm{op}}.$$

The eigenvalues of $B_{(k)}(z)^{-1}$ are given by $\frac{1}{\lambda_i-z}$ where the $\lambda_i > 0$ are the eigenvalues of the symmetric matrix $\frac{1}{N}\mathcal{O}K_{(k)}\mathcal{O}^T$: $\left\| B_{(k)}(z)^{-1} \right\|_{\mathrm{op}} \leq \max_i \frac{1}{|\lambda_i-z|}$ is also bounded by $\frac{1}{|z|}$ if $z \in \mathbb{H}_{<0}$. Thus we get

$$\left| m - m_{(k)} \right|^{2s} \leq \frac{1}{|z|^{2s}N^{2s}}.$$

**Bound on $\mathbb{E}[|m_{(k)} - g_k|^{2s}]$:** The term $\mathbb{E}\left[ \left( (m_{(k)} - g_k)\overline{(m_{(k)} - g_k)} \right)^s \right]$ is equal to

$$\mathbb{E}\left[ \left( \left( \frac{1}{N}\mathrm{Tr}\left[ B_{(k)}^{-1} \right] - \frac{1}{N}\mathcal{O}_{\cdot k}^T B_{(k)}^{-1}\mathcal{O}_{\cdot k} \right)\left( \frac{1}{N}\mathrm{Tr}\left[ \overline{B_{(k)}^{-1}} \right] - \frac{1}{N}\mathcal{O}_{\cdot k}^T \overline{B_{(k)}^{-1}}\mathcal{O}_{\cdot k} \right) \right)^s \right].$$

Let $\boldsymbol{B} = \left(B_{(k)}, \overline{B_{(k)}}, \dots, B_{(k)}, \overline{B_{(k)}}\right)$ and let us denote by $\boldsymbol{B}(i)$ the $i^{th}$ element of $\boldsymbol{B}$. Using Wick's formula (Lemma 19), we have

$$\mathbb{E}\left[\left|m_{(k)} - g_k\right|^{2s}\right] = \frac{1}{N^s} \sum_{\sigma \in \mathfrak{S}_{2s}^\dagger} \frac{1}{N^{s-\boldsymbol{c}(\sigma)}} 2^{2s-\boldsymbol{c}(\sigma)} \mathbb{E}\left[\prod_{c \text{ cycle of } \sigma} \frac{1}{N} \mathrm{Tr}\left[\prod_{i \in c} \boldsymbol{B}(i)\right]\right],$$

where we recall that $\mathfrak{S}_{2s}^\dagger$ is the set of permutations with no fixed points and the product over $i$ is taken according to the order given by the cycle $c$ and does not depend on the starting point. Using the fact that the eigenvalues of $B_{(k)}$ are of the form $1/(\lambda_i - z)$ with $\lambda_i \geq 0$,

$$\left|\frac{1}{N} \mathrm{Tr}\left[\prod_{i \in c} \boldsymbol{B}(i)\right]\right| \leq \frac{1}{|z|^{\#c}}.$$

Hence,

$$\mathbb{E}\left[\left|m_{(k)} - g_k\right|^{2s}\right] \leq \frac{1}{N^s} \frac{1}{|z|^{2s}} \sum_{\sigma \in \mathfrak{S}_{2s}^\dagger} \frac{2^{2s-\boldsymbol{c}(\sigma)}}{N^{s-\boldsymbol{c}(\sigma)}}.$$

Note that, since $\sigma \in \mathfrak{S}_{2s}^\dagger$, it has no fixed point, hence $\boldsymbol{c}(\sigma) \leq s$ and thus $K_s := \sup_N \sum_{\sigma \in \mathfrak{S}_{2s}^\dagger} \frac{2^{2s-\boldsymbol{c}(\sigma)}}{N^{s-\boldsymbol{c}(\sigma)}}$ is finite. This yields the inequality

$$\mathbb{E}\left[\left|m_{(k)} - g_k\right|^{2s}\right] \leq \frac{K_s}{|z|^{2s} N^s}.$$

Using the two bounds on $\mathbb{E}\left[\left|m - m_{(k)}\right|^{2s}\right]$ and $\mathbb{E}\left[\left|m_{(k)} - g_k\right|^{2s}\right]$ in Equation (12), we get

$$\mathbb{E}\left[|m - g_k|^{2s}\right] \leq \frac{\boldsymbol{c}_s}{|z|^{2s} N^s},$$

where $\boldsymbol{c}_s = 2^{2s-1}\left[1 + K_s\right]$. $\qquad\square$

As a result, we can show the concentration of the Stieltjes transform $m$ and of the $g_k$'s around the fixed point $\tilde{m}$:

**Proposition 3.** *For any $N, s \in \mathbb{N}$, and any $z \in \mathbb{H}_{<0}$, we have*

$$\mathbb{E}\left[|\tilde{m} - m|^{2s}\right] \leq \frac{\boldsymbol{c}_s \left(\mathrm{Tr}[T_K]\right)^{2s}}{|z|^{4s} N^{3s}},$$

$$\mathbb{E}\left[|\tilde{m} - g_k|^{2s}\right] \leq \frac{2^{2s-1}\boldsymbol{c}_s \left(\mathrm{Tr}[T_K]\right)^{2s}}{|z|^{4s} N^{3s}} + \frac{2^{2s-1}\boldsymbol{c}_s}{|z|^{2s} N^s}.$$

*where $\boldsymbol{c}_s$ is the same constant as in Lemma 2.*

*Proof.* The second bound is a direct consequence of the first one, Lemma 2 and convexity. It remains to prove the first bound. Recall Equation (11)

$$\tilde{m} - m = \frac{\frac{1}{N} \sum_{k=1}^\infty \frac{d_k(m-g_k)}{(1+d_k\tilde{m})(1+d_k g_k)}}{z - \frac{1}{N} \sum_{k=1}^\infty \frac{d_k}{(1+d_k\tilde{m})(1+d_k g_k)}}.$$

We first bound from below the norm of the denominator using Lemma 22: since $\tilde{m}$ and $g_k$ all lie in the cone spanned by $1$ and $-1/z$ we have

$$\left|z - \frac{1}{N} \sum_{k=1}^\infty \frac{d_k}{(1+d_k\tilde{m})(1+d_k g_k)}\right| \geq |z|.$$

Using this bound, we can bound from below $\mathbb{E}\left[|\tilde{m} - m|^{2s}\right]$ by:

$$\frac{1}{|z|^{2s} N^{2s}} \sum_{k_1,\dots,k_{2s}=1}^\infty \frac{d_{k_1} \cdots d_{k_{2s}}}{|1 + d_{k_1}\tilde{m}| \cdots |1 + d_{k_{2s}}\tilde{m}|} \mathbb{E}\left[|m - g_{k_1}| \cdots |m - g_{k_{2s}}|\right],$$

and hence, using a generalization of Cauchy-Schwarz inequality (Lemma 21), by:

$$\frac{1}{|z|^{2s}\,N^{2s}}\sum_{k_1,\ldots,k_{2s}=1}^{\infty}\frac{d_{k_1}\cdots d_{k_{2s}}}{|1+d_{k_1}\tilde{m}|\cdots|1+d_{k_{2s}}\tilde{m}|}\left(\mathbb{E}\left[|m-g_{k_1}|^{2s}\right]\cdots\mathbb{E}\left[|m-g_{k_{2s}}|^{2s}\right]\right)^{\frac{1}{2s}}.$$

Using the fact that $\Re(\tilde{m})\geq 0$ and hence $|1+d_{k_1}\tilde{m}|\geq 1$, and using Lemma 2, this gives the following upper bound:

$$\mathbb{E}\left[|m-g_k|^{2s}\right]\leq\frac{\boldsymbol{c}_s}{|z|^{4s}\,N^{3s}}\left(\mathrm{Tr}[T_K]\right)^{2s}.$$

$\square$

We now give tighter bounds for $|\tilde{m}-\mathbb{E}[m]|$ and $|\tilde{m}-\mathbb{E}[g_k]|$:

**Proposition 4.** *For any $N\in\mathbb{N}$ and any $z\in\mathbb{H}_{<0}$, we have*

$$|\tilde{m}-\mathbb{E}[m]|\ \leq\ \frac{\mathrm{Tr}[T_K]}{|z|^2\,N^2}+\frac{2\boldsymbol{c}_1\left(\mathrm{Tr}[T_K]\right)^2}{|z|^3\,N^2}+\frac{2\boldsymbol{c}_1\left(\mathrm{Tr}[T_K]\right)^4}{|z|^5\,N^4},$$

$$|\tilde{m}-\mathbb{E}[g_k]|\ \leq\ \frac{1}{|z|\,N}+\frac{\mathrm{Tr}[T_K]}{|z|^2\,N^2}+\frac{2\boldsymbol{c}_1\left(\mathrm{Tr}[T_K]\right)^2}{|z|^3\,N^2}+\frac{2\boldsymbol{c}_1\left(\mathrm{Tr}[T_K]\right)^4}{|z|^5\,N^4},$$

*where $\boldsymbol{c}_1$ is the constant in Lemma 2.*

*Proof.* **First bound:** Following similar ideas to the one which provided Equation (11), notice that

$$\tilde{m}-m=\frac{1}{z}\frac{1}{N}\sum_{k=1}^{\infty}\frac{d_k\,(\tilde{m}-g_k)}{(1+d_k\tilde{m})(1+d_kg_k)}$$

$$=\frac{1}{z}\frac{1}{N}\sum_{k=1}^{\infty}\frac{d_k\,(\tilde{m}-g_k)}{(1+d_k\tilde{m})^2}+\frac{1}{z}\frac{1}{N}\sum_{k=1}^{\infty}\frac{d_k^2\,(\tilde{m}-g_k)^2}{(1+d_k\tilde{m})^2(1+d_kg_k)}$$

$$=\frac{\tilde{m}-m}{z}\frac{1}{N}\sum_{k=1}^{\infty}\frac{d_k}{(1+d_k\tilde{m})^2}+\frac{1}{z}\frac{1}{N}\sum_{k=1}^{\infty}\frac{d_k\,(m-g_k)}{(1+d_k\tilde{m})^2}+\frac{1}{z}\frac{1}{N}\sum_{k=1}^{\infty}\frac{d_k^2\,(\tilde{m}-g_k)^2}{(1+d_k\tilde{m})^2(1+d_kg_k)},$$

hence the new identity:

$$\tilde{m}-m=\frac{\frac{1}{N}\sum_{k=1}^{\infty}\frac{d_k(m-g_k)}{(1+d_k\tilde{m})^2}+\frac{1}{N}\sum_{k=1}^{\infty}\frac{d_k^2(\tilde{m}-g_k)^2}{(1+d_k\tilde{m})^2(1+d_kg_k)}}{z-\frac{1}{N}\sum_{k=1}^{\infty}\frac{d_k}{(1+d_k\tilde{m})^2}}.$$

Again, using Lemma 22, the norm of the denominator is bounded from below by $|z|$. From Lemma 19, $\mathbb{E}[g_k]=\mathbb{E}[m_{(k)}]$, and thus from Lemma 2, $|\mathbb{E}[m-g_k]|\leq\mathbb{E}[|m-m_{(k)}|]\leq\frac{1}{|z|N}$. Furthermore, from Proposition 3, $\mathbb{E}[|g_k-\tilde{m}|^2]\leq\frac{2\boldsymbol{c}_1(\mathrm{Tr}[T_K])^2}{|z|^4N^3}+\frac{2\boldsymbol{c}_1}{|z|^2N}$. Thus, the expectation of the numerator is bounded by

$$\frac{1}{|z|\,N^2}\sum_{k=1}^{\infty}\frac{d_k}{|1+d_k\tilde{m}|^2}+\left(\frac{2\boldsymbol{c}_1\left(\mathrm{Tr}[T_K]\right)^2}{|z|^4\,N^4}+\frac{2\boldsymbol{c}_1}{|z|^2\,N^2}\right)\sum_{k=1}^{\infty}\frac{d_k^2}{|1+d_k\tilde{m}|^2}.$$

Hence, using again the inequality $|1+d_k\tilde{m}|\geq 1$, it is bounded by

$$\frac{\mathrm{Tr}[T_K]}{|z|\,N^2}+\frac{2\boldsymbol{c}_1\left(\mathrm{Tr}[T_K]\right)^2}{|z|^2\,N^2}+\frac{2\boldsymbol{c}_1\left(\mathrm{Tr}[T_K]\right)^4}{|z|^4\,N^4}.$$

This allows us to conclude that

$$|\tilde{m}-\mathbb{E}[m]|\leq\frac{\mathrm{Tr}[T_K]}{|z|^2\,N^2}+\frac{2\boldsymbol{c}_1\left(\mathrm{Tr}[T_K]\right)^2}{|z|^3\,N^2}+\frac{2\boldsymbol{c}_1\left(\mathrm{Tr}[T_K]\right)^4}{|z|^5\,N^4}.$$

**Second bound:** Since $\mathbb{E}[g_k]=\mathbb{E}[m_{(k)}]$, one has

$$|\tilde{m}-\mathbb{E}[g_k]|\leq|\tilde{m}-\tilde{m}_{(k)}|+|\tilde{m}_{(k)}-\mathbb{E}[m_{(k)}]|,$$

where $\tilde{m}_{(k)}$ is the unique solution in the cone spanned by 1 and $-1/z$ to the equation

$$\tilde{m}_{(k)} = -\frac{1}{z}\left(1 - \frac{1}{N}\sum_{m\neq k}^{\infty}\frac{d_m\tilde{m}_{(k)}}{1 + d_m\tilde{m}_{(k)}}\right).$$

From Lemma 23, $\left|\tilde{m} - \tilde{m}_{(k)}\right| \leq \frac{1}{|z|N}$. The second term $\left|\tilde{m}_{(k)} - \mathbb{E}\left[m_{(k)}\right]\right|$ is bounded by applying the first bound of this proposition to the Stieltjes transform $m_{(k)}$. As a result, we obtain

$$\left|\tilde{m} - \mathbb{E}\left[g_k\right]\right| \leq \frac{1}{|z|\,N} + \frac{\mathrm{Tr}[T_K]}{|z|^2\,N^2} + \frac{2c_1\,(\mathrm{Tr}[T_K])^2}{|z|^3\,N^2} + \frac{2c_1\,(\mathrm{Tr}[T_K])^4}{|z|^5\,N^4}.$$

$\square$

## 2.4 Properties of the effective dimension and SCT

### 2.4.1 General properties

We begin with general properties on the Signal Capture Threshold $\vartheta$ (which depends on $\lambda$, $N$ and on the eigenvalues $d_k$ of $T_K$), valid for any kernel $K$.

**Proposition 5.** *For any $\lambda > 0$, we have*

$$\lambda < \vartheta(\lambda, N) \leq \lambda + \frac{1}{N}\mathrm{Tr}[T_K], \qquad 1 \leq \partial_\lambda\vartheta(\lambda, N) \leq \frac{1}{\lambda}\vartheta(\lambda, N),$$

*moreover $\vartheta(\lambda, N)$ is decreasing as a function of $N$ and $\partial_\lambda\vartheta(\lambda, N)$ is decreasing as a function of $\lambda$.*

*Proof.* Let $\lambda > 0$.

1. Recall that $\vartheta(\lambda)$ is the unique positive real number such that

$$\vartheta(\lambda) = \lambda + \frac{\vartheta(\lambda)}{N}\mathrm{Tr}\left[T_K\left(T_K + \vartheta(\lambda)I_{\mathcal{C}}\right)^{-1}\right].$$

Since $T_K$ is a positive operator, $\mathrm{Tr}\left[T_K\left(T_K + \vartheta(\lambda)I_{\mathcal{C}}\right)^{-1}\right] \geq 0$ and thus $\vartheta(\lambda) \geq \lambda$. Moreover, $T_K + \vartheta(\lambda)I_{\mathcal{C}} \geq \vartheta(\lambda)I_{\mathcal{C}}$, thus

$$T_K\left(T_K + \vartheta(\lambda)I_{\mathcal{C}}\right)^{-1} \leq \frac{T_K}{\vartheta(\lambda)}$$

and thus $\vartheta(\lambda) \leq \lambda + \frac{1}{N}\mathrm{Tr}[T_K]$, which gives the desired inequality.

2. Differentiating Equation (5), the derivative $\partial_\lambda\vartheta(\lambda)$ is given by:

$$\partial_\lambda\vartheta(\lambda) = \frac{1}{\left(1 - \frac{1}{N}\mathrm{Tr}\left[\left(T_K\left(T_K + \vartheta(\lambda)I_{\mathcal{C}}\right)^{-1}\right)^2\right]\right)}. \tag{13}$$

Using the fact that $T_K\left(T_K + \vartheta(\lambda)I_{\mathcal{C}}\right)^{-1} \leq I_{\mathcal{C}}$, one has

$$\left(T_K\left(T_K + \vartheta(\lambda)I_{\mathcal{C}}\right)^{-1}\right)^2 \leq T_K\left(T_K + \vartheta(\lambda)I_{\mathcal{C}}\right)^{-1},$$

thus $0 \leq \frac{1}{N}\mathrm{Tr}\left[\left(T_K\left(T_K + \vartheta(\lambda)I_{\mathcal{C}}\right)^{-1}\right)^2\right] \leq \frac{1}{N}\mathrm{Tr}\left[T_K\left(T_K + \vartheta(\lambda)I_{\mathcal{C}}\right)^{-1}\right]$. Using Equation (5), $\frac{1}{N}\mathrm{Tr}\left[T_K\left(T_K + \vartheta(\lambda)I_{\mathcal{C}}\right)^{-1}\right] = 1 - \frac{\lambda}{\vartheta(\lambda)}$. This yields

$$0 \leq \frac{\lambda}{\vartheta(\lambda)} \leq 1 - \frac{1}{N}\mathrm{Tr}\left[\left(T_K\left(T_K + \vartheta(\lambda)I_{\mathcal{C}}\right)^{-1}\right)^2\right] \leq 1.$$

Inverting this inequality yields the desired inequalities.

3. In order to study the variation of $\vartheta(\lambda, N)$ as a function of $N$, we take the derivatives of Equation (5) w.r.t $\lambda$ and $N$, and notice that

$$\partial_N \vartheta(\lambda, N) = \frac{1}{N}(\lambda - \vartheta)\partial_\lambda \vartheta(\lambda, N).$$

In particular, since $\vartheta > \lambda$ and $\partial_\lambda \vartheta \geq 1$, we get that $\partial_N \vartheta(\lambda, N) < 0$ hence $\vartheta(\lambda, N)$ is decreasing as a function of $N$.

4. Finally, we conclude by noting that since $\partial_\lambda \vartheta(\lambda, N) > 0$, $\vartheta(\lambda, N)$ is an increasing function of $\lambda$ and thus, from the Equation (13) we have that $\partial_\lambda \vartheta(\lambda, N)$ is decreasing as a function of $\vartheta$ and thus as a function of $\lambda$.

$\square$

### 2.4.2 Bounds under polynomial decay hypothesis

In this subsection only, we assume that $d_k = \Theta(k^{-\beta})$ with $\beta > 1$, i.e, there exist $c_\ell$ and $c_h$ positive such that for any $k \geq 1$, $c_\ell k^{-\beta} \leq d_k \leq c_h k^{-\beta}$. We first study the asymptotic behavior of $\vartheta(0, N)$ and $\partial_\lambda \vartheta(0, N)$ as $N$ goes to infinity, then using these results, we investigate the asymptotic behavior of $\vartheta(\lambda, N)$ and $\partial_\lambda \vartheta(\lambda, N)$ as $N$ goes to infinity.

For any $t \in \mathbb{R}^+$, let $\mathcal{N}(t)$ denote the $t$-effective dimension [10, 3] defined by

$$\mathcal{N}(t) := \sum_{k=1}^\infty \frac{d_k}{t + d_k}.$$

For any $\lambda > 0$, the SCT is the unique solution of $\vartheta(\lambda, N) = \lambda + \frac{\vartheta(\lambda, N)}{N}\mathcal{N}(\vartheta(\lambda, N))$. In particular, $\vartheta(0, N)$ is the unique solution of $\mathcal{N}(\vartheta(0, N)) = N$.

Since $\mathcal{N}(t)$ is decreasing from $\infty$ to 0, in order to study the asymptotic behavior of $\vartheta(0, N)$ as $N$ goes to infinity, one has to understand the rate of explosion of $\mathcal{N}(t)$ as $t$ goes to zero, as given by the following Lemma (also found in [2, 11]):

**Lemma 6.** If $d_k = \Theta(k^{-\beta})$ with $\beta > 1$, then $\mathcal{N}(t) = \Theta(t^{-\frac{1}{\beta}})$ when $t \to 0$.

*Proof.* For any $m \in \mathbb{R}_+$, $\mathcal{N}(t) = \sum_{k \leq m} \frac{d_k}{t + d_k} + \sum_{k > m}^\infty \frac{d_k}{t + d_k} \leq m + t^{-1}\sum_{k > m} d_k$. Then there exists $c, d > 0$ such that $\sum_{k > m} d_k \leq c\sum_{k > m} k^{-\beta} \leq dm^{1-\beta}$. Thus $\mathcal{N}(t)$ is bounded by $m + dt^{-1}m^{1-\beta}$ for any $m$. Taking $m = t^{-1}m^{1-\beta}$, i.e. $m = t^{-\frac{1}{\beta}}$, one gets that $\mathcal{N}(t) \leq Ct^{-\frac{1}{\beta}}$.

For the lower bound, notice that $\mathcal{N}(t) \geq \sum_{k|d_k \geq t} \frac{d_k}{t + d_k} \geq \frac{1}{2}\#\{k \mid d_k \geq t\}$. Using the fact that there exists $c_\ell > 0$ such that $d_k \geq c_\ell k^{-\beta}$, $\#\{k \mid d_k \geq t\} \geq \#\{k \mid c_\ell k^{-\beta} \geq t\} = \left\lfloor (t/c_\ell)^{-\frac{1}{\beta}} \right\rfloor$. This yields the lower bound on $\mathcal{N}(t)$. $\square$

**Lemma 7.** If $d_k = \Theta(k^{-\beta})$ with $\beta > 1$, then $\vartheta(0, N) = \Theta\left(N^{-\beta}\right)$.

*Proof.* From the previous lemma, there exist $b_\ell, b_h > 0$ such that $b_\ell \vartheta(0, N)^{-\frac{1}{\beta}} \leq \mathcal{N}(\vartheta(0, N)) \leq b_h \vartheta(0, N)^{-\frac{1}{\beta}}$. From the definition of $\vartheta(0, N)$, $\mathcal{N}(\vartheta(0, N)) = N$, thus we get $(N/b_\ell)^{-\beta} \leq \vartheta(0, N) \leq (N/b_h)^{-\beta}$. $\square$

With no assumption on the spectrum of $T_K$, the upper bound for the derivative of the SCT $\partial_\lambda \vartheta$ obtained in Proposition 5, becomes useless in the ridgeless limit $\lambda \to 0$. Yet, with the assumption of power-law decay of the eigenvalues of $T_K$ we can refine the bound with a meaningful one. In order to obtain this we first prove a technical lemma.

**Lemma 8.** If $d_k = \Theta(k^{-\beta})$ with $\beta > 1$, then $\sup_N \partial_\lambda \vartheta(0, N) < \infty$.

*Proof.* The derivative of the SCT with respect to $\lambda$ at $\lambda = 0$ is given by:

$$\partial_\lambda \vartheta(0, N) = \frac{N}{\vartheta(0, N)\sum_{k=1}^\infty \frac{d_k}{(\vartheta(0,N)+d_k)^2}}.$$

Set $\alpha > 1$, then for all $d_k \in [\alpha^{-1}t, \alpha t]$, we have that $\frac{d_k}{(t+d_k)^2} \geq \frac{\alpha t}{(t+\alpha t)^2} = \frac{\alpha}{(1+\alpha)^2}\frac{1}{t}$. Thus,

$$t\sum_{k=1}^{\infty}\frac{d_k}{(t+d_k)^2} \geq t\sum_{\alpha^{-1}t<d_k<\alpha t}\frac{d_k}{(t+d_k)^2} \geq \frac{\alpha}{(1+\alpha)^2}\#\{k \mid \alpha^{-1}t < d_k < \alpha t\}.$$

It follows that

$$\partial_\lambda\vartheta(0, N) \leq N\frac{(1+\alpha)^2}{\alpha}\frac{1}{\#\{k \mid \alpha^{-1}\vartheta(0, N) < d_k < \alpha\vartheta(0, N)\}}$$

Now, using Lemma 7, we are going to find a value of $\alpha$ such that $\#\{k \mid \alpha^{-1}\vartheta(0, N) < d_k < \alpha\vartheta(0, N)\} \geq cN$ for some universal constant $c$: this will conclude the proof.

By using the assumption that there exist $c_\ell, c_h > 0$ such that $c_\ell k^{-\beta} \leq d_k \leq c_h k^{-\beta}$, in Lemma 7 we saw that there exist $c'_\ell, c'_h > 0$ such that $c'_\ell N^{-\beta} \leq \vartheta(0, N) \leq c'_h N^{-\beta}$. For sake of simplicity, let us assume that the ratios $\frac{c_\ell}{c'_\ell}$ and $\frac{c_h}{c'_h}$ are not integer. Hence we have

$$\begin{aligned}\#\{k \mid \alpha^{-1}\vartheta(0, N) \leq d_k \leq \alpha\vartheta(0, N)\} &\geq \#\left\{k \mid \frac{1}{\alpha c_\ell}\vartheta(0, N) \leq k^{-\beta} \leq \frac{\alpha}{c_h}\vartheta(0, N)\right\}\\
&\geq \#\left\{k \mid \frac{1}{\alpha c_\ell}c'_h N^{-\beta} \leq k^{-\beta} \leq \frac{\alpha}{c_h}c'_\ell N^{-\beta}\right\}\\
&= \left(\left\lfloor\left(\frac{\alpha c_\ell}{c'_h}\right)^{\frac{1}{\beta}}\right\rfloor - \left\lfloor\left(\frac{c_h}{\alpha c'_\ell}\right)^{\frac{1}{\beta}}\right\rfloor\right)N\end{aligned}$$

For one of the two values $\alpha \in \{\frac{c_h}{c_\ell}, \alpha = \frac{c'_h}{c'_\ell}\}$, we have a meaningful (positive) bound:

$$\#\left\{k \mid \alpha^{-1}\vartheta(0, N) \leq d_k \leq \alpha\vartheta(0, N)\right\} \geq \left|\left\lfloor\left(\frac{c_h}{c'_h}\right)^{\frac{1}{\beta}}\right\rfloor - \left\lfloor\left(\frac{c_\ell}{c'_\ell}\right)^{\frac{1}{\beta}}\right\rfloor\right|N.$$

This allows us to conclude. $\qquad\square$

**Proposition 9.** *If there exist $\beta > 1$ and $c_\ell, c_h > 0$ s.t. for any $k \in \mathbb{N}$, $c_\ell k^{-\beta} \leq d_k \leq c_h k^{-\beta}$, then for any integer $N$,*

1. $\lambda + a_\ell N^{-\beta} \leq \vartheta(\lambda, N) \leq c\lambda + a_h N^{-\beta}$,

2. $1 \leq \partial_\lambda\vartheta(\lambda, N) \leq c$,

*where $a_\ell, a_h \geq 0$ and $c \geq 1$ depend only on $c_\ell, c_h, \beta$.*

*Proof.* We start by proving the inequalities for the derivative of the SCT $\partial_\lambda\vartheta(\lambda, N)$. The left side of the inequality has already been proven in Proposition 5. For the right side, from Proposition 5, the derivative $\partial_\lambda\vartheta(\lambda, N)$ is decreasing in $\lambda$. In particular, by Lemma 8, $\partial_\lambda\vartheta(\lambda, N) \leq \sup_N\partial_\lambda\vartheta(0, N) < \infty$. Thus, the right side holds with $c := \sup_N\partial_\lambda\vartheta(0, N)$.

The inequality for the SCT $\vartheta(\lambda, N)$ is then obtained by integrating the second inequality and by using the initial value condition $a_\ell N^{-\beta} \leq \vartheta(0, N) \leq a_h N^{-\beta}$ provided by Lemma 7. $\qquad\square$

## 2.5 The Operator $A(z)$

We have now the tools to describe the moments of the operator $A(z)$ which allow us to describe the moments of the predictor $\hat{f}_\lambda$.

### 2.5.1 Expectation

Writing $\tilde{A}_{\vartheta(-z)} = T_K\left(T_K + \vartheta(-z)I_C\right)^{-1}$ and for any diagonalizable operator $A$ writing $|A|$ for the operator with the same eigenfunctions but with eigenvalues replaced by their absolute values, we have:

**Theorem 10.** *For any $z \in \mathbb{H}_{<0}$, for any $f, g \in \mathcal{C}$, we have*

$$\left| \left\langle f, \left( \mathbb{E}\left[A(z)\right] - \tilde{A}_{\vartheta(-z)} \right) g \right\rangle_S \right| \leq \left| \left\langle f, |\tilde{A}_{\vartheta(-z)}| |I_{\mathcal{C}} - \tilde{A}_{\vartheta(-z)}|g \right\rangle_S \right| \left( \frac{1}{N} + \mathcal{P}\left( \frac{\text{Tr}[K]}{|z|N} \right) \right) \quad (14)$$

*using the big-P notation of Definition 1.*

**Remark.** *Note that in particular since the polynomial implicitly embedded in $\mathcal{P}$ vanishes at $0$, the right hand side tends to $0$ as $N \to \infty$.*

*Proof.* As before, let $(f^{(k)})_{k \in \mathbb{N}}$ be the orthonormal basis of $\mathcal{C}$ defined above and $A_{k\ell}(z) = \langle f^{(k)}, A(z)f^{(\ell)} \rangle_S$. Using a symmetry argument, we first show that for any $\ell \neq k$, $\mathbb{E}\left[A_{\ell k}(z)\right] = 0$: this implies that $\mathbb{E}\left[A(z)\right]$ and $\tilde{A}_{\vartheta(-z)}$ have the same eigenfunctions $f^{(k)}$. Thus, to conclude the proof, we only need to prove Equation 14 for $f = g = f^{(k)}$.

- **Off-Diagonal terms:** By a symmetry argument, we show that the off-diagonal terms are null. Consider the map $s_k : \mathcal{C} \to \mathcal{C}$ defined by $s_k : f \mapsto f - 2\langle f, f^{(k)} \rangle_S f^{(k)}$, and note that $s_k(f^{(m)}) = f^{(m)}$ if $m \neq k$ and $s_k(f^{(k)}) = -f^{(k)}$. The map $s_k$ is a symmetry for the observations, i.e. for any observations $o_1, \ldots, o_N$, and any functions $f_1, \ldots, f_N$, the vector $(o_i(s_k(f_i)))_{i=1,\ldots,N}$ and $(o_i(f_i))_{i=1,\ldots,N}$ have the same law. Thus, the sampling operator $\mathcal{O}$ and the operator $\mathcal{O}s_k$ have the same law, hence so do $A(z)$ and $A^{s_k}(z)$, where

$$A^{s_k}(z) := \frac{1}{N} K s_k^T \mathcal{O}^T \left( \frac{1}{N} \mathcal{O} s_k K s_k^T \mathcal{O}^T - zI_N \right)^{-1} \mathcal{O} s_k.$$

  Note that $K s_k^T = s_k K$ and since $s_k^2 = \text{Id}$, $s_k K s_k^T = K$. This implies that $A^{s_k}(z) = s_k A(z) s_k$. For any $\ell \neq k$, $A_{\ell k}^{s_k}(z) = -A_{\ell k}(z)$, hence $\mathbb{E}[A_{\ell k}(z)] = 0$.

- **Diagonal terms:** Using Equation 8, we have

$$A_{kk}(z) = \frac{d_k g_k}{1 + d_k g_k} = \frac{d_k \tilde{m}}{1 + d_k \tilde{m}} + \frac{d_k(g_k - \tilde{m})}{(1 + d_k \tilde{m})(1 + d_k g_k)}$$

$$= \frac{d_k \tilde{m}}{1 + d_k \tilde{m}} + \frac{d_k(g_k - \tilde{m})}{(1 + d_k \tilde{m})^2} - \frac{d_k^2(g_k - \tilde{m})^2}{(1 + d_k \tilde{m})^2(1 + d_k g_k)}.$$

From this, using the fact that $\Re(g_k) > 0$, we obtain

$$\left| \mathbb{E}\left[A_{kk}(z)\right] - \frac{d_k \tilde{m}}{1 + d_k \tilde{m}} \right| \leq \frac{d_k \left|\mathbb{E}\left[g_k\right] - \tilde{m}\right|}{|1 + d_k \tilde{m}|^2} + \frac{d_k^2 \mathbb{E}\left[\left|g_k - \tilde{m}\right|^2\right]}{|1 + d_k \tilde{m}|^2}.$$

Using Proposition 4, we can bound the first fraction by

$$\frac{d_k \left|\mathbb{E}\left[g_k\right] - \tilde{m}\right|}{|1 + d_k \tilde{m}|^2} \leq \frac{d_k}{|1 + d_k \tilde{m}|^2} \left( \frac{1}{|z|N} + \frac{\text{Tr}[T_K]}{|z|^2 N^2} + \frac{2c_1(\text{Tr}[T_K])^2}{|z|^3 N^2} + \frac{2c_1(\text{Tr}[T_K])^4}{|z|^5 N^4} \right)$$

$$\leq \frac{d_k |\vartheta(-z)|^2}{|\vartheta(-z) + d_k|^2} \left( \frac{1}{|z|N} + \frac{\text{Tr}[T_K]}{|z|^2 N} + \frac{2c_1(\text{Tr}[T_K])^2}{|z|^3 N^2} + \frac{2c_1(\text{Tr}[T_K])^4}{|z|^5 N^4} \right)$$

$$\leq \frac{d_k}{|\vartheta(-z) + d_k|} \left| 1 - \frac{d_k}{\vartheta(-z) + d_k} \right| \left( \frac{1}{N} + \mathcal{P}\left( \frac{\text{Tr}[T_K]}{|z|N} \right) \right),$$

by substituting $\vartheta(-z) = \frac{1}{\tilde{m}(z)}$, using the bound $|\vartheta(-z)| \leq |z| + \frac{\text{Tr}[T_K]}{N}$ (see Proposition 5).

Using Proposition 3, the inequality $d_k^2 \leq d_k \text{Tr}[T_K]$ and similar arguments as above, we can bound the second fraction by

$$\frac{d_k^2 \mathbb{E}\left[\left|g_k - \tilde{m}\right|^2\right]}{|1 + d_k \tilde{m}|^2} \leq \frac{d_k^2}{|1 + d_k \tilde{m}|^2} \left( \frac{2c_1(\text{Tr}[T_K])^2}{|z|^4 N^3} + \frac{2c_1}{|z|^2 N} \right)$$

$$\leq \frac{d_k |\vartheta(-z)|^2}{|\vartheta(-z) + d_k|^2} \left( \frac{2c_1(\text{Tr}[T_K])^3}{|z|^4 N^3} + \frac{2c_1 \text{Tr}[T_K]}{|z|^2 N} \right)$$

$$\leq \frac{d_k}{|\vartheta(-z) + d_k|} \left| 1 - \frac{d_k}{\vartheta(-z) + d_k} \right| \mathcal{P}\left( \frac{\text{Tr}[T_K]}{|z|N} \right)$$

Finally, putting everything together, we get:

$$\left| \mathbb{E}\left[A_{kk}(z)\right] - \frac{d_k \tilde{m}}{1 + d_k \tilde{m}} \right| \leq \frac{d_k}{|\vartheta(-z) + d_k|} \left| 1 - \frac{d_k}{\vartheta(-z) + d_k} \right| \left( \frac{1}{N} + \mathcal{P}\left( \frac{\mathrm{Tr}[T_K]}{|z| \, N} \right) \right) \quad (15)$$

$\square$

### 2.5.2 Variance

To study the variance of $A(z)$ we will need to apply the Shermann-Morrison formula twice, to isolate the contribution of the two eigenfunctions $f^{(k)}$ and $f^{(\ell)}$. Similarly to above, we set $K_{(k\ell)} = \sum_{n \notin \{k,\ell\}} d_n f^{(n)} \otimes f^{(n)}$ and we define

$$B_{(k\ell)}(z) = \frac{1}{N}\mathcal{O}K_{(k\ell)}\mathcal{O}^T - zI_N, \qquad m_{(k\ell)}(z) = \frac{1}{N}\mathrm{Tr}\left[B_{(k\ell)}(z)^{-1}\right].$$

Note that the concentration results of Section 2.3 apply to $m_{(k\ell)}$: it concentrates around $\tilde{m}_{(k\ell)}$, the unique solution, in the cone spanned by $1$ and $-1/z$, to the equation

$$\tilde{m}_{(k\ell)} = -\frac{1}{z}\left(1 - \frac{\tilde{m}_{(k\ell)}}{N}\mathrm{Tr}\left[T_{K_{(k\ell)}}\left(T_{K_{(k\ell)}} + \tilde{m}_{(k\ell)}I_{\mathcal{C}}\right)^{-1}\right]\right).$$

In order to compute the off-diagonal entry $A_{k\ell}(z) = \frac{1}{N}d_k\mathcal{O}_{\cdot k}^T B(z)^{-1}\mathcal{O}_{\cdot \ell}$, we use the Shermann-Morrison formula twice: when applied to $B(z) = B_{(k)}(z) + \frac{d_k}{N}\mathcal{O}_{\cdot k}\mathcal{O}_{\cdot k}^T$ we get

$$B(z)^{-1} = B_{(k)}(z)^{-1} - \frac{d_k}{N}\frac{B_{(k)}(z)^{-1}\mathcal{O}_{\cdot k}\mathcal{O}_{\cdot k}^T B_{(k)}(z)^{-1}}{1 + \frac{d_k}{N}\mathcal{O}_{\cdot k}^T B_{(k)}(z)^{-1}\mathcal{O}_{\cdot k}};$$

thus, recalling that $g_{(k)} = \frac{1}{N}\mathcal{O}_{\cdot,k}^T B_{(k)}(z)^{-1}\mathcal{O}_{\cdot,k}$, we have

$$A_{k\ell}(z) = \frac{d_k}{1 + d_k g_k}\frac{1}{N}\mathcal{O}_{\cdot k}^T B_{(k)}(z)^{-1}\mathcal{O}_{\cdot \ell}.$$

We then apply the Shermann-Morrison formula to $B_{(k)}(z) = B_{(k\ell)}(z) + \frac{d_\ell}{N}\mathcal{O}_{\cdot \ell}\mathcal{O}_{\cdot \ell}^T$ and obtain

$$B_{(k)}(z)^{-1} = B_{(k\ell)}(z)^{-1} - \frac{d_\ell}{N}\frac{B_{(k\ell)}(z)^{-1}\mathcal{O}_{\cdot \ell}\mathcal{O}_{\cdot \ell}^T B_{(k\ell)}(z)^{-1}}{1 + \frac{d_\ell}{N}\mathcal{O}_{\cdot \ell}^T B_{(k\ell)}(z)^{-1}\mathcal{O}_{\cdot \ell}}.$$

Thus, we obtain the following formula for the off-diagonal entry:

$$A_{k\ell}(z) = \frac{d_k}{1 + d_k g_k}\frac{h_{k\ell}}{1 + d_\ell h_\ell} \quad (16)$$

where $h_\ell = \frac{1}{N}\left(\mathcal{O}_{\cdot \ell}\right)^T B_{(k\ell)}^{-1}(z)\mathcal{O}_{\cdot \ell}$ and $h_{k\ell} = \frac{1}{N}\left(\mathcal{O}_{\cdot k}\right)^T B_{(k\ell)}^{-1}(z)\mathcal{O}_{\cdot \ell}$.

We can apply the results of Section 2.3 showing the concentration of $g_k$ around $\tilde{m}_{(k)}$: $h_\ell$ concentrates around $\tilde{m}_{(k)}$ which itself is close to $\tilde{m}$:

**Lemma 11.** *For $z \in \mathbb{H}_{<0}$, and $s \in \mathbb{N}$, we have*

$$\mathbb{E}\left[|h_\ell - \tilde{m}|^{2s}\right] \leq \frac{\boldsymbol{a}_s\left(\mathrm{Tr}[T_K]\right)^{2s}}{|z|^{4s}N^{3s}} + \frac{\boldsymbol{b}_s}{|z|^{2s}N^s},$$

*where $\boldsymbol{a}_s, \boldsymbol{b}_s$ only depend on $s$.*

*Proof.* By convexity, for $k \neq \ell$,

$$\mathbb{E}\left[|h_\ell - \tilde{m}|^{2s}\right] \leq 2^{2s-1}\mathbb{E}\left[\left|h_\ell - \tilde{m}_{(k)}\right|^{2s}\right] + 2^{2s-1}\left|\tilde{m}_{(k)} - \tilde{m}\right|^{2s}$$

$$\leq 2^{2s-1}\left(\frac{2^{2s-1}\boldsymbol{c}_s\left(\mathrm{Tr}[T_K]\right)^{2s}}{|z|^{4s}N^{3s}} + \frac{2^{2s-1}\boldsymbol{c}_s}{|z|^{2s}N^s}\right) + \frac{2^{2s-1}}{|z|^{2s}N^{2s}}$$

where for the first term, we applied Proposition 3 to the matrix $B_{(k)}$ instead of $B$ and the second term is bounded by $\left|\tilde{m}_{(k)} - \tilde{m}\right| \leq \frac{1}{|z|N}$ by Lemma 23. Finally, letting $\boldsymbol{a}_s = 4^{2s-1}\boldsymbol{c}_s$ and $\boldsymbol{b}_s = 4^{2s-1}\boldsymbol{c}_s + 2^{2s-1}$, we obtain the result. $\square$

The scalar $h_{k\ell}$ on the other hand has $0$ expectation and, using Wick's formula (Lemma 19), its variance $\mathbb{E}\left[h_{k\ell}^2\right]$ is equal to $\frac{1}{N^2}\mathbb{E}[\mathrm{Tr}[B_{(k\ell)}^{-2}]] = \frac{1}{N}\mathbb{E}\left[\partial_z m_{(k\ell)}(z)\right]$. Since $\mathbb{E}\left[m_{(k\ell)}(z)\right]$ is close to $\tilde{m}(z)$, from Lemma 20, its derivative, and hence the variance of $h_{k\ell}$, is close to $\frac{1}{N}\partial_z \tilde{m}$:

**Lemma 12.** *For $z \in \mathbb{H}_{<0}$, we have:*

$$\left|\mathbb{E}\left[m_{(k\ell)}(z)\right] - \tilde{m}(z)\right| \leq \frac{\mathrm{Tr}[T_K]}{|z|^2\, N^2} + \frac{2c_1\,(\mathrm{Tr}[T_K])^2}{|z|^3\, N^2} + \frac{2c_1\,(\mathrm{Tr}[T_K])^4}{|z|^5\, N^4} + \frac{2}{|z|N},$$

*where $c_1$ is as in Proposition 4.*

*Proof.* We use Proposition 4 and Lemma 23 twice to obtain

$$\left|\mathbb{E}\left[m_{(k\ell)}(z)\right] - \tilde{m}(z)\right| \leq \left|\mathbb{E}\left[m_{(k\ell)}(z)\right] - \tilde{m}_{(k\ell)}(z)\right| + \left|\tilde{m}_{(k\ell)}(z) - \tilde{m}_{(k)}(z)\right| + \left|\tilde{m}_{(k)}(z) - \tilde{m}(z)\right|$$

$$\leq \frac{\mathrm{Tr}[T_K]}{|z|^2\, N^2} + \frac{2c_1\,(\mathrm{Tr}[T_K])^2}{|z|^3\, N^2} + \frac{2c_1\,(\mathrm{Tr}[T_K])^4}{|z|^5\, N^4} + \frac{2}{|z|N},$$

which yields the desired result. $\qquad\square$

To approximate the variance $\mathrm{Var}\left(\langle f^{(k)}, A_\lambda f^*\rangle_S\right)$ of the coordinate of the noiseless predictor, we need the following results regarding the covariance of the entries of $A(z)$.

**Proposition 13.** *For $z \in \mathbb{H}_{<0}$, any $k, \ell \in \mathbb{N}$, we have*

$$\left|\mathrm{Var}\left(A_{kk}(z)\right) - \frac{2}{N}\frac{d_k^2\partial_z\tilde{m}}{(1+d_k\tilde{m})^4}\right| \leq \frac{1}{N}\frac{d_k^2|\partial_z\tilde{m}|}{|1+d_k\tilde{m}|^4}\left(\frac{1}{N} + \frac{|z|}{-\Re(z)}\mathcal{P}\left(\frac{\mathrm{Tr}[T_K]}{|z|N^{\frac{1}{2}}}\right)\right)$$

$$\left|\mathrm{Var}\left(A_{k\ell}(z)\right) - \frac{1}{N}\frac{d_k^2\partial_z\tilde{m}}{(1+d_k\tilde{m})^2(1+d_\ell\tilde{m})^2}\right| \leq \frac{1}{N}\frac{d_k^2|\partial_z\tilde{m}|}{|1+d_k\tilde{m}|^2|1+d_\ell\tilde{m}|^2}\frac{|z|}{-\Re(z)}\mathcal{P}\left(\frac{\mathrm{Tr}[T_K]}{|z|N^{\frac{1}{2}}}\right)$$

$$\left|\mathrm{Cov}\left(A_{k\ell}(z), A_{\ell k}(z)\right) - \frac{1}{N}\frac{d_k d_\ell\partial_z\tilde{m}}{(1+d_k\tilde{m})^2(1+d_\ell\tilde{m})^2}\right| \leq \frac{1}{N}\frac{d_k d_\ell|\partial_z\tilde{m}|}{|1+d_k\tilde{m}|^2|1+d_\ell\tilde{m}|^2}\frac{|z|}{-\Re(z)}\mathcal{P}\left(\frac{\mathrm{Tr}[T_K]}{|z|N^{\frac{1}{2}}}\right)$$

*where we use the big-P notation of Definition 1. Whenever a value in the quadruple $(k, h, n, \ell)$ appears an odd number of times, we have*

$$\mathrm{Cov}\left(A_{kh}(z), A_{n\ell}(z)\right) = 0.$$

*Proof.* Let $s_k$ be the symmetry map in the proof of Theorem 10: the matrices $A(z)$ and $A^{s_k}(z)$ have the same law. Since $A_{\ell n}^{s_k}(z) = -A_{\ell n}(z)$ whenever exactly one of $\ell, n$ is equal to $k$, we have for $h, n, \ell$ distinct from $k$:

$$\mathrm{Cov}\left(A_{kh}(z), A_{n\ell}(z)\right) = \mathrm{Cov}\left(A_{kh}^{s_k}(z), A_{n\ell}^{s_k}(z)\right) = \mathrm{Cov}\left(-A_{kh}(z), A_{n\ell}(z)\right)$$

which implies that $\mathrm{Cov}\left(A_{kh}(z), A_{n\ell}(z)\right) = 0$ when $h, n, \ell$ are distinct from $k$. More generally, it is easy to see that $\mathrm{Cov}\left(A_{kh}(z), A_{n\ell}(z)\right) = 0$ whenever a value in the quadruple $(k, h, n, \ell)$ appears an odd number of times.

**Approximation of** $\mathrm{Var}\left(A_{kk}(z)\right)$**:** Since $\mathbb{E}[A_{kk}(z)] \approx \frac{d_k\tilde{m}}{1+d_k\tilde{m}}$ (Theorem 10), we decompose the variance of $A_{kk}(z)$ as follows:

$$\mathrm{Var}\left(A_{kk}\right) = \mathbb{E}\left[\left(A_{kk} - \frac{d_k\tilde{m}}{1+d_k\tilde{m}}\right)^2\right] - \left[\mathbb{E}\left[A_{kk}\right] - \frac{d_k\tilde{m}}{1+d_k\tilde{m}}\right]^2.$$

This gives us an approximation $\mathrm{Var}\left(A_{kk}\right) \approx \mathbb{E}\left[\left(A_{kk} - \frac{d_k\tilde{m}}{1+d_k\tilde{m}}\right)^2\right]$ since the term $\left|\mathbb{E}\left[A_{kk}\right] - \frac{d_k\tilde{m}}{1+d_k\tilde{m}}\right|^2$, by using Theorem 10, we get the following bound :

$$\left|\mathbb{E}\left[A_{kk}\right] - \frac{d_k\tilde{m}}{1+d_k\tilde{m}}\right|^2 \leq \left|\frac{d_k\tilde{m}}{(1+d_k\tilde{m})^2}\left(\frac{1}{N} + \mathcal{P}\left(\frac{\mathrm{Tr}[T_K]}{|z|N}\right)\right)\right|^2$$

$$= \frac{1}{N}\frac{d_k^2|\tilde{m}|^2}{|1+d_k\tilde{m}|^4}\left(\frac{1}{N} + 2\mathcal{P}\left(\frac{\mathrm{Tr}[T_K]}{|z|N}\right) + N\mathcal{P}\left(\frac{\mathrm{Tr}[T_K]}{|z|N}\right)^2\right)$$

Since $\mathcal{P}\left(\frac{\text{Tr}[T_K]}{|z|N}\right) = \mathcal{P}\left(\frac{\text{Tr}[T_K]}{|z|N^{1/2}}\right)$ and $N\mathcal{P}\left(\frac{\text{Tr}[T_K]}{|z|N}\right)^2 = \mathcal{P}\left(\frac{(\text{Tr}[T_K])^2}{|z|^2 N}\right)$, we can bound $\left|\mathbb{E}\left[A_{kk}\right] - \frac{d_k\tilde{m}}{1+d_k\tilde{m}}\right|^2$ by

$$\frac{1}{N}\frac{d_k^2|\tilde{m}|^2}{|1+d_k\tilde{m}|^4}\left(\frac{1}{N} + \mathcal{P}\left(\frac{\text{Tr}[T_K]}{|z|N^{\frac{1}{2}}}\right)\right).$$

Using Formula (8) for the diagonal entries of $A$, we have:

$$\left(A_{kk} - \frac{d_k\tilde{m}}{1+d_k\tilde{m}}\right)^2 = \frac{d_k^2\left[g_k - \tilde{m}\right]^2}{(1+d_k g_k)^2(1+d_k\tilde{m})^2}.$$

which can be also expressed as:

$$\left(\frac{d_k\left[g_k - \tilde{m}\right]}{(1+d_k g_k)(1+d_k\tilde{m})}\right)^2 = \left(\frac{d_k\left[g_k - \tilde{m}\right]}{(1+d_k\tilde{m})(1+d_k\tilde{m})} - \frac{d_k^2\left[g_k - \tilde{m}\right]^2}{(1+d_k g_k)(1+d_k\tilde{m})^2}\right)^2.$$

This yields

$$\mathbb{E}\left[\left(A_{kk} - \frac{d_k\tilde{m}}{1+d_k\tilde{m}}\right)^2\right] - \mathbb{E}\left[\left(\frac{d_k\left[g_k - \tilde{m}\right]}{(1+d_k\tilde{m})^2}\right)^2\right]$$

$$= -\mathbb{E}\left[\frac{d_k^2\left[g_k - \tilde{m}\right]^2}{(1+d_k g_k)(1+d_k\tilde{m})^2}\left(\frac{2d_k\left[g_k - \tilde{m}\right]}{(1+d_k\tilde{m})(1+d_k\tilde{m})} - \frac{d_k^2\left[g_k - \tilde{m}\right]^2}{(1+d_k g_k)(1+d_k\tilde{m})^2}\right)\right].$$

Using Proposition 3, the absolute value of the r.h.s. can now be bounded by

$$\frac{d_k^3\left(2\mathbb{E}\left[|g_k - \tilde{m}|^3\right] + d_k\mathbb{E}\left[|g_k - \tilde{m}|^4\right]\right)}{|1+d_k\tilde{m}|^4} \leq \frac{d_k^3}{|1+d_k\tilde{m}|^4}2\left(\frac{2^3\boldsymbol{c_2}\left(\text{Tr}[T_K]\right)^4}{|z|^8 N^6} + \frac{2^3\boldsymbol{c_2}}{|z|^4 N^2}\right)^{\frac{3}{4}}$$

$$+ \frac{d_k^4}{|1+d_k\tilde{m}|^4}\left(\frac{2^3\boldsymbol{c_2}\left(\text{Tr}[T_K]\right)^4}{|z|^8 N^6} + \frac{2^3\boldsymbol{c_2}}{|z|^4 N^2}\right)$$

$$\leq \frac{2}{N}\frac{d_k^2|\tilde{m}|^2}{|1+d_k\tilde{m}|^4}\frac{\text{Tr}[T_K]}{|\tilde{m}|^2}\left(\frac{2^{\frac{9}{4}}\boldsymbol{c_2^{\frac{3}{4}}}\left(\text{Tr}[T_K]\right)^3}{|z|^6 N^{\frac{7}{2}}} + \frac{2^{\frac{9}{4}}\boldsymbol{c_2^{\frac{3}{4}}}}{|z|^3 N^{\frac{1}{2}}}\right)$$

$$+ \frac{1}{N}\frac{d_k^2|\tilde{m}|^2}{|1+d_k\tilde{m}|^4}\frac{(\text{Tr}[T_K])^2}{|\tilde{m}|^2}\left(\frac{2^3\boldsymbol{c_2}\left(\text{Tr}[T_K]\right)^4}{|z|^8 N^5} + \frac{2^3\boldsymbol{c_2}}{|z|^4 N}\right),$$

using the inequality $(a+b)^{\frac{3}{4}} \leq a^{\frac{3}{4}} + b^{\frac{3}{4}}$ and the fact that $d_k \leq \text{Tr}[T_K]$. From Proposition 5, we have $\frac{1}{\tilde{m}^2} \leq \left(|z| + \frac{\text{Tr}[T_K]}{N}\right)^2$, so that

$$\left|\mathbb{E}\left[\left(A_{kk} - \frac{d_k\tilde{m}}{1+d_k\tilde{m}}\right)^2\right] - \mathbb{E}\left[\left(\frac{d_k\left[g_k - \tilde{m}\right]}{(1+d_k\tilde{m})^2}\right)^2\right]\right| \leq \frac{1}{N}\frac{d_k^2|\tilde{m}|^2}{|1+d_k\tilde{m}|^4}\mathcal{P}\left(\frac{\text{Tr}[T_K]}{|z|N^{\frac{1}{2}}}\right).$$

This yields the approximation $\text{Var}\left(A_{kk}\right) \approx \frac{d_k^2\mathbb{E}\left[(g_k - \tilde{m})^2\right]}{(1+d_k\tilde{m})^4}$.

Using Wick's formula (Lemma 19),

$$\mathbb{E}\left[(g_k - \tilde{m})^2\right] = \mathbb{E}\left[\left(m_{(k)} - \tilde{m}\right)^2\right] + \frac{2}{N}\mathbb{E}[\partial_z m_{(k)}(z)],$$

hence we get:

$$\frac{d_k^2\mathbb{E}\left[(g_k - \tilde{m})^2\right]}{(1+d_k\tilde{m})^4} = \frac{\frac{2}{N}d_k^2\partial_z\mathbb{E}[m_{(k)}(z)]}{(1+d_k\tilde{m})^4} + \frac{d_k^2\mathbb{E}\left[\left(m_{(k)} - \tilde{m}\right)^2\right]}{(1+d_k\tilde{m})^4}.$$

Using Proposition 3,

$$\frac{d_k^2 \mathbb{E}\left[\left|m_{(k)} - \tilde{m}\right|^2\right]}{|1 + d_k \tilde{m}|^4} \leq \frac{d_k^2}{|1 + d_k \tilde{m}|^4}\left|\frac{\boldsymbol{c}_1\left(\mathrm{Tr}[T_K]\right)^2}{|z|^4 N^3}\right|$$

$$\leq \frac{1}{N}\frac{d_k^2 |\tilde{m}|^2}{|1 + d_k \tilde{m}|^4}\mathcal{P}\left(\frac{\mathrm{Tr}[T_K]}{|z|N}\right),$$

hence the approximation $\mathrm{Var}\left(A_{kk}\right) \approx \frac{\frac{2}{N}d_k^2 \partial_z \mathbb{E}[m_{(k)}(z)]}{(1 + d_k \tilde{m})^4}$.

At last, by using the approximation $\mathbb{E}[\partial_z m_{(k)}(z)] = \mathbb{E}[\partial_z g_k(z)] \approx \partial_z \tilde{m}(z)$ (Proposition 4 and Lemma 20), we obtain

$$\left|\frac{\frac{2}{N}d_k^2 \partial_z \mathbb{E}[m_{(k)}(z)]}{(1 + d_k \tilde{m})^4} - \frac{\frac{2}{N}d_k^2 \partial_z \tilde{m}(z)}{(1 + d_k \tilde{m}(z))^4}\right|$$

$$\leq \frac{2}{N}\frac{d_k^2}{|1 + d_k \tilde{m}|^4}\frac{2}{-\Re(z)}\left(\frac{2^2 \mathrm{Tr}[T_K]}{|z|^2 N^2} + \frac{2^4 \boldsymbol{c}_1\left(\mathrm{Tr}[T_K]\right)^2}{|z|^3 N^2} + \frac{2^6 \boldsymbol{c}_1\left(\mathrm{Tr}[T_K]\right)^4}{|z|^5 N^4} + \frac{2^2}{|z|N}\right)$$

$$\leq \frac{2}{N}\frac{d_k^2 |\tilde{m}|^2}{|1 + d_k \tilde{m}|^4}\frac{2|z|}{-\Re(z)}\mathcal{P}\left(\frac{\mathrm{Tr}[T_K]}{|z|N}\right).$$

Hence we get the approximation $\mathrm{Var}\left(A_{kk}\right) \approx \frac{\frac{2}{N}d_k^2 \partial_z \tilde{m}(z)}{(1 + d_k \tilde{m}(z))^4}$, more precisely $\left|\mathrm{Var}\left(A_{kk}\right) - \frac{\frac{2}{N}d_k^2 \partial_z \tilde{m}(z)}{(1 + d_k \tilde{m}(z))^4}\right|$ is bounded by

$$\frac{2}{N}\frac{d_k^2 |\tilde{m}|^2}{|1 + d_k \tilde{m}|^4}\left(\frac{1}{N} + \mathcal{P}\left(\frac{\mathrm{Tr}[T_K]}{|z|N^{\frac{1}{2}}}\right) + \mathcal{P}\left(\frac{\mathrm{Tr}[T_K]}{|z|N}\right) + \frac{|z|}{-\Re(z)}\mathcal{P}\left(\frac{\mathrm{Tr}[T_K]}{|z|N}\right)\right).$$

Putting everything together, we get

$$\left|\mathrm{Var}\left(A_{kk}\right) - \frac{\frac{2}{N}d_k^2 \partial_z \tilde{m}(z)}{(1 + d_k \tilde{m}(z))^4}\right| \leq \frac{2}{N}\frac{d_k^2 |\tilde{m}|^2}{|1 + d_k \tilde{m}|^4}\left(\frac{1}{N} + \frac{|z|}{-\Re(z)}\mathcal{P}\left(\frac{\mathrm{Tr}[T_K]}{|z|N^{\frac{1}{2}}}\right)\right).$$

Since $\partial_z \vartheta = \frac{\partial_z \tilde{m}}{\tilde{m}^2}$, from Proposition 5 we have $|\partial_\lambda \vartheta(\lambda)| \geq 1$, i.e. $|\tilde{m}|^2 \leq |\partial_\lambda \tilde{m}|$ and thus we conclude.

**Approximation of** $\mathrm{Cov}\left(A_{k\ell}(z), A_{\ell k}(z)\right)$**:** Note that $A_{k\ell}(z) = \frac{d_k}{N}\mathcal{O}_{\cdot k}^T B(z)^{-1}\mathcal{O}_{\cdot \ell}$, hence, since $B(z)$ is symmetric,

$$A_{k\ell}(z) = \frac{d_k}{d_\ell}A_{\ell k}(z).$$

In particular, we have $\mathrm{Cov}\left(A_{k\ell}(z), A_{\ell k}(z)\right) = \frac{d_\ell}{d_k}\mathrm{Var}\left(A_{k\ell}(z)\right)$. Hence the approximation of $\mathrm{Cov}\left(A_{k\ell}(z), A_{\ell k}(z)\right)$ follows from the one of $\mathrm{Var}\left(A_{k\ell}(z)\right)$.

**Approximation of** $\mathrm{Var}\left(A_{k\ell}(z)\right)$**:** We have seen in Theorem 10 that $\mathbb{E}\left(A_{k\ell}(z)\right) = 0$: we need to bound $\mathbb{E}\left(A_{k\ell}(z)^2\right)$. Using Equation (16):

$$\mathbb{E}\left[A_{k\ell}(z)^2\right] = \mathbb{E}\left[\left(\frac{d_k}{1 + d_k g_k}\frac{h_{k\ell}}{1 + d_\ell h_\ell}\right)^2\right],$$

where we recall that $h_\ell = \frac{1}{N}\mathcal{O}_{\cdot,\ell}^T B_{(k\ell)}(z)^{-1}\mathcal{O}_{\cdot,\ell}$ and $h_{k\ell} = \frac{1}{N}\mathcal{O}_{\cdot,k}^T B_{(k\ell)}(z)^{-1}\mathcal{O}_{\cdot,\ell}$. Since

$$\frac{d_k}{1 + d_k g_k}\frac{h_{k\ell}}{1 + d_\ell h_\ell} = \frac{d_k}{1 + d_k \tilde{m}}\frac{h_{k\ell}}{1 + d_\ell \tilde{m}} - d_k h_{k\ell}\left(\frac{d_k\left(g_k - \tilde{m}\right)\left(1 + d_\ell h_\ell\right) + d_\ell\left(1 + d_k \tilde{m}\right)\left(h_\ell - \tilde{m}\right)}{\left(1 + d_k \tilde{m}\right)\left(1 + d_\ell \tilde{m}\right)\left(1 + d_k g_k\right)\left(1 + d_\ell h_\ell\right)}\right),$$
(17)

using Lemma 14 below, we get the approximation $\mathbb{E}\left[A_{k\ell}(z)^2\right] \approx \mathbb{E}\left[\frac{d_k^2 h_{k\ell}^2}{(1 + d_k \tilde{m})^2(1 + d_\ell \tilde{m})^2}\right]$. Using Wick's formula (Lemma 19 below):

$$\mathbb{E}\left[h_{k\ell}^2\right] = \frac{1}{N}\partial_z \mathbb{E}\left[m_{(k\ell)}(z)\right].$$

Hence the approximation $\mathbb{E}\left[A_{k\ell}(z)^2\right] \approx \frac{\frac{1}{N}d_k^2\partial_z\mathbb{E}\left[m_{(k\ell)}(z)\right]}{(1+d_k\tilde{m})^2(1+d_\ell\tilde{m})^2}$. At last, by using the approximation $\mathbb{E}[\partial_z m_{(kl)}(z)] \approx \partial_z\tilde{m}(z)$ (Lemma 12 above and the technical complex analysis Lemma 20 below), we can bound the difference $\left|\frac{\frac{1}{N}d_k^2\partial_z\mathbb{E}\left[m_{(k\ell)}(z)\right]}{(1+d_k\tilde{m})^2(1+d_\ell\tilde{m})^2} - \frac{\frac{1}{N}d_k^2\partial_z\tilde{m}(z)}{(1+d_k\tilde{m})^2(1+d_\ell\tilde{m})^2}\right|$ by

$$\frac{1}{N}\frac{d_k^2}{|1+d_k\tilde{m}|^2|1+d_\ell\tilde{m}|^2}\frac{2}{-\Re(z)}\left(\frac{2^2\operatorname{Tr}[T_K]}{|z|^2\,N^2} + \frac{2^4c_1\left(\operatorname{Tr}[T_K]\right)^2}{|z|^3\,N^2} + \frac{2^6c_1\left(\operatorname{Tr}[T_K]\right)^4}{|z|^5\,N^4} + \frac{2^2}{|z|N}\right)$$

$$\leq \frac{1}{N}\frac{d_k^2|\tilde{m}|^2}{|1+d_k\tilde{m}|^2|1+d_\ell\tilde{m}|^2}\frac{2|z|}{-\Re(z)}\mathcal{P}\left(\frac{\operatorname{Tr}[T_K]}{|z|N}\right)$$

Finally, we can bound the error $\left|\mathbb{E}\left[\left(A_{k\ell}(z)\right)^2\right] - \frac{1}{N}\frac{d_k^2\partial_z\tilde{m}}{(1+d_k\tilde{m})^2(1+d_\ell\tilde{m})^2}\right|$ by

$$\frac{1}{N}\frac{d_k^2|\partial_z\tilde{m}|}{|1+d_k\tilde{m}|^2\,|1+d_\ell\tilde{m}|^2}\left(\mathcal{P}\left(\frac{\operatorname{Tr}[T_K]}{|z|N^{\frac{1}{2}}}\right) + \frac{2|z|}{-\Re(z)}\mathcal{P}\left(\frac{\operatorname{Tr}[T_K]}{|z|N^{\frac{1}{2}}}\right)\right)$$

$$\leq \frac{1}{N}\frac{d_k^2|\partial_z\tilde{m}|}{|1+d_k\tilde{m}|^2\,|1+d_\ell\tilde{m}|^2}\frac{|z|}{-\Re(z)}\mathcal{P}\left(\frac{\operatorname{Tr}[T_K]}{|z|N^{\frac{1}{2}}}\right).$$

$\square$

**Lemma 14.** *Using the same notation as in the proof of Proposition 13,*

$$\epsilon_{k\ell} = \mathbb{E}\left[A_{k\ell}(z)^2\right] - \frac{d_k^2 h_{k\ell}^2}{(1+d_k\tilde{m})^2(1+d_\ell\tilde{m})^2}$$

*is bounded by:*

$$|\epsilon_{k\ell}| \leq \frac{1}{N}\frac{d_k^2\partial_\lambda\tilde{m}}{|1+d_\ell\tilde{m}|^2\,|1+d_k\tilde{m}|^2}\mathcal{P}\left(\frac{\operatorname{Tr}[T_K]}{|z|N^{\frac{1}{2}}}\right)$$

*Proof.* Using Equation 17, by setting $c = 2\frac{1}{1+d_k\tilde{m}}\frac{1}{1+d_\ell\tilde{m}}$, $X_1 = d_k h_{k\ell}$, and

$$X_2 = \frac{d_k\left(g_k - \tilde{m}\right)}{\left(1+d_k\tilde{m}\right)\left(1+d_\ell\tilde{m}\right)\left(1+d_k g_k\right)} + \frac{d_\ell\left(h_\ell - \tilde{m}\right)}{\left(1+d_\ell\tilde{m}\right)\left(1+d_k g_k\right)\left(1+d_\ell h_\ell\right)},$$

we have that $\epsilon_{k\ell}$ is equal to:

$$\epsilon_{k\ell} = \mathbb{E}\left[-X_1^2 X_2(c - X_2)\right]$$

we can thus control $\epsilon_{k\ell}$ with the following bound

$$|\epsilon_{k\ell}| \leq c\mathbb{E}\left[\left|X_1\right|^2|X_2|\right] + \mathbb{E}\left[\left|X_1\right|^2|X_2|^2\right]$$

$$\leq \mathbb{E}\left[|X_1|^4\right]^{\frac{1}{2}}\left(c\mathbb{E}\left[|X_2|^2\right]^{\frac{1}{2}} + \mathbb{E}\left[|X_2|^4\right]^{\frac{1}{2}}\right).$$

- Bound on $\mathbb{E}[|X_1|^4]$: using the same argument as for $\mathbb{E}\left[|m_{(k)} - g_k|^{2s}\right]$ and Wick's formula (Lemma 19), there exists a constant $a$ such that

$$\mathbb{E}\left[|X_1|^4\right]^{\frac{1}{2}} = \mathbb{E}\left[|d_k h_{k\ell}|^4\right]^{\frac{1}{2}} = d_k^2\mathbb{E}\left[|h_{k\ell}|^4\right]^{\frac{1}{2}} \leq \frac{a d_k^2}{|z|^2\,N}$$

- Bound on $\mathbb{E}[|X_2|^{2s}]$: in order to bound $\mathbb{E}[|X_2|^{2s}]$ we decompose $X_2$ as $X_2 = Y_1 + Y_2 + Y_3$ where

$$Y_1 = \frac{d_k\left(g_k - \tilde{m}\right)}{\left(1+d_k\tilde{m}\right)\left(1+d_\ell\tilde{m}\right)\left(1+d_k g_k\right)},$$

$$Y_2 = \frac{d_\ell\left(h_\ell - \tilde{m}\right)}{\left(1+d_\ell\tilde{m}\right)\left(1+d_k\tilde{m}\right)\left(1+d_\ell h_\ell\right)},$$

$$Y_3 = \frac{d_\ell d_k\left(h_\ell - \tilde{m}\right)\left(\tilde{m} - g_k\right)}{\left(1+d_\ell\tilde{m}\right)\left(1+d_k\tilde{m}\right)\left(1+d_k g_k\right)\left(1+d_\ell h_\ell\right)},$$

so that by Minkowski inequality,

$$\mathbb{E}\left[|X_2|^{2s}\right]^{\frac{1}{2s}} \le \mathbb{E}\left[|Y_1|^{2s}\right]^{\frac{1}{2s}} + \mathbb{E}\left[|Y_2|^{2s}\right]^{\frac{1}{2s}} + \mathbb{E}\left[|Y_3|^{2s}\right]^{\frac{1}{2s}},$$

We can bound the terms in the r.h.s. of the above by applying Proposition 3 and Lemma 11:

– Bound on $\mathbb{E}[|Y_1|^{2s}]$:

$$\mathbb{E}\left[|Y_1|^{2s}\right]^{\frac{1}{2s}} \le \frac{d_k}{|1+d_\ell\tilde{m}|\,|1+d_k\tilde{m}|}\mathbb{E}\left[|\,(g_k-\tilde{m})\,|^{2s}\right]^{\frac{1}{2s}} \le \frac{d_k}{|1+d_\ell\tilde{m}|\,|1+d_k\tilde{m}|}\left[\frac{2^{2s-1}\boldsymbol{c}_s(\mathrm{Tr}[T_K])^{2s}}{|z|^{4s}N^{3s}} + \frac{2^{2s-1}\boldsymbol{c}_s}{|z|^{2s}N^s}\right]^{\frac{1}{2s}}$$

– Bound on $\mathbb{E}[|Y_2|^{2s}]$:

$$\mathbb{E}\left[|Y_2|^{2s}\right]^{\frac{1}{2s}} \le \frac{d_\ell}{|1+d_\ell\tilde{m}|\,|1+d_k\tilde{m}|}\mathbb{E}\left[|\,(h_\ell-\tilde{m})\,|^{2s}\right]^{\frac{1}{2s}} \le \frac{d_k}{|1+d_\ell\tilde{m}|\,|1+d_k\tilde{m}|}\left[\frac{\boldsymbol{a}_s(\mathrm{Tr}[T_K])^{2s}}{|z|^{4s}N^{3s}} + \frac{\boldsymbol{b}_s}{|z|^{2s}N^s}\right]^{\frac{1}{2s}}$$

– Bound on $\mathbb{E}\left[|Y_3|^{2s}\right]^{\frac{1}{2s}}$:

$$\mathbb{E}\left[|Y_3|^{2s}\right]^{\frac{1}{2s}} \le \frac{d_\ell d_k}{|1+d_\ell\tilde{m}|\,|1+d_k\tilde{m}|}\mathbb{E}\left[|(h_\ell-\tilde{m})|^{2s}\,|(\tilde{m}-g_k)|^{2s}\right]^{\frac{1}{2s}}$$

$$\le \frac{d_\ell d_k}{|1+d_\ell\tilde{m}|\,|1+d_k\tilde{m}|}\mathbb{E}\left[|(h_\ell-\tilde{m})|^{4s}\right]^{\frac{1}{4s}}\mathbb{E}\left[|(\tilde{m}-g_k)|^{4s}\right]^{\frac{1}{4s}}$$

$$\le \frac{d_\ell d_k}{|1+d_\ell\tilde{m}|\,|1+d_k\tilde{m}|}\left[\frac{\boldsymbol{a}_{2s}(\mathrm{Tr}[T_K])^{4s}}{|z|^{8s}N^{6s}} + \frac{\boldsymbol{b}_{2s}}{|z|^{4s}N^{2s}}\right]^{\frac{1}{4s}}\left[\frac{2^{4s-1}\boldsymbol{c}_{2s}(\mathrm{Tr}[T_K])^{4s}}{|z|^{8s}N^{6s}} + \frac{2^{4s-1}\boldsymbol{c}_{2s}}{|z|^{4s}N^{2s}}\right]^{\frac{1}{4s}}$$

Let $\boldsymbol{r}_s = \max\{2^{2s-1}\boldsymbol{c}_s, \boldsymbol{a}_s\}$ and $\boldsymbol{t}_s = \max\{2^{2s-1}\boldsymbol{c}_s, \boldsymbol{b}_s\}$; then putting the pieces together we have

$$\mathbb{E}\left[|X_2|^{2s}\right]^{\frac{1}{2s}} \le \frac{d_\ell + d_k}{|1+d_\ell\tilde{m}|\,|1+d_k\tilde{m}|}\left[\frac{\boldsymbol{r}_s(\mathrm{Tr}[T_K])^{2s}}{|z|^{4s}N^{3s}} + \frac{\boldsymbol{t}_s}{|z|^{2s}N^s}\right]^{\frac{1}{2s}} + \frac{d_\ell d_k}{|1+d_\ell\tilde{m}|\,|1+d_k\tilde{m}|}\left[\frac{\boldsymbol{r}_{2s}(\mathrm{Tr}[T_K])^{4s}}{|z|^{8s}N^{6s}} + \frac{\boldsymbol{t}_{2s}}{|z|^{4s}N^{2s}}\right]^{\frac{1}{2s}}$$

and thus

$$\mathbb{E}\left[|X_2|^{2}\right]^{\frac{1}{2}} \le \frac{d_\ell + d_k}{|1+d_\ell\tilde{m}|\,|1+d_k\tilde{m}|}\left[\frac{\boldsymbol{r}_1^{1/2}(\mathrm{Tr}[T_K])}{|z|^{2}N^{3/2}} + \frac{\boldsymbol{t}_1^{1/2}}{|z|\sqrt{N}}\right] + \frac{d_\ell d_k}{|1+d_\ell\tilde{m}|\,|1+d_k\tilde{m}|}\left[\frac{\boldsymbol{r}_2^{1/2}(\mathrm{Tr}[T_K])^2}{|z|^{4}N^3} + \frac{\boldsymbol{t}_2^{1/2}}{|z|^{2}N^1}\right]$$

$$\mathbb{E}\left[|X_4|^{4}\right]^{\frac{1}{2}} \le \frac{2(d_\ell + d_k)^2}{|1+d_\ell\tilde{m}|^2\,|1+d_k\tilde{m}|^2}\left[\frac{\boldsymbol{r}_2^{1/2}(\mathrm{Tr}[T_K])^2}{|z|^{4}N^3} + \frac{\boldsymbol{t}_2^{1/2}}{|z|^{2}N}\right] + \frac{2d_\ell^2 d_k^2}{|1+d_\ell\tilde{m}|^2\,|1+d_k\tilde{m}|^2}\left[\frac{\boldsymbol{r}_4^{1/2}(\mathrm{Tr}[T_K])^4}{|z|^{8}N^6} + \frac{\boldsymbol{t}_4^{1/2}}{|z|^{4}N^2}\right]$$

And finally, putting all the pieces together, we have

$$|\epsilon_{k\ell}| \le \mathbb{E}\left[|X_1|^4\right]^{\frac{1}{2}}\left(c\mathbb{E}\left[|X_2|^2\right]^{\frac{1}{2}} + \mathbb{E}\left[|X_2|^4\right]^{\frac{1}{2}}\right)$$

$$\le \frac{\boldsymbol{a}d_k^2}{|z|^2\,N}\frac{2(d_\ell + d_k)}{|1+d_\ell\tilde{m}|^2\,|1+d_k\tilde{m}|^2}\left[\frac{\boldsymbol{r}_1^{1/2}(\mathrm{Tr}[T_K])}{|z|^2N^{3/2}} + \frac{\boldsymbol{t}_1^{1/2}}{|z|\sqrt{N}}\right]$$

$$+ \frac{\boldsymbol{a}d_k^2}{|z|^2\,N}\frac{2(d_\ell + d_k)^2 + 2d_\ell d_k}{|1+d_\ell\tilde{m}|^2\,|1+d_k\tilde{m}|^2}\left[\frac{\boldsymbol{r}_2^{1/2}(\mathrm{Tr}[T_K])^2}{|z|^4N^3} + \frac{\boldsymbol{t}_2^{1/2}}{|z|^2N}\right]$$

$$+ \frac{\boldsymbol{a}d_k^2}{|z|^2\,N}\frac{2d_\ell^2 d_k^2}{|1+d_\ell\tilde{m}|^2\,|1+d_k\tilde{m}|^2}\left[\frac{\boldsymbol{r}_4^{1/2}(\mathrm{Tr}[T_K])^4}{|z|^8N^6} + \frac{\boldsymbol{t}_4^{1/2}}{|z|^4N^2}\right].$$

Using the fact that $|\partial_z\tilde{m}| \le |\tilde{m}|^2$ and Proposition 5, we get:

$$\frac{1}{|z|^2} \le \frac{|\partial_z\tilde{m}|}{|z|^2|\tilde{m}|^2} \le |\partial_z\tilde{m}|\left(1 + 2\frac{\mathrm{Tr}[T_K]}{|z|N} + \frac{(\mathrm{Tr}[T_K])^2}{|z|^2N^2}\right),$$

we conclude saying that

$$|\epsilon_{k\ell}| \le \frac{1}{N}\frac{d_k^2\partial_z\tilde{m}}{|1+d_\ell\tilde{m}|^2\,|1+d_k\tilde{m}|^2}\mathcal{P}\left(\frac{\mathrm{Tr}[T_K]}{|z|N^{\frac{1}{2}}}\right).$$

$\square$

**Remark.** *Since* $\tilde{m}(z) = \frac{1}{\vartheta(-z)}$, *the derivative* $\partial_z \tilde{m}(z)$ *can also be expressed in terms of the SCT:* $\partial_z \tilde{m}(z) = \partial_z \vartheta(-z) \frac{1}{\vartheta(-z)^2}$, *hence the previous approximations can also be written as:*

$$\text{Var}\left(A_{kk}(z)\right) \approx \frac{2}{N} \frac{d_k^2 \vartheta(-z)^2 \partial_z \vartheta(-z)}{(\vartheta(-z) + d_k)^4} \qquad \text{Var}\left(A_{k\ell}(z)\right) \approx \frac{1}{N} \frac{d_k^2 \vartheta(-z)^2 \partial_z \vartheta(-z)}{(\vartheta(-z) + d_k)^2 (\vartheta(-z) + d_\ell)^2}.$$

We can now describe the variance of the predictor. The variance of the predictor along the eigenfunction $f^{(k)}$ is estimated by $V_k$, where

$$V_k(f^*, \lambda, N, \epsilon) = \frac{\partial_\lambda \vartheta(\lambda)}{N} \left( \left\| (I_{\mathcal{C}} - \tilde{A}_\vartheta) f^* \right\|_S^2 + \epsilon^2 + \left\langle f^{(k)}, f^* \right\rangle_S^2 \frac{\vartheta^2(\lambda)}{(\vartheta(\lambda) + d_k)^2} \right) \frac{d_k^2}{(\vartheta(\lambda) + d_k)^2}.$$

**Theorem 15.** *There is a constant* $C_1 > 0$ *such that, with the notation of Definition 1, we have*

$$\left| \text{Var}\left( \left\langle f^{(k)}, \hat{f}_\lambda^\epsilon \right\rangle_S \right) - V_k(f^*, \lambda, N, \epsilon) \right| \leq \left( \frac{C_1}{N} + \mathcal{P}\left( \frac{\text{Tr}[T_K]}{\lambda N^{\frac{1}{2}}} \right) \right) V_k(f^*, \lambda, N, \epsilon).$$

*Proof.* Using the law of total variance, we decompose the variance with respect to the observations $\mathcal{O}$ and the vector of noise $E = (e_1, \ldots, e_N)^T$

$$\text{Var}\left( \left\langle f^{(k)}, \hat{f}_\lambda^\epsilon \right\rangle_S \right) = \text{Var}_{\mathcal{O}}\left( \left\langle f^{(k)}, \mathbb{E}_E\left[ \hat{f}_\lambda^\epsilon \right] \right\rangle_S \right) + \epsilon^2 \mathbb{E}_{\mathcal{O}} \left[ \text{Var}_E \left( \frac{d_k}{N} (\mathcal{O}_{\cdot k})^T \left( \frac{1}{N} \mathcal{O} K \mathcal{O}^T + \lambda I_N \right)^{-1} E \right) \right]$$

$$= \text{Var}_{\mathcal{O}}\left( \left\langle f^{(k)}, A(-\lambda) f^* \right\rangle_S \right) + \epsilon^2 \mathbb{E}_{\mathcal{O}} \left[ \frac{d_k}{N} \partial_\lambda A_{kk}(-\lambda) \right].$$

Since the randomness is now only on $A$ through $\mathcal{O}$, from now on, we will lighten the notation by sometimes omitting the $\mathcal{O}$ dependence in the expectations.

We first show how the approximation $V_k(f^*, \lambda, N, \epsilon)$ appears, and then establish the bounds which allow one to study the quality of this approximation.

**Approximations:** Decomposing the true function along the principal components $f^* = \sum_{k=1}^{\infty} b_k f^{(k)}$ with $b_k = \left\langle f^{(k)}, f^* \right\rangle_S$, we have

$$\text{Var}(\langle f^{(k)}, A(-\lambda) f^* \rangle_S) = \sum_\ell b_\ell^2 \text{Var}\left( A_{k\ell}(-\lambda) \right).$$

From Proposition 13 and the remark after, we have two different approximations for $\text{Var}\left( A_{k\ell}(-\lambda) \right)$. For any $\ell \neq k$, we have

$$\text{Var}\left( A_{kk}(-\lambda) \right) \approx \frac{2}{N} \frac{d_k^2 \vartheta(\lambda)^2 \partial_\lambda \vartheta(\lambda)}{(\vartheta(\lambda) + d_k)^4}, \qquad \text{Var}\left( A_{k\ell}(-\lambda) \right) \approx \frac{1}{N} \frac{d_k^2 \vartheta(\lambda)^2 \partial_\lambda \vartheta(\lambda)}{(\vartheta(\lambda) + d_k)^2 (\vartheta(\lambda) + d_\ell)^2}.$$

Hence

$$\text{Var}(\langle f^{(k)}, A_\lambda f^* \rangle_S) \approx \frac{b_k^2}{N} \frac{d_k^2 \vartheta(\lambda)^2 \partial_\lambda \vartheta(\lambda)}{(\vartheta(\lambda) + d_k)^4} + \sum_\ell \frac{b_\ell^2}{N} \frac{d_k^2 \vartheta(\lambda)^2 \partial_\lambda \vartheta(\lambda)}{(\vartheta(\lambda) + d_k)^2 (\vartheta(\lambda) + d_\ell)^2}$$

$$= \frac{\partial_\lambda \vartheta(\lambda)}{N} \left( \left\langle f^{(k)}, f^* \right\rangle_S^2 \frac{\vartheta^2(\lambda)}{(\vartheta(\lambda) + d_k)^2} + \sum_\ell b_\ell^2 \frac{\vartheta(\lambda)^2}{(\vartheta(\lambda) + d_\ell)^2} \right) \frac{d_k^2}{(\vartheta(\lambda) + d_k)^2}.$$

Since $\sum_\ell b_\ell^2 \frac{\vartheta(\lambda)^2}{(\vartheta(\lambda) + d_\ell)^2} = \|(I_{\mathcal{C}} - \tilde{A}_\vartheta) f^*\|_S^2$, this provides the approximation:

$$\text{Var}(\langle f^{(k)}, A_\lambda f^* \rangle_S) \approx \frac{\partial_\lambda \vartheta(\lambda)}{N} \left( \|(I_{\mathcal{C}} - \tilde{A}_\vartheta) f^*\|_S^2 + \langle f^{(k)}, f^* \rangle_S^2 \frac{\vartheta^2(\lambda)}{(\vartheta(\lambda) + d_k)^2} \right) \frac{d_k^2}{(\vartheta(\lambda) + d_k)^2}. \tag{18}$$

Now, using Lemma 20 and Theorem 10:

$$\epsilon^2 \mathbb{E}_{\mathcal{O}} \left[ \frac{d_k}{N} \partial_\lambda A_{kk}(-\lambda) \right] \approx \epsilon^2 \frac{\partial_\lambda \vartheta(\lambda)}{N} \frac{d_k^2}{(\vartheta(\lambda) + d_k)^2}. \tag{19}$$

Combining Equations 18 and 19, we obtain the approximation

$$\mathrm{Var}(\langle f^{(k)}, \hat{f}_\lambda^\epsilon \rangle_S) \approx V_k(f^*, \lambda, N, \epsilon).$$

Now, we explain how to quantify the quality of the approximations, and thus how to get the bound stated in the theorem. Recall that we decomposed $\mathrm{Var}(\langle f^{(k)}, \hat{f}_\lambda^\epsilon \rangle_S)$ into two terms using the law of total variance.

**First term:** We have seen that:

$$\mathrm{Var}\left( \left\langle f^{(k)}, A_\lambda f^* \right\rangle_S \right) = b_k^2 \mathrm{Var}\left( A_{kk}(-\lambda) \right) + \sum_{\ell \neq k} b_\ell^2 \mathrm{Var}\left( A_{k\ell}(-\lambda) \right).$$

By Proposition 13, we have

$$\left| b_k^2 \mathrm{Var}\left( A_{kk}(-\lambda) \right) - 2 b_k^2 \frac{\partial_\lambda \vartheta(\lambda)}{N} \frac{\vartheta(\lambda)^2 d_k^2}{(\vartheta(\lambda) + d_k)^4} \right| = b_k^2 \left| \mathrm{Var}\left( A_{kk}(-\lambda) \right) - \frac{2}{N} \frac{d_k^2 \partial_\lambda \tilde{m}}{(1 + d_k \tilde{m})^4} \right|$$

$$\leq b_k^2 \frac{|\partial_\lambda \vartheta(\lambda)|}{N} \frac{|\vartheta(\lambda)|^2 d_k^2}{|\vartheta(\lambda) + d_k|^4} \left( \frac{1}{N} + \mathcal{P}\left( \frac{\mathrm{Tr}[T_K]}{\lambda N^{\frac{1}{2}}} \right) \right)$$

and

$$\left| b_\ell^2 \mathrm{Var}\left( A_{k\ell}(-\lambda) \right) - \frac{1}{N} b_\ell^2 \frac{d_k^2 \partial_\lambda \tilde{m}}{(1 + d_k \tilde{m})^2 (1 + d_\ell \tilde{m})^2} \right| \leq b_\ell^2 \frac{1}{N} \frac{d_k^2 |\vartheta(\lambda)|^2 |\partial_\lambda \vartheta(\lambda)|}{|\vartheta(\lambda) + d_k|^2 |\vartheta(\lambda) + d_\ell|^2} \mathcal{P}\left( \frac{\mathrm{Tr}[T_K]}{\lambda N^{\frac{1}{2}}} \right).$$

Thus we have

$$\left| \sum_\ell b_\ell^2 \mathrm{Var}\left( A_{k\ell}(-\lambda) \right) - \frac{\partial_\lambda \vartheta(\lambda)}{N} \frac{d_k^2}{(\vartheta(\lambda) + d_k)^2} \left( 2 b_k^2 \frac{\vartheta(\lambda)^2}{(\vartheta(\lambda) + d_k)^2} + \sum_{\ell \neq k} b_\ell^2 \frac{\vartheta(\lambda)^2}{(\vartheta(\lambda) + d_\ell)^2} \right) \right|$$

$$\leq b_k^2 \left| \mathrm{Var}\left( A_{kk}(-\lambda) \right) - \frac{2}{N} \frac{d_k^2 \partial_\lambda \tilde{m}}{(1 + d_k \tilde{m})^4} \right| + \sum_{\ell \neq k} b_\ell^2 \left| \mathrm{Var}\left( A_{k\ell}(-\lambda) \right) - \frac{1}{N} \frac{d_k^2 \partial_\lambda \tilde{m}}{(1 + d_k \tilde{m})^2 (1 + d_\ell \tilde{m})^2} \right|$$

$$\leq b_k^2 \frac{1}{N} \frac{d_k^2 |\vartheta(\lambda)|^2 |\partial_\lambda \vartheta(\lambda)|}{|\vartheta(\lambda) + d_k|^4} \left( \frac{1}{N} + \mathcal{P}\left( \frac{\mathrm{Tr}[T_K]}{\lambda N^{\frac{1}{2}}} \right) \right) + \sum_{\ell \neq k} b_\ell^2 \frac{1}{N} \frac{d_k^2 |\vartheta(\lambda)|^2 |\partial_\lambda \vartheta(\lambda)|}{|\vartheta(\lambda) + d_k|^2 |\vartheta(\lambda) + d_\ell|^2} \mathcal{P}\left( \frac{\mathrm{Tr}[T_K]}{\lambda N^{\frac{1}{2}}} \right)$$

$$\leq \frac{|\partial_\lambda \vartheta(\lambda)|}{N} \frac{d_k^2}{|\vartheta(\lambda) + d_k|^2} \sum_\ell b_\ell^2 \frac{|\vartheta(\lambda)|^2}{|\vartheta(\lambda) + d_\ell|^2} \left( \frac{1}{N} + \mathcal{P}\left( \frac{\mathrm{Tr}[T_K]}{\lambda N^{\frac{1}{2}}} \right) \right)$$

$$\leq \frac{|\partial_\lambda \vartheta(\lambda)|}{N} \frac{d_k^2}{|\vartheta(\lambda) + d_k|^2} \left\| \left( I_{\mathcal{C}} - \tilde{A}_{\vartheta(\lambda)} \right) f^* \right\|_S^2 \left( \frac{1}{N} + \mathcal{P}\left( \frac{\mathrm{Tr}[T_K]}{\lambda N^{\frac{1}{2}}} \right) \right).$$

We deduce:

$$\left| \mathrm{Var}_{\mathcal{O}}\left( \left\langle f^{(k)}, A_\lambda f^* \right\rangle_S \right) - \frac{\partial_\lambda \vartheta(\lambda)}{N} \left( \left\| \left( I_{\mathcal{C}} - \tilde{A}_\lambda \right) f^* \right\|_S^2 + \left\langle f^{(k)}, f^* \right\rangle_S^2 \frac{\vartheta(\lambda)^2}{(\vartheta(\lambda) + d_k)^2} \right) \frac{d_k^2}{(\vartheta(\lambda) + d_k)^2} \right|$$

$$\leq \frac{\partial_\lambda \vartheta(\lambda)}{N} \left( \left\| \left( I_{\mathcal{C}} - \tilde{A}_{\vartheta(\lambda)} \right) f^* \right\|_S^2 \right) \frac{d_k^2}{(\vartheta(\lambda) + d_k)^2} \left( \frac{1}{N} + \mathcal{P}\left( \frac{\mathrm{Tr}[T_K]}{\lambda N^{\frac{1}{2}}} \right) \right)$$

$$\leq \frac{\partial_\lambda \vartheta(\lambda)}{N} \left( \left\| \left( I_{\mathcal{C}} - \tilde{A}_{\vartheta(\lambda)} \right) f^* \right\|_S^2 + \left\langle f^{(k)}, f^* \right\rangle_S^2 \frac{\vartheta(\lambda)^2}{(\vartheta(\lambda) + d_k)^2} \right) \frac{d_k^2}{(\vartheta(\lambda) + d_k)^2} \left( \frac{1}{N} + \mathcal{P}\left( \frac{\mathrm{Tr}[T_K]}{\lambda N^{\frac{1}{2}}} \right) \right).$$

**Second term:** To approximate, we apply Cauchy's inequality to Equation (15) of Theorem 10:

$$\left| \mathbb{E}\left[ \partial_z A_{kk}(z) \right] - \partial_z \vartheta(-z) \frac{d_k}{(\vartheta(-z) + d_k)^2} \right| \leq \frac{2}{-\Re(z)} \sup_{|w - z| = -\frac{1}{2} \Re(z)} \left| \mathbb{E}[A_{kk}(w)] - \frac{d_k}{\vartheta(-w) + d_k} \right|$$

$$\leq \frac{2}{-\Re(z)} \sup_{|w - z| = -\frac{1}{2} \Re(z)} \frac{d_k |\vartheta(-w)|}{|\vartheta(-w) + d_k|^2} \left( \frac{1}{N} + \mathcal{P}\left( \frac{\mathrm{Tr}[T_K]}{|w| N} \right) \right).$$

By choosing $z = -\lambda$, in the region $\{w \in \mathbb{C} \mid |w + \lambda| = \frac{\lambda}{2}\}$ the polynomial $\mathcal{P}\left(\frac{\text{Tr}[T_K]}{|w|N}\right)$ is uniformly bounded by $\mathcal{P}\left(\frac{2\text{Tr}[T_K]}{\lambda N}\right)$ and $\frac{d_k|\vartheta(-w)|}{|\vartheta(-w)+d_k|^2} \leq \frac{d_k|\vartheta(\lambda)|}{|\vartheta(\lambda)+d_k|^2}$. Thus we get

$$\left|\mathbb{E}\left[\partial_\lambda A_{kk}(-\lambda)\right] - \partial_\lambda\vartheta(\lambda)\frac{d_k}{(\vartheta(\lambda)+d_k)^2}\right| \leq 2\frac{d_k}{|\vartheta(\lambda)+d_k|^2}\frac{\vartheta(\lambda)}{\lambda}\left(\frac{1}{N} + \mathcal{P}\left(\frac{\text{Tr}[T_K]}{\lambda N}\right)\right)$$

$$\leq 2\frac{d_k}{|\vartheta(\lambda)+d_k|^2}\left(1 + \frac{\text{Tr}[T_K]}{\lambda N}\right)\left(\frac{1}{N} + \mathcal{P}\left(\frac{\text{Tr}[T_K]}{\lambda N}\right)\right)$$

$$\leq \frac{d_k}{|\vartheta(\lambda)+d_k|^2}\left(\frac{2}{N} + \mathcal{P}\left(\frac{\text{Tr}[T_K]}{\lambda N}\right)\right).$$

By using the fact that $1 \leq |\partial_\lambda\vartheta(\lambda)|$ (see Proposition 5), we have that

$$\left|\frac{d_k}{N}\mathbb{E}\left[\partial_\lambda A_{kk}(-\lambda)\right] - \frac{\partial_\lambda\vartheta(\lambda)}{N}\frac{d_k^2}{(\vartheta(\lambda)+d_k)^2}\right| \leq \frac{|\partial_\lambda\vartheta(\lambda)|}{N}\frac{d_k^2}{|\vartheta(\lambda)+d_k|^2}\left(\frac{2}{N} + \mathcal{P}\left(\frac{\text{Tr}[T_K]}{|z|N}\right)\right).$$

Finally, by putting the bounds for the two terms together we have

$$\left|\text{Var}\left(\left\langle f^{(k)}, \hat{f}_\lambda^\epsilon\right\rangle_S\right) - \frac{\partial_\lambda\vartheta(\lambda)}{N}\frac{d_k^2}{(\vartheta(\lambda)+d_k)^2}\left(2b_k^2\frac{\vartheta(\lambda)^2}{(\vartheta(\lambda)+d_k)^2} + \sum_{\ell\neq k}b_\ell^2\frac{\vartheta(\lambda)^2}{(\vartheta(\lambda)+d_\ell)^2} + \epsilon^2\right)\right|$$

$$\leq \left|\text{Var}\left(\left\langle f^{(k)}, A_\lambda f^*\right\rangle_S\right) - \frac{\partial_\lambda\vartheta(\lambda)}{N}\frac{d_k^2}{(\vartheta(\lambda)+d_k)^2}\left(2b_k^2\frac{\vartheta(\lambda)^2}{(\vartheta(\lambda)+d_k)^2} + \sum_{\ell\neq k}b_\ell^2\frac{\vartheta(\lambda)^2}{(\vartheta(\lambda)+d_\ell)^2}\right)\right|$$

$$+ \epsilon^2\frac{d_k}{N}\left|\partial_\lambda\mathbb{E}[A_{kk}(-\lambda)] - \partial_\lambda\vartheta(\lambda)\frac{d_k}{(\vartheta(\lambda)+d_k)^2}\right|$$

$$\leq \frac{\partial_\lambda\vartheta(\lambda)}{N}\left(\left\|\left(I_\mathcal{C} - \tilde{A}_{\vartheta(\lambda)}\right)f^*\right\|_S^2 + \epsilon^2\right)\frac{d_k^2}{(\vartheta(\lambda)+d_k)^2}\left(\frac{2}{N} + \mathcal{P}\left(\frac{\text{Tr}[T_K]}{\lambda N^{\frac{1}{2}}}\right)\right).$$

This concludes the proof. $\qquad\square$

## 2.6 Expected Risk

We now have all the tools required to describe the expected risk and empirical risk. In particular, we now show that the distance between the expected risk $\mathbb{E}[R^\epsilon(\hat{f}_\lambda^\epsilon)]$ and

$$\tilde{R}^\epsilon(f^*, \lambda) = \partial_\lambda\vartheta(\lambda)(\|(I_\mathcal{C} - \tilde{A}_{\vartheta(\lambda)})f^*\|_S^2 + \epsilon^2)$$

is relatively small:

**Theorem 16.** *We have*

$$\left|\mathbb{E}\left[R^\epsilon\left(\hat{f}_\lambda^\epsilon\right)\right] - \tilde{R}^\epsilon(f^*, \lambda)\right| \leq \tilde{R}^\epsilon(f^*, \lambda)\left(\frac{1}{N} + \mathcal{P}\left(\frac{\text{Tr}[T_K]}{\lambda N^{\frac{1}{2}}}\right)\right).$$

*Proof.* The expected risk can be written as $\mathbb{E}[R^\epsilon(\hat{f}_\lambda^\epsilon)] = \mathbb{E}[\|\hat{f}_\lambda^\epsilon - f^*\|_S^2] + \epsilon^2 = \sum_k \mathbb{E}[(a_k - b_k)^2] + \epsilon^2$, where $a_k = \langle f^{(k)}, \hat{f}_\lambda^\epsilon\rangle_S$ and $b_k = \langle f^{(k)}, f^*\rangle_S$. Hence, using the classical bias-variance decomposition for each summand, we get that the expected risk is equal to:

$$\mathbb{E}[R^\epsilon(\hat{f}_\lambda^\epsilon)] = R^\epsilon(\mathbb{E}[\hat{f}_\lambda^\epsilon]) + \sum_{k=1}^\infty \text{Var}(\langle f^{(k)}, \hat{f}_\lambda^\epsilon\rangle_S).$$

Similarly to the proof of Theorem 15, we explain how the approximation of the expected arises, then we establish the bounds which allow one to study the quality of this approximation.

**Approximations:** The bias term $R^\epsilon(\mathbb{E}[\hat{f}_\lambda^\epsilon])$ is equal to $\|\mathbb{E}[\hat{f}_\lambda^\epsilon] - f^*\|_S^2 + \epsilon^2 = \|(I_\mathcal{C} - \mathbb{E}[A_\lambda])f^*\|_S^2 + \epsilon^2$. Using Theorem 10, one gets the approximation of the bias term:

$$R^\epsilon(\mathbb{E}[\hat{f}_\lambda^\epsilon]) \approx \|(I_\mathcal{C} - \tilde{A}_{\vartheta(\lambda)})f^*\|_S^2 + \epsilon^2.$$

As for the variance term $\sum_{k=1}^{\infty} \mathrm{Var}(\langle f^{(k)}, \hat{f}_\lambda^\epsilon \rangle_S)$, we use Theorem 15.

$$\sum_{k=1}^{\infty} \mathrm{Var}(\langle f^{(k)}, \hat{f}_\lambda^\epsilon \rangle_S) \approx \sum_{k=1}^{\infty} V_k(f^*, \lambda, N, \epsilon),$$

where

$$V_k(f^*, \lambda, N, \epsilon) = \frac{\partial_\lambda \vartheta(\lambda)}{N} \left( \left\| (I_\mathcal{C} - \tilde{A}_\vartheta)f^* \right\|_S^2 + \epsilon^2 + \left\langle f^{(k)}, f^* \right\rangle_S^2 \frac{\vartheta^2(\lambda)}{(\vartheta(\lambda) + d_k)^2} \right) \frac{d_k^2}{(\vartheta(\lambda) + d_k)^2}.$$

Thus the variance term is approximately equal to:

$$(\|(I_\mathcal{C} - \tilde{A}_\vartheta)f^*\|_S^2 + \epsilon^2) \frac{\partial_\lambda \vartheta(\lambda)}{N} \sum_{k=1}^{\infty} \frac{d_k^2}{(\vartheta(\lambda) + d_k)^2} + \frac{\partial_\lambda \vartheta(\lambda)}{N} \sum_{k=1}^{\infty} \langle f^{(k)}, f^* \rangle_S^2 \frac{\vartheta^2(\lambda) d_k^2}{(\vartheta(\lambda) + d_k)^4}.$$

Noting that from Equation 13, we have $(\partial_\lambda \vartheta(\lambda) - 1) = \frac{\partial_\lambda \vartheta(\lambda)}{N} \sum_{k=1}^{\infty} \frac{d_k^2}{(\vartheta(\lambda) + d_k)^2}$, we get:

$$\sum_{k=1}^{\infty} \mathrm{Var}(\langle f^{(k)}, \hat{f}_\lambda^\epsilon \rangle_S) \approx (\partial_\lambda \vartheta(\lambda) - 1)(\|(I_\mathcal{C} - \tilde{A}_\vartheta)f^*\|_S^2 + \epsilon^2) + \frac{\partial_\lambda \vartheta(\lambda)}{N} \sum_{k=1}^{\infty} \langle f^{(k)}, f^* \rangle_S^2 \frac{\vartheta^2(\lambda) d_k^2}{(\vartheta(\lambda) + d_k)^4}.$$

The second term in the r.h.s. is a residual term: using the fact that $\frac{d_k^2}{(\vartheta(\lambda) + d_k)^2} \leq 1$, this term is bounded by $\frac{\partial_\lambda \vartheta(\lambda)}{N} \|(I_\mathcal{C} - \tilde{A}_{\vartheta(\lambda)})f^*\|_S^2$.

Hence, we get the following approximation of the variance term:

$$\sum_{k=1}^{\infty} \mathrm{Var}(\langle f^{(k)}, \hat{f}_\lambda^\epsilon \rangle_S) \approx (\partial_\lambda \vartheta(\lambda) - 1)(\|(I_\mathcal{C} - \tilde{A}_\vartheta)f^*\|_S^2 + \epsilon^2).$$

Putting the approximations of the bias and variance terms together, we obtain:

$$\mathbb{E}\left[ R^\epsilon \left( \hat{f}_\lambda^\epsilon \right) \right] \approx \tilde{R}^\epsilon (f^*, \lambda).$$

Now, we explain how to quantify the quality of the approximations, and thus how to get the bound stated in the theorem. Recall that, using the bias-variance decomposition, we split the expected risk into two terms, the bias term and the variance term. We show now that:

$$\left| R^\epsilon(\mathbb{E}_{\mathcal{O}, E}[\hat{f}_\lambda^\epsilon]) - \left( \|(I_\mathcal{C} - \tilde{A}_{\vartheta(\lambda)})f^*\|_S^2 + \epsilon^2 \right) \right| \leq \|(I_\mathcal{C} - \tilde{A}_{\vartheta(\lambda)})f^*\|_S^2 \left( \frac{1}{N} + \mathcal{P}\left( \frac{\mathrm{Tr}\,[T_K]}{\lambda N} \right) \right)$$

and

$$\left| \sum_{k=1}^{\infty} \mathrm{Var}(\langle f^{(k)}, \hat{f}_\lambda^\epsilon \rangle_S) - (\partial_\lambda \vartheta(\lambda) - 1) \left( \|(I_\mathcal{C} - \tilde{A}_{\vartheta(\lambda)})f^*\|_S^2 + \epsilon^2 \right) \right| \leq$$

$$\partial_\lambda \vartheta(\lambda) \left( \|(I_\mathcal{C} - \tilde{A}_{\vartheta(\lambda)})f^*\|_S^2 + \epsilon^2 \right) \left( \frac{2}{N} + \mathcal{P}\left( \frac{\mathrm{Tr}\,[T_K]}{\lambda N^{\frac{1}{2}}} \right) \right).$$

Combining the two inequations, and using the fact that $1 \leq \partial_\lambda \vartheta(\lambda)$, we then get the desired inequality.

**Bias term:** Since $|\tilde{A}_{\vartheta(\lambda), kk}| \leq 1$, Equation (15) of Theorem 10 implies that

$$\left| \tilde{A}_{\vartheta(\lambda), kk} - \mathbb{E}\left[ A_{kk}(-\lambda) \right] \right| \leq |1 - \tilde{A}_{\vartheta(\lambda), kk}| \left( \frac{1}{N} + \mathcal{P}\left( \frac{\mathrm{Tr}[T_K]}{\lambda N} \right) \right).$$

We then get

$$1 - \mathbb{E}\left[ A_{\lambda, kk} \right] \leq 1 - \tilde{A}_{\lambda, kk} + \frac{c}{\lambda^2 N} \left( 1 - \tilde{A}_{\lambda, kk} \right) \tilde{A}_{\lambda, kk} \leq \left( 1 - \tilde{A}_{\lambda, kk} \right) \left( 1 + \frac{c}{\lambda^2 N} \right).$$

We decompose the true function $f^*$ into $f^* = \sum_{k=1}^{\infty} b_k f^{(k)}$ for $b_k = \langle f^*, f^{(k)} \rangle_S$, and obtain

$$\left| R^{\epsilon}\left(\mathbb{E}_{\mathcal{O},E}\left[\hat{f}_{\lambda}^{\epsilon}\right]\right) - \left(\left\|\left(I_{\mathcal{C}} - \tilde{A}_{\vartheta(\lambda)}\right) f^*\right\|_S^2 + \epsilon^2\right)\right| = \left|\left\|(I_{\mathcal{C}} - \mathbb{E}\left[A_{\lambda}\right]) f^*\right\|_S^2 - \left\|\left(I_{\mathcal{C}} - \tilde{A}_{\vartheta(\lambda)}\right) f^*\right\|_S^2\right|$$

$$= \left|\sum_{k=1}^{\infty} b_k^2 \left((1 - \mathbb{E}\left[A_{\lambda,kk}\right])^2 - \left(1 - \tilde{A}_{\vartheta(\lambda),kk}\right)^2\right)\right|$$

$$\leq \sum_{k=1}^{\infty} b_k^2 \left|\tilde{A}_{\vartheta(\lambda),kk} - \mathbb{E}\left[A_{\lambda,kk}\right]\right| \left|2 - \tilde{A}_{\vartheta(\lambda),kk} - \mathbb{E}\left[A_{\vartheta(\lambda),kk}\right]\right|.$$

By the triangular inequality, we get that

$$\left|2 - \tilde{A}_{\vartheta(\lambda),kk} - \mathbb{E}\left[A_{\vartheta(\lambda),kk}\right]\right| \leq \left|1 - \tilde{A}_{\vartheta(\lambda),kk}\right| \left(2 + \left(\frac{1}{N} + \mathcal{P}\left(\frac{\mathrm{Tr}[T_K]}{\lambda N}\right)\right)\right)$$

and thus

$$\left|R^{\epsilon}\left(\mathbb{E}_{\mathcal{O},E}\left[\hat{f}_{\lambda}^{\epsilon}\right]\right) - \left(\left\|\left(I_{\mathcal{C}} - \tilde{A}_{\vartheta(\lambda)}\right) f^*\right\|_S^2 + \epsilon^2\right)\right|$$

$$\leq \sum_{k=1}^{\infty} b_k^2 \left|1 - \tilde{A}_{\vartheta(\lambda),kk}\right|^2 \left(2 + \frac{1}{N} + \mathcal{P}\left(\frac{\mathrm{Tr}[T_K]}{\lambda N}\right)\right)\left(\frac{1}{N} + \mathcal{P}\left(\frac{\mathrm{Tr}[T_K]}{\lambda N}\right)\right)$$

$$\leq \sum_{k=1}^{\infty} b_k^2 \left|1 - \tilde{A}_{\vartheta(\lambda),kk}\right|^2 \left(\frac{C_2}{N} + \mathcal{P}\left(\frac{\mathrm{Tr}[T_K]}{\lambda N}\right)\right).$$

**Variance term:** For the second term, recall that $(\partial_{\lambda}\vartheta(\lambda) - 1) = \frac{\partial_{\lambda}\vartheta(\lambda)}{N} \sum_{k=1}^{\infty} \frac{d_k^2}{(\vartheta(\lambda) + d_k)^2}$, and that

$$\left|\sum_{k=1}^{\infty} \mathrm{Var}\left(\left\langle f^{(k)}, \hat{f}_{\lambda}^{\epsilon}\right\rangle_S\right) - (\partial_{\lambda}\vartheta(\lambda) - 1)\left(\left\|\left(I_{\mathcal{C}} - \tilde{A}_{\vartheta(\lambda)}\right) f^*\right\|_S^2 + \epsilon^2\right)\right|$$

$$\leq \sum_{k=1}^{\infty} \left|\mathrm{Var}_{\mathcal{O}}\left(\left\langle f^{(k)}, A(-\lambda)f^*\right\rangle_S\right) - \frac{\partial_{\lambda}\vartheta(\lambda)}{N}\left(\left\|\left(I_{\mathcal{C}} - \tilde{A}_{\vartheta(\lambda)}\right) f^*\right\|_S^2 + \epsilon^2 + \left\langle f^{(k)}, f^*\right\rangle_S^2 \frac{\vartheta(\lambda)^2}{(\vartheta(\lambda) + d_k)^2}\right)\frac{d_k^2}{(\vartheta(\lambda) + d_k)^2}\right|$$

$$+ \sum_{k=1}^{\infty} \frac{\partial_{\lambda}\vartheta(\lambda)}{N}\left\langle f^{(k)}, f^*\right\rangle_S^2 \frac{\vartheta(\lambda)^2 d_k^2}{(\vartheta(\lambda) + d_k)^4}.$$

Using Theorem 15, we can control the terms in the first series: there is a constant $C_1 > 0$ such that

$$\sum_{k=1}^{\infty} \left|\mathrm{Var}_{\mathcal{O}}\left(\left\langle f^{(k)}, A(-\lambda)f^*\right\rangle_S\right) - \frac{\partial_{\lambda}\vartheta(\lambda)}{N}\left(\left\|\left(I_{\mathcal{C}} - \tilde{A}_{\vartheta(\lambda)}\right) f^*\right\|_S^2 + \epsilon^2 + \left\langle f^{(k)}, f^*\right\rangle_S^2 \frac{\vartheta(\lambda)^2}{(\vartheta(\lambda) + d_k)^2}\right)\frac{d_k^2}{(\vartheta(\lambda) + d_k)^2}\right|$$

$$\leq \left(\frac{C_1}{N} + \mathcal{P}\left(\frac{\mathrm{Tr}[T_K]}{\lambda N^{\frac{1}{2}}}\right)\right)\frac{\partial_{\lambda}\vartheta(\lambda)}{N} \sum_{k=1}^{\infty}\left(\left\|(I_{\mathcal{C}} - \tilde{A}_{\vartheta(\lambda)})f^*\right\|_S^2 + \epsilon^2 + \left\langle f^{(k)}, f^*\right\rangle_S^2 \frac{\vartheta^2(\lambda)}{(\vartheta(\lambda) + d_k)^2}\right)\frac{d_k^2}{(\vartheta(\lambda) + d_k)^2}$$

$$\leq \left(\frac{C_1}{N} + \mathcal{P}\left(\frac{\mathrm{Tr}[T_K]}{\lambda N^{\frac{1}{2}}}\right)\right)\frac{\partial_{\lambda}\vartheta(\lambda)}{N}\left(\left\|\left(I_{\mathcal{C}} - \tilde{A}_{\vartheta(\lambda)}\right)\tilde{A}_{\vartheta(\lambda)}f^*\right\|_S^2 + \left(\left\|(I_{\mathcal{C}} - \tilde{A}_{\vartheta(\lambda)})f^*\right\|_S^2 + \epsilon^2\right)\sum_{k=1}^{\infty}\frac{d_k^2}{(\vartheta(\lambda) + d_k)^2}\right)$$

$$\leq \left(\frac{C_1}{N} + \mathcal{P}\left(\frac{\mathrm{Tr}[T_K]}{\lambda N^{\frac{1}{2}}}\right)\right)\left(\frac{\partial_{\lambda}\vartheta(\lambda)}{N}\left\|\left(I_{\mathcal{C}} - \tilde{A}_{\vartheta(\lambda)}\right)\tilde{A}_{\vartheta(\lambda)}f^*\right\|_S^2 + (\partial_{\lambda}\vartheta(\lambda) - 1)\left(\left\|(I_{\mathcal{C}} - \tilde{A}_{\vartheta(\lambda)})f^*\right\|_S^2 + \epsilon^2\right)\right)$$

$$\leq \left(\frac{C_1}{N} + \mathcal{P}\left(\frac{\mathrm{Tr}[T_K]}{\lambda N^{\frac{1}{2}}}\right)\right)\left(\frac{\partial_{\lambda}\vartheta(\lambda)}{N}\left\|\left(I_{\mathcal{C}} - \tilde{A}_{\vartheta(\lambda)}\right)f^*\right\|_S^2 + (\partial_{\lambda}\vartheta(\lambda) - 1)\left(\left\|(I_{\mathcal{C}} - \tilde{A}_{\vartheta(\lambda)})f^*\right\|_S^2 + \epsilon^2\right)\right),$$

whereas for the second series, as explained already above, we have

$$\sum_{k=1}^{\infty} \frac{\partial_{\lambda}\vartheta(\lambda)}{N}\left\langle f^{(k)}, f^*\right\rangle_S^2 \frac{\vartheta(\lambda)^2 d_k^2}{(\vartheta(\lambda) + d_k)^4} = \frac{\partial_{\lambda}\vartheta(\lambda)}{N}\left\|\left(I_{\mathcal{C}} - \tilde{A}_{\vartheta(\lambda)}\right)\tilde{A}_{\vartheta(\lambda)}f^*\right\|_S^2 \leq \frac{\partial_{\lambda}\vartheta(\lambda)}{N}\left\|\left(I_{\mathcal{C}} - \tilde{A}_{\vartheta(\lambda)}\right)f^*\right\|_S^2.$$

Finally, putting the pieces together, we conclude. $\qquad\square$

## 2.7 Expected Empirical Risk

The expected empirical risk can be approximated as follows:

**Theorem 17.** *We have*

$$\left| \mathbb{E}\left[ \hat{R}^\epsilon\left( \hat{f}^\epsilon_{\lambda,E} \right) \right] - \frac{\lambda^2}{\vartheta(\lambda)^2} \tilde{R}^\epsilon\left( f^*, \lambda \right) \right| \leq \tilde{R}^\epsilon\left( f^*, \lambda \right) \mathcal{P}\left( \frac{\mathrm{Tr}\left[ T_K \right]}{\lambda N} \right).$$

*Proof.* A small computation allows one to show that:

$$\hat{R}^\epsilon\left( \hat{f}^\epsilon_{\lambda,E} \right) = \frac{\lambda^2}{N} \left( y^\epsilon \right)^T \left( \frac{1}{N} G + \lambda I_N \right)^{-2} y^\epsilon.$$

Using the definition of $y^\epsilon$ and the fact that the noise on the labels is centered and independent from the observations, this yields:

$$\mathbb{E}\left[ \hat{R}^\epsilon\left( \hat{f}^\epsilon_{\lambda,E} \right) \right] = \frac{\lambda^2}{N} f^* \mathbb{E}\left[ \mathcal{O}^T \left( \frac{1}{N} G + \lambda I_N \right)^{-2} \mathcal{O} \right] f^* + \lambda^2 \epsilon^2 \mathbb{E}\left[ \frac{1}{N} \mathrm{Tr}\left( \frac{1}{N} G + \lambda I_N \right)^{-2} \right]$$

$$= \lambda^2 \sum_{k=1}^N \langle f^{(k)}, f^* \rangle_S^2 \frac{\mathbb{E}\left[ \partial_\lambda A_{kk}(-\lambda) \right]}{d_k} + \lambda^2 \epsilon^2 \mathbb{E}\left[ \partial_z m(-\lambda) \right].$$

Similarly to the proof of Theorem 15, we explain how the approximation of the expected empirical risk appears, then we establish the bounds which allow one to study the quality of this approximation.

**Approximations:** Using Equation 19, $\mathbb{E}\left[ \partial_\lambda A_{kk}(-\lambda) \right] \approx \partial_\lambda \vartheta(\lambda) \frac{d_k}{(\vartheta(\lambda) + d_k)^2}$ hence

$$\lambda^2 \sum_{k=1}^N \langle f^{(k)}, f^* \rangle_S^2 \frac{\mathbb{E}\left[ \partial_\lambda A_{kk}(-\lambda) \right]}{d_k} \approx \frac{\partial_\lambda \vartheta(\lambda) \lambda^2}{\vartheta(\lambda)^2} \sum_{k=1}^N \langle f^{(k)}, f^* \rangle_S^2 \frac{\vartheta(\lambda)^2}{(\vartheta(\lambda) + d_k)^2}$$

$$= \frac{\partial_\lambda \vartheta(\lambda) \lambda^2}{\vartheta(\lambda)^2} \| (I_\mathcal{C} - \tilde{A}_{\vartheta(\lambda)}) f^* \|_S^2.$$

The second term can be approximated using Proposition 4 and Lemma 20: this yields

$$\mathbb{E}\left[ \partial_\lambda m(-\lambda) \right] \approx \partial_\lambda \tilde{m}(-\lambda) = \frac{\partial_\lambda \vartheta(\lambda)}{\vartheta(\lambda)^2}.$$

Hence, putting the two approximations together, the expected empirical risk is approximated by:

$$\mathbb{E}\left[ \hat{R}^\epsilon\left( \hat{f}^\epsilon_{\lambda,E} \right) \right] \approx \frac{\partial_\lambda \vartheta(\lambda) \lambda^2}{\vartheta(\lambda)^2} \left( \| (I_\mathcal{C} - \tilde{A}_{\vartheta(\lambda)}) f^* \|_S^2 + \epsilon^2 \right) = \frac{\lambda^2}{\vartheta(\lambda)^2} R^\epsilon(f^*, \lambda).$$

Now, we explain how to quantify the quality of the approximations, and thus how to get the bound stated in the theorem. Recall that, we split the expected empirical risk into two terms.

**First term**: We have already seen in Theorem 15 that by applying Lemma 20 to Equation (15) of Theorem 10 we get

$$\left| \mathbb{E}\left[ \partial_z A_{kk}(-\lambda) \right] - \partial_\lambda \vartheta(\lambda) \frac{d_k}{(\vartheta(\lambda) + d_k)^2} \right| \leq \frac{d_k}{|\vartheta(\lambda) + d_k|^2} \left( \frac{2}{N} + \mathcal{P}\left( \frac{\mathrm{Tr}[T_K]}{\lambda N} \right) \right)$$

and thus

$$
\left| \lambda^2 \sum_{k=1}^{N} \left\langle f^{(k)}, f^* \right\rangle_S^2 \frac{\mathbb{E}\left[\partial_\lambda A_{kk}(-\lambda)\right]}{d_k} - \frac{\partial_\lambda \vartheta(\lambda)\lambda^2}{\vartheta(\lambda)^2} \left\| \left( I_{\mathcal{C}} - \tilde{A}_{\vartheta(\lambda)} \right) f^* \right\|_S^2 \right|
$$

$$
\leq \lambda^2 \sum_{k=1}^{N} \left\langle f^{(k)}, f^* \right\rangle_S^2 \left| \frac{\mathbb{E}\left[\partial_\lambda A_{kk}(-\lambda)\right]}{d_k} - \frac{\partial_\lambda \vartheta(\lambda)}{\vartheta(\lambda)^2} \frac{\vartheta(\lambda)^2}{(\vartheta(\lambda) + d_k)^2} \right|
$$

$$
\leq \lambda^2 \sum_{k=1}^{N} \left\langle f^{(k)}, f^* \right\rangle_S^2 \frac{1}{|\vartheta(\lambda) + d_k|^2} \left( \frac{2}{N} + \mathcal{P}\left( \frac{\mathrm{Tr}[T_K]}{\lambda N} \right) \right)
$$

$$
= \frac{\lambda^2}{\vartheta(\lambda)^2} \left\| \left( I_{\mathcal{C}} - \tilde{A}_{\vartheta(\lambda)} \right) f^* \right\|_S^2 \left( \frac{2}{N} + \mathcal{P}\left( \frac{\mathrm{Tr}[T_K]}{\lambda N} \right) \right)
$$

$$
\leq \frac{\partial_\lambda \vartheta(\lambda)\lambda^2}{\vartheta(\lambda)^2} \left\| \left( I_{\mathcal{C}} - \tilde{A}_{\vartheta(\lambda)} \right) f^* \right\|_S^2 \left( \frac{2}{N} + \mathcal{P}\left( \frac{\mathrm{Tr}[T_K]}{\lambda N} \right) \right).
$$

**Second Term:** Using Proposition 4 and Lemma 20:

$$
\left| \mathbb{E}\left[\partial_z m(z)\right] - \partial_z \tilde{m}(z) \right| \leq \frac{|z|}{-\Re(z)} \left( \frac{2^3 \mathrm{Tr}\left[T_K\right]}{|z|^3 N^2} + \frac{2^4 c_1 \left(\mathrm{Tr}\left[T_K\right]\right)^2}{|z|^4 N^2} + \frac{2^6 c_1 \left(\mathrm{Tr}\left[T_K\right]\right)^4}{|z|^6 N^4} \right).
$$

Thus, since $\partial_\lambda \tilde{m}(-\lambda) = \frac{\partial_\lambda \vartheta(\lambda)}{\vartheta(\lambda)^2}$,

$$
\left| \lambda^2 \epsilon^2 \mathbb{E}\left[\partial_\lambda m(-\lambda)\right] - \frac{\lambda^2 \epsilon^2}{\vartheta(\lambda)^2} \partial_\lambda \vartheta(\lambda) \right| \leq \epsilon^2 \mathcal{P}\left( \frac{\mathrm{Tr}[T_K]}{\lambda N} \right).
$$

$\square$

## 2.8 Bayesian Setting

In this section, we consider the following Bayesian setting: let the true function $f^*$ be random with zero mean and covariance kernel $\Sigma(x,y) = \mathbb{E}_{f^*}[f^*(x)f^*(y)]$. We will first show that in this setting the KRR predictor with kernel $K = \Sigma$ and ridge $\lambda = \frac{\epsilon^2}{N}$ is optimal amongst all predictors which depend linearly on the noisy labels $y^\epsilon$. Second, given a kernel $K$ and a ridge $\lambda$, we provide a simple formula for the expected risk.

Let us consider predictors $\hat{f}$ that depend linearly on the labels $y^\epsilon$, i.e. for all $x$, there is a $M_x \in \mathbb{R}^N$ such that $\hat{f}(x) = M_x^T y^\epsilon$. Clearly, the KRR predictor belongs to this family of predictors. The pointwise expected squared error can be expressed for any such predictors in terms of the Gram matrix $\mathcal{O}\Sigma\mathcal{O}^T + \epsilon^2 I_N$ and the vector $\mathcal{O}\Sigma(\cdot, x)$

$$
\mathbb{E}[(M_x^T y^\epsilon - f^*(x))^2] = M_x^T (\mathcal{O}\Sigma\mathcal{O}^T + \epsilon^2 I_N)M_x - 2M_x^T \mathcal{O}\Sigma(\cdot, x) + \Sigma(x, x).
$$

Differentiating w.r.t. $M_x$, we obtain that the above error is minimized when

$$
M_x = \Sigma(x, \cdot)\mathcal{O}^T (\mathcal{O}\Sigma\mathcal{O}^T + \epsilon^2 I_N)^{-1}.
$$

In other terms, in this Bayesian setting, the KRR predictor with kernel $K = \Sigma$ and ridge $\lambda = \frac{\epsilon^2}{N}$ minimizes the expected squared error at all points $x$.

Using Theorem 16, we obtain the following approximation of the expected risk for a general kernel $K$ and ridge $\lambda$:

**Corollary 18.** *For a random true function of zero mean and covariance kernel $\Sigma$ the expected risk is approximated by*

$$
B(\lambda, K; \epsilon^2, \Sigma) = N\vartheta(\lambda, K) + N\partial_\lambda \vartheta(\lambda, K)(\frac{\epsilon^2}{N} - \lambda) + \partial_\tau \vartheta(\lambda, K + \tau(\Sigma - K))\big|_{\tau=0},
$$

*in the sense that*

$$
\left| \mathbb{E}[R^\epsilon(\hat{f}_\lambda^\epsilon)] - B(\lambda, K; \epsilon^2, \Sigma) \right| \leq B(\lambda, K; \epsilon^2, \Sigma) \left( \frac{1}{N} + \mathcal{P}\left( \frac{\mathrm{Tr}\left[T_K\right]}{|z| N^{\frac{1}{2}}} \right) \right).
$$

*Proof.* Denoting by $\mathbb{E}$ the expectation taken with respect to the data points and the noise, and by $\mathbb{E}_{f^*}$ the expectation taken with respect to the random true function $f^*$, from Theorem 16 we obtain

$$\left| \mathbb{E}_{f^*} \left[ \mathbb{E} \left[ R^\epsilon \left( \hat{f}_\lambda^\epsilon \right) \right] \right] - \mathbb{E}_{f^*} \left[ \tilde{R}^\epsilon \left( f^*, \lambda \right) \right] \right| \leq \mathbb{E}_{f^*} \left[ \left| \mathbb{E} \left[ R^\epsilon \left( \hat{f}_\lambda^\epsilon \right) \right] - \tilde{R}^\epsilon \left( f^*, \lambda \right) \right| \right]$$

$$\leq \mathbb{E}_{f^*} \left[ \tilde{R}^\epsilon \left( f^*, \lambda \right) \right] \left( \frac{1}{N} + \mathcal{P} \left( \frac{\mathrm{Tr}\,[T_K]}{|z|\, N^{\frac{1}{2}}} \right) \right)$$

it therefore suffices to show that $\mathbb{E}_{f^*} \left[ \tilde{R}^\epsilon \left( f^*, \lambda \right) \right] = B(\lambda, K; \epsilon^2, \Sigma)$.

$$\mathbb{E}_{f^*}[\tilde{R}^\epsilon(f^*, \lambda, N)] = \partial_\lambda \vartheta(\lambda) \left( \mathbb{E}_{f^*} \left[ \|(I_\mathcal{C} - \tilde{A}_\vartheta)f^*\|_S^2 \right] + \epsilon^2 \right)$$

$$= \partial_\lambda \vartheta(\lambda, K) \left( \mathrm{Tr} \left[ T_\Sigma (I_\mathcal{C} - \tilde{A}_\vartheta)^2 \right] + \epsilon^2 \right)$$

$$= \partial_\lambda \vartheta(\lambda, K) \left( \vartheta^2 \mathrm{Tr} \left[ T_K (T_K + \vartheta(\lambda, K)I_\mathcal{C})^{-2} \right] + \epsilon^2 \right)$$

$$+ \partial_\lambda \vartheta(\lambda, K) \mathrm{Tr} \left[ (T_\Sigma - T_K)(I_\mathcal{C} - \tilde{A}_\vartheta)^2 \right].$$

This formula can be further simplified. First note that differentiating both sides of Equation 5 w.r.t. to $\lambda$, we obtain that

$$\frac{\vartheta^2}{N} \mathrm{Tr} \left[ T_K (T_K + \vartheta(\lambda, K)I_\mathcal{C})^{-2} \right] = \frac{\vartheta}{\partial_\lambda \vartheta} - \lambda.$$

Secondly, differentiating both sides of Equation 5, we obtain, writing $K(\tau) = K + \tau(\Sigma - K)$

$$\partial_\tau \vartheta(\lambda, K(\tau)) = \frac{\partial_\tau \vartheta}{N} \mathrm{Tr} \left[ \tilde{A}_\vartheta \right] + \frac{\vartheta}{N} \mathrm{Tr} \left[ \partial_\tau \tilde{A}_\vartheta \right]$$

$$= \frac{\partial_\tau \vartheta}{N} \mathrm{Tr} \left[ T_K (T_K + \vartheta I_\mathcal{C})^{-1} \right] + \frac{\vartheta^2}{N} \mathrm{Tr} \left[ (T_K + \vartheta I_\mathcal{C})^{-1} T_{(\Sigma - K)} (T_K + \vartheta I_\mathcal{C})^{-1} \right]$$

$$- \frac{\partial_\tau \vartheta \vartheta}{N} \mathrm{Tr} \left[ T_K (T_K + \vartheta I_\mathcal{C})^{-2} \right]$$

$$= \frac{\partial_\tau \vartheta}{N} \mathrm{Tr} \left[ T_K (T_K + \vartheta I_\mathcal{C})^{-1} - \vartheta T_K (T_K + \vartheta I_\mathcal{C})^{-2} \right] + \frac{\vartheta^2}{N} \mathrm{Tr} \left[ T_{(\Sigma - K)} (T_K + \vartheta I_\mathcal{C})^{-2} \right]$$

$$= \frac{\partial_\tau \vartheta}{N} \mathrm{Tr} \left[ T_K^2 (T_K + \vartheta I_\mathcal{C})^{-2} \right] + \frac{\vartheta^2}{N} \mathrm{Tr} \left[ T_{(\Sigma - K)} (T_K + \vartheta I_\mathcal{C})^{-2} \right]$$

$$= \partial_\tau \vartheta - \frac{\partial_\tau \vartheta}{\partial_\lambda \vartheta} + \frac{\vartheta^2}{N} \mathrm{Tr} \left[ T_{(\Sigma - K)} (T_K + \vartheta I_\mathcal{C})^{-2} \right],$$

where we used the fact that $\frac{1}{N} \mathrm{Tr} \left[ T_K^2 (T_K + \vartheta I_\mathcal{C})^{-2} \right] = 1 - 1/\partial_\lambda \vartheta$. This implies that

$$\partial_\tau \vartheta = \partial_\lambda \vartheta \frac{\vartheta^2}{N} \mathrm{Tr} \left[ T_{(\Sigma - K)} (T_K + \vartheta I_\mathcal{C})^{-2} \right]$$

$$= \partial_\lambda \vartheta(\lambda, K) \mathrm{Tr} \left[ (T_\Sigma - T_K)(I_\mathcal{C} - \tilde{A}_\vartheta)^2 \right].$$

Putting everything together, we obtain that

$$\mathbb{E}_{f^*}[\tilde{R}^\epsilon(f^*, \lambda, N)] = N\vartheta(\lambda, K) + N\partial_\lambda \vartheta(\lambda, K)(\frac{\epsilon^2}{N} - \lambda) + \partial_\tau \vartheta(\lambda, K + \tau(\Sigma - K))\big|_{\tau=0}.$$

$\square$

## 2.9 Technical Lemmas

### 2.9.1 Matricial observations and Wick formula

For any family $\boldsymbol{A} = \left( A^{(1)}, \ldots, A^{(k)} \right)$ of $k$ square matrices of same size, any permutation $\sigma \in \mathfrak{S}_k$, we define:

$$\sigma\left( \boldsymbol{A} \right) = \prod_{c \text{ cycle of } \sigma} \mathrm{Tr} \left[ \prod_{i \in c} A^{(i)} \right],$$

where the product inside the trace is taken following the order given by the cycle and, by the cyclic property, does not depend on the starting point (see [7]). For example if $k = 4$ and $\sigma$ is the product of transpositions $(1, 3)(2, 4)$,

$$\sigma\left(\boldsymbol{A}\right) = \mathrm{Tr}(A^{(1)}A^{(3)})\mathrm{Tr}(A^{(2)}A^{(4)}).$$

The number of cycles of $\sigma$ is denoted by $\mathrm{c}(\sigma)$. The set of permutations without fixed points, i.e. such that $\sigma(i) \neq i$ for any $i \in [1, \ldots, k]$ is denoted by $\mathfrak{S}_k^\dagger$ and the set of permutations with cycles of even size is denoted by $\mathfrak{S}_k^{\mathrm{even}}$.

The following lemma, which is reminiscent of Lemma 4.5 in [1] and which is a rephrasing of Lemma C.3 of [8], is a consequence of Wick's formula for Gaussian random variables and is key to study the $g_k$ and $h_{k,\ell}$.

**Lemma 19.** *If $\boldsymbol{A} = \left(A^{(1)}, \ldots, A^{(k)}\right)$ is a family of $k$ square symmetric random matrices of size $P$ independent from a standard Gaussian vector $w$ of size $P$, we have*

$$\mathbb{E}\left[\prod_{i=1}^k w^T A^{(i)} w\right] = \sum_{\sigma \in \mathfrak{S}_k} 2^{k-\mathrm{c}(\sigma)} \mathbb{E}\left[\sigma\left(\boldsymbol{A}\right)\right], \tag{20}$$

*and,*

$$\mathbb{E}\left[\prod_{i=1}^k \left(w^T A^{(i)} w - \mathrm{Tr}\left(A^{(i)}\right)\right)\right] = \sum_{\sigma \in \mathfrak{S}_k^\dagger} 2^{k-\mathrm{c}(\sigma)} \mathbb{E}\left[\sigma\left(\boldsymbol{A}\right)\right]. \tag{21}$$

*Furthermore, if $w$ and $v$ are independent Gaussian vectors of size $P$ and independent from $\boldsymbol{A}$, then*

$$\mathbb{E}\left[\prod_{i=1}^k w^T A^{(i)} v\right] = \sum_{\sigma \in \mathfrak{S}_k^{\mathrm{even}}} \mathbb{E}\left[\sigma\left(\boldsymbol{A}\right)\right]. \tag{22}$$

*Proof.* The only differences with Lemma C.3 of [8] are in the r.h.s. and the combinatorial sets used to express the left side. We only prove Equation (20); Equations (21) and (22) can be proven similarly. Let $\boldsymbol{P}_2(2k)$ be the set of pair partitions of $\{1, \ldots, 2k\}$ and let $p \in \boldsymbol{P}_2(2k)$. Let $p\left[\boldsymbol{A}\right] = \sum_{\substack{p \leq \mathrm{Ker}(i_1, \ldots, i_{2k}) \\ i_1, \ldots, i_{2k} \in \{1, \ldots, P\}}} \mathbb{E}\left[A_{i_1 i_2}^{(1)} \ldots A_{i_{2k-1} i_{2k}}^{(k)}\right]$ where $\leq$ is the coarsed order (i.e. $p \leq q$ if $q$ is coarser than $p$) and where for any $i_1, \ldots, i_{2k}$ in $1, \ldots, P$, $\mathrm{Ker}(i_1, \ldots, i_{2k})$ is the partition of $\{1, \ldots, 2k\}$ such that two elements $u$ and $v$ in $\{1, \ldots, 2k\}$ are in the same block (i.e. pair) of $\mathrm{Ker}(i_1, \ldots, i_{2k})$ if and only if $i_u = i_v$. By Wick's formula, we have

$$\mathbb{E}\left[\prod_{i=1}^k w^T A^{(i)} w\right] = \sum_{p \in \boldsymbol{P}_2(2k)} p\left[\boldsymbol{A}\right];$$

therefore, it is sufficient to prove that

$$\sum_{p \in \boldsymbol{P}_2(2k)} p\left[\boldsymbol{A}\right] = \sum_{\sigma \in \mathfrak{S}_k} 2^{k-\mathrm{c}(\sigma)} \sigma\left(\boldsymbol{A}\right),$$

Let Po be the set of polygons on $\{1, \ldots, k\}$, i.e. the set of collections of non-crossing loops (disjoint unoriented cycles) which cover $\{1, \ldots, k\}$. Consider the two maps $F : \boldsymbol{P}_2(2k) \to \mathrm{Po}$ and $G : \mathfrak{S}_k \mapsto \mathrm{Po}$ obtained by forgetting the underlying structure: for any partition $p \in \boldsymbol{P}_2(2k)$, $F(p)$ is the collection of edges $(\ell, m)$ (viewed as collection of non-crossing loops) such that there exists $u \in \{2\ell - 1, 2\ell\}$ and $v \in \{2m - 1, 2m\}$ with $\{u, v\} \in p$; for any permutation $\sigma \in \mathfrak{S}_k$, $G(\sigma)$ is the set of its loops (unoriented cycles).

One can check that for any $\pi \in \mathrm{Po}$,

$$\#\left\{p \in \boldsymbol{P}_2(2k) \mid F(p) = \pi\right\} = 2^{k-\mathrm{c}_{\leq 2}(\pi)}, \quad \#\left\{\sigma \in \mathfrak{S}_k \mid G(\sigma) = \pi\right\} = 2^{\mathrm{c}(\pi)-\mathrm{c}_{\leq 2}(\pi)},$$

where $\mathrm{c}(\pi)$, resp. $\mathrm{c}_{\leq 2}(\pi)$, is the number of unoriented cycles, resp. unoriented cycles of size smaller than or equal to 2, of $\pi$. Note that $\mathrm{c}(\pi)$, resp. $\mathrm{c}_{\leq 2}(\pi)$ are also the number of cycles, resp.

cycles of size smaller than or equal to 2 of any $\sigma$ such that $G(\sigma) = \pi$. Notice also that, since the matrices are symmetric, for any $p, p' \in \boldsymbol{P}_2(2k)$ and any $\sigma \in \mathfrak{S}_k$, if $F(p) = F(p') = G(\sigma)$, then $p\,[\boldsymbol{A}] = p'\,[\boldsymbol{A}] = \sigma\,[\boldsymbol{A}]$. Hence:

$$\sum_{p\in\boldsymbol{P}_2(2k)} p\,[\boldsymbol{A}] = \sum_{p\in\boldsymbol{P}_2(2k)} \sum_{\pi=F(p)} p\,[\boldsymbol{A}] = \sum_{\pi\in\mathrm{Po}} \sum_{p:F(p)=\pi} \pi\,[\boldsymbol{A}] = \sum_{\pi\in\mathrm{Po}} 2^{k-\mathrm{c}_{\leq 2}(\pi)} \pi\,[\boldsymbol{A}]$$

hence

$$\sum_{p\in\boldsymbol{P}_2(2k)} p\,[\boldsymbol{A}] = \sum_{\pi\in\mathrm{Po}} 2^{k-\mathrm{c}_{\leq 2}(\pi)} \frac{1}{2^{\mathrm{c}(\pi)-\mathrm{c}_{\leq 2}(\pi)}} \sum_{\sigma:G(\sigma)=\pi} \pi\,[\boldsymbol{A}] = \sum_{\sigma\in\mathfrak{S}_k} 2^{k-\mathrm{c}(\pi)} \sigma\,[\boldsymbol{A}],$$

as required. $\qquad\square$

### 2.9.2 Bound on derivatives

Given a bound on a holomorphic function, one can obtain a bound on its derivative.

**Lemma 20.** *Let $f, g : \mathbb{H}_{<0} \to \mathbb{C}$ be two holomorphic functions such that for any $z \in \mathbb{H}_{<0}$,*

$$|f(z) - g(z)| \leq F(|z|),$$

*where $F : \mathbb{R}^+ \to \mathbb{R}$ is a decreasing function, then for any $z \in \mathbb{H}_{<0}$:*

$$|\partial_z f(z) - \partial_z g(z)| \leq \frac{2}{-\Re(z)} F\left(\frac{|z|}{2}\right),$$

*Proof.* This is a consequence of Cauchy's inequality: for any $r < -\Re(z)$ (so that the circle of center $z$ and radius $r$ lies inside $\mathbb{H}_{<0}$),

$$|\partial_z f(z) - \partial_z g(z)| \leq \frac{1}{r} \sup_{|w-z|=r} |f(w) - g(w)| \leq \frac{1}{r} \sup_{|w-z|=r} F(|w|).$$

The inequality follows by considering $r = -\frac{1}{2}\Re(z)$ and using the fact that $F$ is decreasing. $\qquad\square$

### 2.9.3 Generalized Cauchy-Schwarz inequality

Another result that we will use is the following generalization of the Cauchy-Schwarz inequality, which is a consequence of Hölder's inequality.

**Lemma 21.** *For complex random variables $a_1, ..., a_s$, we have*

$$\mathbb{E}\left[|a_1 \cdots a_s|\right] \leq \sqrt[s]{\mathbb{E}\left[|a_1|^s\right] \cdots \mathbb{E}\left[|a_s|^s\right]}.$$

*Proof.* The proof is done using an induction argument. The initialization, i.e. when $s = 1$, is trivial.

For the induction step, assume that the result is true for $s$ terms and let us prove it for $s + 1$ terms. By Hölder's inequality applied for $p = s + 1$ and $q = \frac{s+1}{s}$, we obtain:

$$\begin{aligned}
\mathbb{E}\left[|a_0 a_1 \cdots a_s|\right] &\leq \left(\mathbb{E}\left[|a_0|^{s+1}\right]\right)^{\frac{1}{s+1}} \left(\mathbb{E}\left[|a_1 \cdots a_s|^{\frac{s+1}{s}}\right]\right)^{\frac{s}{s+1}} \\
&\leq \left(\mathbb{E}\left[|a_0|^{s+1}\right]\right)^{\frac{1}{s+1}} \left(\mathbb{E}\left[|a_1|^{s+1}\right] \cdots \mathbb{E}\left[|a_s|^{s+1}\right]\right)^{\frac{1}{s+1}},
\end{aligned}$$

where the second inequality is obtained by the induction hypothesis. $\qquad\square$

### 2.9.4 Control on fixed points

**Lemma 22.** *Let $z \in \mathbb{H}_{<0}$, let $(a_k)_k$ and $(b_k)_k$ be sequences of complex numbers in the cone spanned by $1$ and $-1/z$ and let $(d_k)_k$ be positive numbers. Then*

$$\left| z - \sum_{k=1}^{\infty} \frac{d_k}{(1 + a_k)(1 + b_k)} \right| \geq |z|.$$

*Proof.* For any complex numbers $z_1$ and $z_2$, let $\Gamma_{z_1,z_2}$ be the cone spanned by $z_1$ and $z_2$, i.e. $\Gamma_{z_1,z_2} = \{w \in \mathbb{C} : w = az_1 + bz_2 \text{ for } a, b \geq 0\}$. Since $a_k, b_k \in \Gamma_{1,-1/z}$, $1/1+a_k$ and $1/1+b_k$ are in $\Gamma_{1,-z}$. All the summands $\frac{d_k}{(1+a_k)(1+b_k)}$ lie in $\Gamma_{1,z^2}$, hence so does $\sum_{k=1}^{\infty} \frac{d_k}{(1+a_k)(1+b_k)}$. Since $\Re(z) < 0$, the closest point to $z$ in this cone is $0$ and this yields the lower bound:

$$\left| z - \sum_{k=1}^{\infty} \frac{d_k}{(1+a_k)(1+b_k)} \right| \geq |z|,$$

hence the result. $\qquad\square$

Recall that $\tilde{m}(z)$, resp. $\tilde{m}_{(k)}(z)$, is the unique fixed point of the function $\psi(x) := -\frac{1}{z}\left(1 - \frac{1}{N}\sum_{\ell=1}^{\infty} \frac{d_\ell x}{1+d_\ell x}\right)$, resp. $\psi_{(k)}(x) := -\frac{1}{z}\left(1 - \frac{1}{N}\sum_{\ell\neq k} \frac{d_\ell x}{1+d_\ell x}\right)$, inside the cone spanned by $1$ and $-1/z$. We have the following control on the distance between $\tilde{m}(z)$ and $\tilde{m}_{(k)}(z)$.

**Lemma 23.** *For any $z \in \mathbb{H}_{<0}$,*

$$\left|\tilde{m}_{(k)}(z) - \tilde{m}(z)\right| \leq \frac{1}{|z|\,N}.$$

*Proof.* Let $z \in \mathbb{H}_{<0}$, $\tilde{m} = \tilde{m}(z)$ and $\tilde{m}_{(k)} = \tilde{m}_{(k)}(z)$. We have:

$$\tilde{m}_{(k)} - \tilde{m} = -\frac{1}{z}\left(-\frac{1}{N}\sum_{\ell\neq k} \frac{d_\ell \tilde{m}_{(k)}}{1+d_\ell \tilde{m}_{(k)}} + \frac{1}{N}\sum_{m} \frac{d_\ell \tilde{m}}{1+d_\ell \tilde{m}}\right)$$

$$= -\frac{1}{z}\left(\frac{1}{N}\sum_{\ell\neq k}^{\infty} \frac{d_\ell}{\left(1+d_\ell \tilde{m}_{(k)}\right)\left(1+d_\ell \tilde{m}\right)}(\tilde{m} - \tilde{m}_{(k)}) + \frac{1}{N}\frac{d_k \tilde{m}}{1+d_k \tilde{m}}\right)$$

which allows us to express the difference $\tilde{m}_{(k)} - \tilde{m}$ as

$$\tilde{m}_{(k)} - \tilde{m} = \frac{\frac{1}{N}\frac{d_k \tilde{m}}{1+d_k \tilde{m}}}{\left(\frac{1}{N}\sum_{\ell\neq k}^{\infty} \frac{d_\ell}{\left(1+d_\ell \tilde{m}_{(k)}\right)\left(1+d_\ell \tilde{m}\right)} - z\right)}.$$

Since $\tilde{m}_{(k)}$ and $\tilde{m}$ lie in the cone spanned by $1$ and $-\frac{1}{z}$, from Lemma 22, we have the lower bound on the norm of the denominator:

$$\left| \frac{1}{N}\sum_{\ell\neq k}^{\infty} \frac{d_\ell}{\left(1+d_\ell \tilde{m}_{(k)}\right)\left(1+d_\ell \tilde{m}\right)} - z \right| \geq |z|.$$

Since $\Re(\tilde{m}) \geq 0$, $|1 + d_k \tilde{m}| \geq |d_k \tilde{m}|$ and hence $\left|\frac{1}{N}\frac{d_k \tilde{m}}{1+d_k \tilde{m}}\right| \leq \frac{1}{N}$. This yields the inequality $\left|\tilde{m}_{(k)}(z) - \tilde{m}(z)\right| \leq \frac{1}{N|z|}$. $\qquad\square$