[Reviews · NeurIPS 2020]

Review 1

Summary and Contributions: The paper introduces signal capture threshold, a function of the amount of regularization, the training set size, the kernel function, and the distribution of data, for kernel ridge regression generalized for the functional setting. The parts of the target function to be learned that consist of those eigenfunctions of the underlying integral operator of the kernel on the data whose eigenvalue is larger than the threshold are learnable with the expected training set but the components with the smaller eigenvalues will be lost. These findings are then used to introduce kernel alignment risk estimator, a practical tool for model selection for KRR, such as the kernel width and ridge parameter values. Properties of SCT and KARE are experimentally evaluated on both classical data and in functional setting.

Strengths: The theoretical analysis is good and the interpretations on signal capture are elegant. New insigths on the classical method even after decades of analysis are remarkable.

Weaknesses: After reading the rebuttal, I believe the authors can easily address all my concerns. =========================================================== The generalization of kernel ridge regression for functional setting, while claiming the paper its own clear zone, makes the paper less readable for the audience that would use the proposed approaches for traditional KRR settings. The practical relevance of the assumption of the KRR estimator only depending on the first two moments of the observation distribution could be opened more. KARE resembles the well-known aproaches for computing leave-one-out cross-validation via matrix manipulations so closely that the authors should have explored this connection much more. Also the connections to spectral filtering regularization framework that provides similar interpretations should be discussed more.

Correctness: Could not spot any problems but but with the large amount of review work in a short time made it impossible to check the proofs in the supplementary properly.

Clarity: The paper is very well written but the contents is also so dense and nonverbose that it makes it cumbersome to read and may open it only for a limited audience.

Relation to Prior Work: A reader can be left missing the connections to the well-known aproaches for computing leave-one-out cross-validation via matrix manipulations. For example, the leave-one-out error formulas written in Rifkin, R. M., & Lippert, R. A. (2007). Notes on regularized least squares. are so close to KARE that the similarities and differences should be explored much more in detail than what the authors now do on page 8, as the cross-valudation formulas are no more expensive to compute than KARE, so the possible extra benefits should be described and tested more verbosely. A reader could also expect connections of SCT to the regularization by spectral filtering point of view, discussed for example the work and its references Gerfo, L. L., Rosasco, L., Odone, F., Vito, E. D., & Verri, A. (2008). Spectral algorithms for supervised learning. Neural Computation, 20(7), 1873-1897. as it also has a long history related to KRR especially in the inverse problems literature.

Reproducibility: Yes

Additional Feedback:


Review 2

Summary and Contributions: This paper re-visits Kernel Ridge Estimation by adopting an elegant reformulation, where the observations are seen as noisy images of the true function by some iid linear functional forms sampled from some distribution pi. This functional view allows an original and extremely interesting analysis of the KRR error notably by leveraging a novel quantity called Signal Capture Threshold (SCT(lambda)), depending on the level of regularization, and the eigendecomposition of the true (target) function, noisily observed. The signal along the k-th principal component of the true function is captured whenever the corresponding k-th eigenvalue is largely bigger than SCT(Lambda). There are important additionnal contributions: the true (noisy) risk can be approximated by a linear function of the empirical (noisy) risk, where the linear coefficient is the square of SCT. Realetd to this last point, KARE an estimatori fot eh true risk of KRR is also presented and studied. Numerical experiments are only in the supplements but some figures (very welcome) exhibit numerical illustrations along the paper.

Strengths: Strengths: first of all I think the paper contains strong and relevant results, present an important sum of novel and rich analysis of KRR, completely renewing the topic. * the study of the estimator error along the eigenfuction decomposition of the true function is original and very relevant * whitin the context of Kernel Ridge where so many efforts have been deployed to understand it and offer bounds to the true risk, it is an original work and a very important step in that field. * the framework enjoys remarkable properties where the Signal Capture Threshold pays a central role, meaning that with this single quantity you can resume the most important properties of the KRR estimator (variance, bias of the error)

Weaknesses: Weaknesses: I have some regret that the writing is sometimes imprecise, providing a chain of interesting results without discussing them properly. So the paper appears sometimes as not self-content, maybe suffering from the limited length of the submission. Important part of the findings are presented on the supplementary, either be theoretical or experimental. I think that the mathematical tools of Stielje's transform is not known from the NeurIPS community in general. *definition 1 is no way a definition: it is a theorem that can be proved if one set up clearly the problem to be tackled; It seems that you also assumed that the noise is not only on outputs but also on kernel values. Can you confirm ? * Eigen decomposition of a kernel, e.g. T_k is in general a difficult problem and can be a brake to use these results. A clear treatment for Laplacian or Gaussian kernels could have been added here. * Part 4 - From the study in part 3, the authors propose an estimator of the KKR risk, which is based on the previous approximations featuring SCT. This part is less clear than the rest of the paper and should benefit from some comments. * It is a bit problematic that all experiments are in the supplements.

Correctness: The claims seem correct while I have to admit that I could not check properly all the very long supplements. I hope I will have done it before the discussion.

Clarity: the paper could improve in terms of clarity (see weaknesses).

Relation to Prior Work: This work clearly differs from previous works on the domain and has given a reasonable digest of the very numerous works on the topic.

Reproducibility: Yes

Additional Feedback:


Review 3

Summary and Contributions: This paper studies the expected (empirical) risk of KRR in the large dimension case in a non-asymptotic view, and derives data-dependent bounds via the introduced two measures SCT and KARE.

Strengths: - introduce a SCT measure that relates to the effective dimension in learning theory. It is related to 1/m_G(-\lambda), Stieltjes transformation. - provides refined data-dependent analysis based on RMT and introduces KARE for the expected risk.

Weaknesses: I have read the rebuttal and other reviewers' comments. The rebuttal addressed my questions and thus I increased the score to 7. Nevertheless, I suggest the authors carefully organize the paper for clarity in the final version. =========================================================== The quality of this paper is good, but there are some issues that I concerned: Motivation and related works - The Gaussianity assumption: This paper considers the gaussian data, that follows with previous works in RMT, e.g., [17,20]. Also the following refs forcuses on the risk convergence of (centered) KRR by RMT in an asymptotic regime. [S1] Risk Convergence of Centered Kernel Ridge Regression with Large Dimensional Data. IEEE Trans on Signal Processing, 2020. It's ok, but in this case, the motivation in the introduction should be better presented when compared to current results in RMT. "however, these estimates typically require a priori knowledge of the data distribution. It remains a challenge to have estimates based on the training data alone, enabling one to make informed decisions on the choices of the ridge and of the kernel." - I think the main motivation is to extend asymptotic results to non-asymptotic ones instead of "require a priori knowledge of the data distribution". Theoretical results - It's not clear to me how does the dimension d, or the lengthscale \ell/d effects the convergence rates in the presented theorems. I guess they are associated with \delta and \vartheta? - I just take a quick glance on the supplemental material, and find that Lemma 6 regarding to the effective dimension under polynomial decay has been actually studied in learning theory, , being almost folklore, e.g., [S2] Bach. Sharp analysis of low-rank kernel matrix approximations. COLT 2013. [S3] Zhang et al. Divide and conquer kernel ridge regression: A distributed algorithm with minimax optimal rates. JMLR2015. Clarity and organization - Some notations are mixed and unclear. For example, K represents the kernel function, the kernel matrix, and the linear operator. -- The used Stieltjes transformation in finite-size regime needs carefully considered and analyzed, as indicated by [12]. It would better to illustrate this in the main part instead of supplimentary materials. - Besides, this paper needs a better organization: the main results are introduced until page 5.

Correctness: The quality of this paper is good and the claims seem correct, though I have not carefully checked the proof.

Clarity: The presentation of this paper needs clarified.

Relation to Prior Work: This paper clearly discussed the difference between this work and previous work. But, it's better to clarify the motivation.

Reproducibility: Yes

Additional Feedback:


Review 4

Summary and Contributions: The paper focuses on studying KRR predictor and its risk. To that end, it proposes two new tools, the Signal Capture Threshold (SCT) and Kernel Alignment Risk Estimator (KARE). The SCT identifies the components of the data that KRR estimator captures (identified by the eigenvalues larger than SCT of decomposition of KRR integral operator), while KARE estimates the risk with training data.

Strengths: The theoretical grounding of the work seems excellent. The contribution is very relevant to NeurIPS and ML community in general.

Weaknesses: The only weakness is in (understandably) short experimental section. However in my opinion this is not very relevant weakness; the main contribution of the paper is undoubtedly theoretical

Correctness: Seems so, although I have not read the very long appendix rigorously.

Clarity: The paper is extremely well written, and manages a clear representation of theory that requires about 30 pages in the appendix.

Relation to Prior Work: Yes

Reproducibility: Yes

Additional Feedback: In the figures 2 and 3, please mention more clearly that l is the parameter of RBF kernel; I only found from appendix what you mean by it. I believe this is usually denoted as gamma, or 2\sigma^2. This should be made more clear in the captions. In the conclusion you say that KARE gives accurate predictions of the risk for Higgs and MNIST datasets for a variety of classical kernels". I know that the space for NeurIPS submissions is very limited, but as there are only MNIST + RBF kernel results in the paper, this sentence is not accurate without referring to the appendix. Moreover, it would be interesting if you could show / mention in the paper itself what are all the "variety of classical kernels", as at the moment you have only the RBF kernel in the main paper. Moreover, continuing with this, it seems that all your experiments have been performed with shift-invariant kernels (it holds that k(x,z)=k(x-z) ). But you have not made any such assumptions for your theoretical work, right? Would your methodology work equally well with non shift-invariant kernels? ------------------- edit after rebuttal -------------------- I have read the other reviews and the author's responses. It seems that all of the issues have been adequately addressed by the authors, and I don't see a reason to change my score.

[Author Response · NeurIPS 2020]

Thanks for the thoughtful reviews, we address the reviewers comments and questions below:

**Reviewer 1:**

Regarding the Gaussianity/universality assumption, we will discuss it in terms of earlier works, in particular [Elkhalil et
al., 2020] (referenced in Review 3), and we will clarify how it is supported by the numerical experiments.

Thank you for pointing the works of [Rifkin & Lippert, 2007] on the leave-one-out error! We will cite these works and
discuss the similarities and differences with the resulting estimators, in particular, the role that the trace plays in the
KARE and the computational implications and benefits.

We will add a discussion of [Gerfo et al., 2008] in relation to the decomposition of the risk along the principal
components of the data.

**Reviewer 2:**

We agree that adding more details on the mathematical tools (in particular around the Stieltjes transform) would benefit
the audience; we found it hard to do this with the page limit, but we should be able to do this with the extra allowed
page if the paper is accepted.

About Def 1, this is our working definition; we will add a remark explaining how it arises from the kernel optimization
problem (the theorem you refer to). There is random noise on the outputs and the inputs are also random, which makes
the kernel Gram matrix a random matrix (though conceptually, we don't view the input randomness as a "noise".

Expliciting the eigendecomposition of a concrete kernel is indeed difficult; the SCT is hence more a theoretical tool
allowing one to reveal the KARE, which is not dependent on this eigendecomposition and easily computable. The
eigendecomposition of integral operator $T_K$ of the RBF kernels for Gaussian inputs is explicited on page 4 of the
Appendix; we will add a reference to this in the main (in addition to Figure 2).

We will add some comments to clarify the exposition of the Part 4, and clarify the connection with the previous parts (in
particular, how we arrive to the KARE).

We will take advantage of the extra page allowed if the paper is accepted to discuss the numerical experiments in the
main.

**Reviewer 3:**

We will improve the discussion on the Gaussianity assumption, see response to Reviewer 1 above. We will emphasize
that although the input distribution influences the observation distribution, the moments of the observations are what
matters to study the expected risk; in other words the input dependence can be understood through the lens of the
observation distribution moments. We will discuss this in relation with existing RMT results.

Regarding non-asymptotic vs data distribution agnostic results, both are crucial in our paper: (1) For the SCT, the
non-asymptotic results are indeed the main challenge (2) For the KARE, a central point of interest is the fact that it is
agnostic to the data distribution.

The effect of the dimension d is only indirect in our result indeed. You are right that convergence rates of the theorems
depends on the SCT $\vartheta$. In the RBF with Gaussian data case, the dependence of the SCT on the dimension d is made
explicit in the Appendix on page 4, where we can add a note in relation to your question.

Regarding Lemma 6 of the Appendix, you are perfectly right, we will add the references you mentioned. We will add a
few lines to clarify the difference between the kernel function $K(x, x')$, the kernel operator $K$ and the Gram matrix $G$.

The discussion on the Stieltjes should indeed be augmented, and we will do this thanks to the extra space available.

Regarding the organization, we will detail the contributions part a little more, in particular by making precise references
to the key theorems.

**Reviewer 4:**

We agree with all your comments we will update the paper accordingly.

Regarding the question about shift-invariant kernels: we did experiments with non-shift-invariant kernels and we will
add them to the appendix. You are right that our theoretical work does not assume shift-invariance.

[Meta-Review · NeurIPS 2020]

The paper was reviewed by experts on the topic and discussed after authors rebuttal. Results were found to be interesting and valuable. The reviewers comments should be taken into account while preparing the final version of the paper.